# Learning in Stackelberg Mean Field Games: A Non-Asymptotic Analysis

**Sihan Zeng**[1], **Benjamin Patrick Evans**[2], **Sujay Bhatt**[1], **Leo Ardon**[2],
**Sumitra Ganesh**[1], **Alec Koppel**[1]

[1]J.P.Morgan AI Research, United States
[2]J.P.Morgan AI Research, United Kingdom

{sihan.zeng, benjamin.x.evans, sujay.bhatt, leo.ardon,
sumitra.ganesh, alec.koppel}@jpmorgan.com

## Abstract

We study policy optimization in Stackelberg mean field games (MFGs), a hierarchical framework for modeling the strategic interaction between a single leader and an infinitely large population of homogeneous followers. The objective can be formulated as a structured bi-level optimization problem, in which the leader needs to learn a policy maximizing its reward, anticipating the response of the followers. Existing methods for solving these (and related) problems often rely on restrictive independence assumptions between the leader's and followers' objectives, use samples inefficiently due to nested-loop algorithm structure, and lack finite-time convergence guarantees. To address these limitations, we propose **AC-SMFG**, a single-loop actor-critic algorithm that operates on continuously generated Markovian samples. The algorithm alternates between (semi-)gradient updates for the leader, a representative follower, and the mean field, and is simple to implement in practice. We establish the finite-time and finite-sample convergence of the algorithm to a stationary point of the Stackelberg objective. To our knowledge, this is the first Stackelberg MFG algorithm with non-asymptotic convergence guarantees. Our key assumption is a "gradient alignment" condition, which requires that the full policy gradient of the leader can be approximated by a partial component of it, relaxing the existing leader-follower independence assumption. Simulation results in a range of well-established economics environments demonstrate that AC-SMFG outperforms existing multi-agent and MFG learning baselines in policy quality and convergence speed.

## 1 Introduction

Mean field games (MFGs) provide a framework for studying the strategic interaction among an infinite number of rational agents, with a wide range of applications in resource allocation [Li et al., 2020], telecommunication [Narasimha et al., 2019], and power system optimization [Alasseur et al., 2020]. An important extension of MFGs is Stackelberg mean field games (SMFGs), incorporating a hierarchical structure where a single leader agent influences a population of follower agents and enjoys a first-mover advantage. A prominent example of SMFG is optimal liquidation [Almgren and Chriss, 1997], in which a large institutional investor (leader) strategically sells assets, e.g., central bank treasuries, in a market while accounting for the reactions of smaller traders (followers) who adjust their behavior, e.g., bond purchasing, in response [Chen et al., 2024]. Other examples arise in public policy domains such as taxation, mortgage regulation, and epidemic control [Zheng et al., 2020, Mi et al., 2024, Aurell et al., 2022], where the leader, typically a government, designs interventions to promote social welfare while anticipating and influencing the collective response of the population.

Despite the increasing number of empirical studies applying the SMFG framework to real-world problems, there remains a lack of practical and theoretically grounded algorithms for learning in such hierarchical environments. The central challenge lies in the inherently coupled dynamics among the leader, a representative follower, and the mean field representing the collective behavior of the follower population. This interdependence gives rise to a complex bi-level structure: the leader's objective depends on the equilibrium response of the follower population, which in turn is shaped by the leader's policy. Existing approaches [Dayanıklı and Laurière, 2024, Cui et al., 2024] to SMFGs (and the closely related major-minor MFGs) need to make strong assumptions to decouple the leader and followers, lack non-asymptotic guarantees for finding a solution, and empirically exhibit slow convergence or tend towards sub-optimal limit points, possibly due to cyclic behaviors. In contrast, our work relaxes the assumptions and establishes a fast finite-sample rate of convergence for a simple actor-critic algorithm to the stationary point of the SMFG objective.

## 1.1 Main Contributions

We introduce AC-SMFG, a single-loop actor-critic algorithm for SMFGs which operates under continuously generated Markovian samples. Our first key contribution is to characterize the finite-time and finite-sample complexity of AC-SMFG. We achieve this through a multi-time-scale analysis, where the updates of the leader's policy, the follower's policy, and the mean field are coordinated on distinct time scales via properly chosen step sizes. We show that AC-SMFG converges to a stationary point of the Stackelberg objective with rate $\widetilde{\mathcal{O}}(k^{-1/2})$, under a new assumption termed "gradient alignment". Intuitively, the assumption allows the leader to improve its objective by acting to the best response from the followers. As the algorithm draws exactly two samples in each iteration, this translates to a sample complexity of the same order. This is the first time that non-asymptotic convergence guarantees are established for any SMFG algorithm, and also the first time that a single-loop SMFG algorithm is analyzed.

Notably, our convergence rate surpasses that of a comparable single-loop algorithm for bi-level optimization under lower-level strong convexity [Hong et al., 2023]

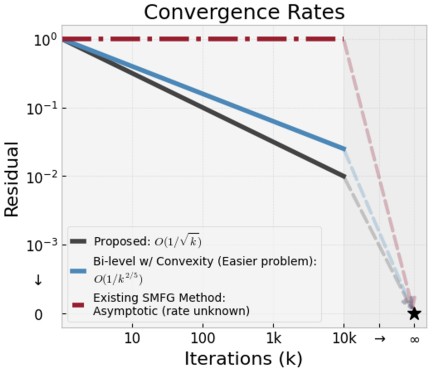

Figure 1: Theoretical convergence rates. The proposed algorithm is the first to have non-asymptotic convergence guarantees (black). Existing algorithms in similar settings are only known to converge asymptotically (red). Convergence rate of a single-loop algorithm for bi-level optimization [Hong et al., 2023] under a non-convex upper-level objective and lower-level strong convexity (blue).

$(\widetilde{\mathcal{O}}(k^{-1/2})$ vs $\widetilde{\mathcal{O}}(k^{-2/5}))$. The policy optimization problem in a SMFG can be viewed as bi-level optimization, where the upper-level objective is non-convex and the lower-level problem is to solve a standard MFG that does not exhibit any convexity. The reason that we achieve a better rate in a more challenging setting is two-fold. First, as a key technical innovation, we develop an analytical argument to handle bi-level optimization with lower-level Polyak-Lojasiewicz (PL) condition, which the MFG objective satisfies with respect to the follower's softmax policy parameter under regularization [Mei et al., 2020]. This technique allows us to obtain the same convergence rate in SMFGs as in bi-level optimization with lower-level strong convexity, and may be of independent interest and applicable to general bi-level optimization methods. Second, we make the rate improvement by incorporating the latest advances in multi-time-scale stochastic approximation [Han et al., 2024], leveraging the smoothness condition of the reward and transition.

We support the merit of proposed method through numerical simulations. We apply AC-SMFG to a range of environments inspired by real-world economic scenarios involving a prominent leader and a population of followers. Previous agent-based modeling (ABM) approaches to these problems simulate a large number of agents with complex interactions, leading to computational inefficiency and lack of theoretical guarantees [Evans et al., 2025]. In contrast, we show that the SMFG formulation offers a more tractable alternative by summarizing the collective follower behavior through the mean field, thereby enabling faster computation of a (possibly local) optimum for the leader's policy. Compared to the existing SMFG algorithms, AC-SMFG exhibits a significantly faster convergence rate.

## 1.2 Related Literature on Stackelberg Mean Field Games

(i) *Continuous-time setting*: [Djete, 2023] is among the first works to study SMFGs and to provide a characterization of the problem structure. Building on this, Dayanıklı and Laurière [2024] introduces a penalty-based approach that reformulates a SMFG as a single-level mean field control problem, offering a conceptual simplification. However, this approach does not come with a provably convergent learning algorithm, leaving open the question of how to efficiently and reliably solve SMFGs. (ii) *Discrete-time setting*: [Guo et al., 2022] consider a finite-horizon setup, and propose a minimax optimization framework (known model) for finding the policy of the leader considering a worst-case objective for the followers. Cui et al. [2024] consider finding an equilibrium in a (related, but distinct setup) of major-minor MFGs and proposes a nested-loop algorithm based on fictitious play, which approximately alternates best-response updates between the leader and the follower. Such a nested-loop structure often poses practical challenges, as it requires the inner loop (best response computation) to converge before each outer-loop update. This results in increased computational burden and reduced flexibility, particularly in large-scale or online settings where full convergence at each iteration may be infeasible or undesirable. Moreover, Cui et al. [2024] establishes only asymptotic convergence, offering limited insight into the algorithm's performance in practice. Compared to the related literature discussed above, our paper is the first work to propose a single-loop algorithm for learning equilibria in SMFGs, supported by a non-asymptotic analysis.

## 2 Stackelberg Mean Field Game: Formulation

We formulate a discrete-time infinite-horizon discounted-reward SMFG between a leader and an infinite population of homogeneous rational followers. The leader's state and action spaces are $\mathcal{S}_l$ and $\mathcal{B}$, while a representative follower's state and action spaces are $\mathcal{S}_f$ and $\mathcal{A}$. The state transition of the leader depends not only on its own action, but also the aggregate behavior of the follower population. Such an aggregate behavior is denoted by $\mu \in \Delta_{\mathcal{S}_f}$, where $\Delta_{\mathcal{S}_f}$ denotes the probability simplex over $\mathcal{S}_f$. We use $\mathcal{P}_l : \mathcal{S}_l \times \mathcal{B} \times \Delta_{\mathcal{S}_f} \to \Delta_{\mathcal{S}_l}$ to denote the transition kernel for the leader's state – $\mathcal{P}_l^\mu(s_l' \mid s_l, b)$ denotes the probability that the next state is $s_l'$ when the leader takes action $b$ under mean field $\mu$ in state $s_l$. Similarly, for the follower, we use $\mathcal{P}_f^\mu(s_f' \mid s_f, a, b)$ to denote the probability that the next follower's state is $s_f'$ when the follower takes action $a$ and the leader takes action $b$ under mean field $\mu$ in state $s_f$. We define a product state space $\mathcal{S} = \mathcal{S}_l \times \mathcal{S}_f$. For notational simplicity, we do not distinguish the leader's and follower's state in the rest of the paper and assume that $\mathcal{S}$ is the common state space observed and shared by both the leader and the representative follower.

In this way, we can compactly represent a SMFG by the tuple $(\mathcal{S}, \mathcal{A}, \mathcal{B}, \mathcal{P}, r_f, r_l, \gamma)$, where $\mathcal{P} : \mathcal{S} \times \mathcal{A} \times \mathcal{B} \times \Delta_{\mathcal{S}} \to \Delta_{\mathcal{S}}$ is the transition kernel and $\gamma \in (0, 1)$ is the discount factor. The reward function of a representative follower is $r_f : \mathcal{S} \times \mathcal{A} \times \mathcal{B} \times \Delta_{\mathcal{S}} \to [0, 1]$, with the understanding here that $\mathcal{S}$ is restricted to $\mathcal{S}_f$. Similarly, the reward of the leader $r_l : \mathcal{S} \times \mathcal{B} \times \Delta_{\mathcal{S}} \to [0, 1]$ does not depend on the state of the representative follower $s_f$, but only on the mean field. The state and action spaces of all players is assumed to be finite to enable the analysis, but continuous space approximations are studied experimentally in Section 5.

**Followers' Interaction.** The followers play a mean field game in response to a policy $\phi : \mathcal{S} \to \Delta_{\mathcal{B}}$ of the leader. As in a standard MFG, let the representative follower take actions according to a randomized policy $\pi : \mathcal{S} \to \Delta_{\mathcal{A}}$. Given a leader-follower policy pair $(\phi, \pi)$ and a mean field $\mu$, the sequentially generated states form a Markov chain, for which the transition matrix is $P^{\pi,\phi,\mu} \in \mathbb{R}^{|\mathcal{S}| \times |\mathcal{S}|}$. The matrix is entry-wise expressed as $P^{\pi,\phi,\mu}(s' \mid s) = \sum_{a \in \mathcal{A}, b \in \mathcal{B}} \mathcal{P}^\mu(s' \mid s, a, b)\pi(a \mid s)\phi(b \mid s)$. We denote by $\nu^{\pi,\phi,\mu}$ the stationary distribution of the Markov chain, which is the singular vector of $P^{\pi,\phi,\mu}$ associated with the (only) largest singular value "$1''$ (under an ergodicity assumption). Let the discounted visitation/occupancy measure $d_\rho^{\pi,\phi,\mu} \triangleq \mathbb{E}_{s_0 \sim \rho}[d_s^{\pi,\phi,\mu}]$ under the initial state distribution $\rho$, where

$$d_s^{\pi,\phi,\mu} \triangleq (1 - \gamma)\mathbb{E}_{\pi,\phi,\mathcal{P}^\mu}[\textstyle\sum_{k=0}^\infty \gamma^k \mathbf{1}(s_k = s) \mid s_0 = s],$$

where the expectation is taken over

$$a_k \sim \pi(\cdot \mid s_k), b_k \sim \phi(\cdot \mid s_k), s_{k+1} \sim \mathcal{P}^\mu(\cdot \mid s_k, a_k, b_k).$$

Under $(\phi, \pi, \mu)$, the follower expects to collect the following (regularized) cumulative reward

$$J_f(\pi, \phi, \mu) \triangleq \mathbb{E}_{\pi, \phi, \mathcal{P}^\mu} \left[ \sum_{k=0}^\infty \gamma^k \left( r_f(s_k, a_k, b_k, \mu) - \tau \log \pi(a_k \mid s_k) \right) \mid s_0 \sim \rho \right]$$
$$= \tfrac{1}{1-\gamma} \mathbb{E}_{s \sim d_\rho^{\pi, \phi, \mu}, \, a \sim \pi(\cdot \mid s), \, b \sim \phi(\cdot \mid s)} [r_f(s, a, b, \mu) + \tau E(\pi, s)], \tag{1}$$

where the entropy function $E(\pi, s) \triangleq - \sum_a \pi(a \mid s) \log \pi(a \mid s)$, and regularization weight $\tau \geq 0$. Define the follower's policy as $\pi^\star : \Delta_\mathcal{B}^\mathcal{S} \times \Delta_\mathcal{S} \to \Delta_\mathcal{A}^\mathcal{S}$ such that

$$\pi^\star(\phi, \mu) \triangleq \mathrm{argmax}_{\pi \in \Delta_\mathcal{A}^\mathcal{S}} J_f(\pi, \phi, \mu), \quad \forall \mu. \tag{2}$$

Let $\mu^\star : \Delta_\mathcal{B}^\mathcal{S} \to \Delta_\mathcal{S}$ denote the mapping from the leader's policy to the mean field induced by the leader. It is known from the literature on standard MFGs that $\mu^\star(\phi)$ satisfies for all $\phi$

$$\mu^\star(\phi) \triangleq \nu^{\pi^\star(\phi, \mu^\star(\phi)), \, \phi, \, \mu^\star(\phi)}. \tag{3}$$

With some abuse of notation, we write

$$\pi^\star(\phi) \triangleq \pi^\star(\phi, \mu^\star(\phi)), \tag{4}$$

Fixing the leader's policy to $\phi$ reduces the environment to a standard MFG among the followers. Using the terminology from the MFG literature, we refer to $(\pi^\star(\phi), \mu^\star(\phi))$ as the mean field equilibrium (MFE) for such MFG. We will later impose an assumption which guarantees that $\pi^\star(\phi)$ and $\mu^\star(\phi)$ are unique and well-defined.

**Leader's Interaction and Game Objective.** Given a leader's policy $\phi$ and the follower's mean field $\mu$, the leader's cumulative reward is

$$J_l(\phi, \mu) \triangleq \mathbb{E}_{\pi, \phi, \mathcal{P}^\mu} \left[ \sum_{k=0}^\infty \gamma^k r_l(s_k, b_k, \mu) \mid s_0 \sim \rho \right] = \tfrac{1}{1-\gamma} \mathbb{E}_{s \sim d_\rho^{\pi, \phi, \mu}, \, b \sim \phi(\cdot \mid s)} [r_l(s, b, \mu)]. \tag{5}$$

If the mean field $\mu$ were fixed, the aim of the leader would be to find a policy $\phi$ that maximizes $J_l(\phi, \mu)$. However, the stable mean field changes with $\phi$ as the followers try to best respond to the leader and each other. To solve a SMFG is to find an optimal policy for the leader, given that the followers best respond. We define $\Phi(\phi) = J_l(\phi, \mu^\star(\phi))$ for all $\phi$ and express the objective as follows

$$\phi^\star \triangleq \mathrm{argmax}_{\phi \in \Delta_\mathcal{B}^\mathcal{S}} \Phi(\phi) = \mathrm{argmax}_{\phi \in \Delta_\mathcal{B}^\mathcal{S}} J_l(\phi, \mu^\star(\phi)). \tag{6}$$

## 3 Algorithm

The algorithm developed in this work is based on the principle of **independent learning**. To motivate our approach, consider a scenario where oracle knowledge is available on $\mu^\star(\phi)$ for any $\phi$. In this case, we can optimize $\phi$ iteratively with gradient descent – we maintain $\phi_k$ (where $k$ is the iteration index) and update $\phi_k$ in the direction of the gradient $\nabla \Phi(\phi_k)$, which can be evaluated using $\mu^\star(\phi_k)$.

In practice, oracle access to $\mu^\star(\phi_k)$ is unavailable, and we must solve the lower-level MFG to approximate it. To this end, we introduce the iterates $\pi_k$ and $\hat{\mu}_k$ as estimates of $\pi^\star(\phi_k)$ and $\mu^\star(\phi_k)$, and refine them via (semi-)gradient descent. This naturally leads to a nested-loop structure, where the follower policy and mean field are updated to convergence in the inner loop to support the outer-loop gradient computation. Our approach eliminates this nested structure: we perform alternating updates of the leader, follower, and mean field in a single loop, using appropriately chosen step sizes to implicitly approximate a nested loop. The updates are formally stated in Algorithm 1.

We use a tabular softmax policy parameterization and maintain $\theta \in \mathbb{R}^{|\mathcal{S}| \times \mathcal{A}}$, $\omega \in \mathbb{R}^{|\mathcal{S}| \times \mathcal{B}}$ that encode the policies according to

$$\phi_\omega(b \mid s) = \frac{\exp(\omega(s, b))}{\sum_{b' \in \mathcal{B}} \exp(\omega(s, a'))}, \quad \pi_\theta(a \mid s) = \frac{\exp(\theta(s, a))}{\sum_{a' \in \mathcal{A}} \exp(\theta(s, a'))}.$$

The tabular softmax parameterization is considered for the purpose of mathematical analysis – optimizing over the space of softmax parameters allows us to exploit the PL structure that each player's objective observes with respect to its parameter [Mei et al., 2020]. In practice, the proposed algorithm can be applied with function approximations such as neural networks, as discussed in detail

in Remark 3. We can express the policy gradients of $\omega$ and $\theta$ in the closed form below [Sutton et al., 1999].

$$\nabla_\omega J_l(\phi_\omega, \mu) = \mathbb{E}_{\pi,\phi,\mathcal{P}^\mu}\left[\left(r_l(s,b,\mu) + \gamma V_l^{\phi_\omega,\mu}(s') - V_l^{\phi_\omega,\mu}(s)\right)\nabla_\omega \log \phi_\omega(b \mid s)\right],$$

$$\nabla_\theta J_f(\pi_\theta, \phi, \mu) = \mathbb{E}_{\pi,\phi,\mathcal{P}^\mu}\left[\left(r_f(s,a,b,\mu) - \tau \log \pi(a \mid s) + \gamma V_f^{\pi_\theta,\phi,\mu}(s') - V_f^{\pi_\theta,\phi,\mu}(s)\right)\nabla_\theta \log \pi_\theta(a \mid s)\right].$$

The policy updates (7), (8) of Algorithm 2 are exactly based on the policy gradient expressions above, substituting in the latest leader's policy parameter $\omega_k$, follower's policy parameter $\theta_k$, and mean field iterate $\hat\mu_k$. Here the value functions are defined as

$$V_l^{\phi,\mu}(s) = \mathbb{E}[\textstyle\sum_k^\infty \gamma^k r_l(s,b,\mu) \mid s_0 = s], \quad V_f^{\pi,\phi,\mu}(s) = \mathbb{E}[\textstyle\sum_k^\infty \gamma^k (r_f(s,a,b,\mu) - \tau \log \pi(a \mid s)) \mid s_0 = s].$$

Algorithm 1 requires access to $V_l^{\phi_{\omega_k},\hat\mu_k}$ and $V_f^{\pi_{\theta_k},\phi_{\omega_k},\hat\mu_k}$ for policy gradient evaluation. Since these value functions are not directly available, we estimate them by $V_{l,k}$ and $V_{f,k}$, which are updated via temporal difference (TD) learning in (10). Note that to ensure stability we leverage projections in (9) and (10): $\Pi_{\Delta_\mathcal{S}}$ denotes the projection to the probability simplex over the state space, and $\Pi_{B_V}$ denotes the element-wise projection to the interval $[0, B_V]$, where $B_V = 1/(1 - \gamma) + \tau \log |\mathcal{A}|$.

Our method exemplifies independent learning in the sense that $\omega_k, \theta_k, \hat\mu_k$ are each updated to optimize its own objective without any knowledge of the other two, though the samples used for the updates are from an environment determined by all variables. We note that the algorithm is single-loop and uses two trajectories of continuously generated samples[1], making it significantly more practical than the existing methods with complex nested loops [Dayanıklı and Laurière, 2024, Cui et al., 2024].

---

**Algorithm 1** Single loop Actor-Critic Algorithm for Stackelberg Mean Field Games (**AC-SMFG**)

---

1: **Initialize:** leader's policy parameter $\omega_0$, follower's policy parameter $\theta_0$, value function estimates $\hat{V}_{l,0}, \hat{V}_{f,0}$, mean field estimate $\hat\mu_0 \in \Delta_\mathcal{S}$, initial state $s_0, \bar{s}_0 \sim \rho$, step sizes $\zeta_k \leq \xi_k \leq \alpha_k \leq \beta_k$.
2: **for** iteration $k = 0, 1, 2, ...$ **do**
3:   **Sample path 1** for tracking the discounted occupancy measure:
    Follower and leader take actions $a_k \sim \pi_{\theta_k}(\cdot \mid s_k)$, $b_k \sim \phi_{\omega_k}(\cdot \mid s_k)$, receive rewards $r_f(s_k, a_k, b_k, \hat\mu_k)$, and observe the next state according to the transition probability

$$s_{k+1} \sim \begin{cases} \rho, & \text{with probability } 1 - \gamma \\ \mathcal{P}^{\hat\mu_k}(\cdot \mid s_k, a_k, b_k), & \text{with probability } \gamma \end{cases}$$

4:   **Sample path 2** for tracking the stationary distribution (mean field):
    Follower and leader take actions $\bar{a}_k \sim \pi_{\theta_k}(\cdot \mid \bar{s}_k)$, $\bar{b}_k \sim \phi_{\omega_k}(\cdot \mid \bar{s}_k)$, and observe the next state $\bar{s}_{k+1} \sim \mathcal{P}^{\hat\mu_k}(\cdot \mid \bar{s}_k, \bar{a}_k, \bar{b}_k)$
5:   Leader's policy update:

$$\omega_{k+1} = \omega_k + \zeta_k \nabla_\omega \log \phi_{\omega_k}(b_k \mid s_k)\Big(r_l(s_k, b_k, \hat\mu_k) + \gamma \hat{V}_{l,k}(s_{k+1}) - \hat{V}_{l,k}(s_k)\Big) \quad (7)$$

6:   Follower's policy (actor) update:

$$\theta_{k+1} = \theta_k + \alpha_k \nabla_\theta \log \pi_{\theta_k}(a_k \mid s_k)\big(r_f(s_k, a_k, b_k, \hat\mu_k) + \tau E(\pi_{\theta_k}, s_k) + \gamma \hat{V}_{f,k}(s_{k+1}) - \hat{V}_{f,k}(s_k)\big) \quad (8)$$

7:   Mean field update:

$$\hat\mu_{k+1} = \Pi_{\Delta_\mathcal{S}}\big(\hat\mu_k + \xi_k(e_{\bar{s}_k} - \hat\mu_k)\big) \quad (9)$$

8:   Value function (critic) update:

$$\hat{V}_{l,k+1} = \Pi_{B_V}\left(\hat{V}_{l,k} + \beta_k e_{s_k}\big(r_l(s_k, b_k, \hat\mu_k) + \gamma \hat{V}_{l,k}(s_{k+1}) - \hat{V}_{l,k}(s_k)\big)\right) \quad (10)$$

$$\hat{V}_{f,k+1} = \Pi_{B_V}\left(\hat{V}_{f,k} + \beta_k e_{s_k}\big(r_f(s_k, a_k, b_k, \hat\mu_k) + \tau E(\pi_{\theta_k}, s_k) + \gamma \hat{V}_{f,k}(s_{k+1}) - \hat{V}_{f,k}(s_k)\big)\right)$$

9: **end for**

---

[1]The second trajectory of samples is used solely to estimate the mean field and may be eliminated in applications where there exists a generative model/oracle for the mean field.

# 4 Convergence Analysis

In this section, we establish the convergence of AC-SMFG. We first introduce the main technical assumptions, which can be segregated into the standard ones (Assumptions 1 – 5) required for finite-sample analysis of standard MFGs, and the additional regularity assumption, Assumption 6 (termed *gradient alignment*), required to deal with the more challenging setting of SMFGs. To illustrate that the gradient alignment assumption is weaker than those in the existing literature, we provide a non-trivial example that satisfies our assumption but not the stronger ones in prior works.

**Assumption 1 (Lipschitz and Smooth Transition and Reward)** *There exist bounded constants* $L_P, L_r$ *such that for all* $\pi, \phi, \mu_1, \mu_2, s, a, b$

$$\|P^{\pi,\phi,\mu_1} - P^{\pi,\phi,\mu_2}\| \le L_P \|\mu_1 - \mu_2\|, \quad \|\nabla_\mu P^{\pi,\phi,\mu_1} - \nabla_\mu P^{\pi,\phi,\mu_2}\| \le L_P \|\mu_1 - \mu_2\|,$$
$$|r_l(s, b, \mu_1) - r_l(s, b, \mu_2)| \le L_r \|\mu_1 - \mu_2\|, \quad |r_f(s, a, b, \mu_1) - r_f(s, a, b, \mu_2)| \le L_r \|\mu_1 - \mu_2\|,$$
$$\|\nabla_\mu r_l(s, b, \mu_1) - \nabla_\mu r_l(s, b, \mu_2)\| \le L_r \|\mu_1 - \mu_2\|,$$
$$\|\nabla_\mu r_f(s, a, b, \mu_1) - \nabla_\mu r_f(s, a, b, \mu_2)\| \le L_r \|\mu_1 - \mu_2\|.$$

The Lipschitz continuity is a common assumption made in the literature on MFGs [Cui and Koeppl, 2021b, Zeng et al., 2025, Cui et al., 2024]. We additionally assume that they are smooth.

**Assumption 2 (Lipschitz Best Response)** *The best response operators* $\mu^\star$ *and* $\pi^\star$ *are Lipschitz and* $\mu^\star$ *has Lipschitz gradients, i.e. there exists a constant* $L \in [1, \infty)$ *such that for all* $\phi, \phi', \mu, \mu'$

$$\|\mu^\star(\phi) - \mu^\star(\phi')\| \le L \|\phi - \phi'\|, \quad \|\nabla_\phi \mu^\star(\phi) - \nabla_\phi \mu^\star(\phi')\| \le L \|\phi - \phi'\|, \tag{11}$$
$$\|\pi^\star(\phi, \mu) - \pi^\star(\phi', \mu')\| \le L(\|\phi - \phi'\| + \|\mu - \mu'\|).$$

The condition (11) in fact can be shown to follow from the Lipschitz continuity and smoothness of the transition and reward function in Assumption 1. We impose Assumption 2 directly for simplicity.

**Assumption 3 (Exploration)** *There exists a constant* $\rho_{\min} > 0$ *such that the initial state distribution satisfies* $\rho(s) \ge \rho_{\min}, \forall s \in \mathcal{S}$. *In addition, the follower's policy iterates are uniformly bounded away from zero, i.e. there exists a constant* $p_{\min}$ *such that* $\pi_{\theta_k}(a \mid s) \ge p_{\min}$ *for all* $k \ge 0, s \in \mathcal{S}, a \in \mathcal{A}$.

The first part of Assumption 3 implies that the discounted occupancy measure $d_\rho^{\pi,\phi,\mu}$ is bounded away from zero for any $\pi, \phi, \mu$ and is a standard assumption in RL [Agarwal et al., 2021, Mei et al., 2020], while the second part can be readily satisfied by capping the norm of $\theta_k$ away from infinity. The assumption guarantee sufficient exploration of every state and action pair.

**Assumption 4 (Contraction)** *There exists a constant* $\delta \in (0, 1)$ *such that for any* $\phi, \mu$

$$\|\nu^{\pi^\star(\phi,\mu_1),\phi,\mu_1} - \nu^{\pi^\star(\phi,\mu_2),\phi,\mu_2}\| \le \delta \|\mu_1 - \mu_2\|. \tag{12}$$

Assumption 4 states that in a standard MFG determined by any fixed $\phi$, an "optimality-consistency operator contraction condition" holds, which implies the uniqueness of the MFE. This is a key assumption made in the existing literature on MFGs [Xie et al., 2021, Zaman et al., 2023, Yardim et al., 2023] and can be guaranteed when the regularization weight $\tau$ is large enough.

**Assumption 5 (Uniform Geometric Ergodicity)** *For any* $\pi, \phi, \mu$, *the Markov chain* $\{s_t\}$ *generated by* $P^{\pi,\mu}$ *according to* $s_{t+1} \sim P^{\pi,\phi,\mu}(\cdot \mid s_t)$ *is irreducible and aperiodic. In addition, we have*

$$\sup_s d_{TV}\big(\mathbb{P}(s_t = \cdot \mid s_0 = s), \nu^{\pi,\phi,\mu}\big) \le C_0 C_1^t, \forall t \ge 0,$$

*for some constants* $C_0 \ge 1$ *and* $C_1 \in (0, 1)$, *where* $d_{TV}$ *denotes the total variation distance.*

Assumption 5 is again standard in analyzing stochastic approximation and RL algorithms under Markovian samples [Zou et al., 2019, Wu et al., 2020, Wang et al., 2024].

**Assumption 6 (Gradient Alignment)** *There exist positive bounded constants* $\eta_1, \eta_2$ *such that* $\forall \omega$

$$\eta_1 \langle \nabla_\omega \Phi(\phi_\omega), \nabla_\omega J_l(\phi_\omega, \mu) \mid_{\mu=\mu^\star(\phi_\omega)} \rangle \ge \|\nabla_\omega \Phi(\phi_\omega)\|^2, \tag{13}$$
$$\eta_2 \langle \nabla_\omega \Phi(\phi_\omega), \nabla_\omega J_l(\phi_\omega, \mu) \mid_{\mu=\mu^\star(\phi_\omega)} \rangle \ge \|\nabla_\omega J_l(\phi_\omega, \mu) \mid_{\mu=\mu^\star(\phi_\omega)} \|^2. \tag{14}$$

Note that $\nabla_\omega J_l(\phi_\omega, \mu) \mid_{\mu = \mu^\star(\phi_\omega)}$ is a partial component of the full gradient $\nabla_\omega \Phi(\phi_\omega)$. The condition states that the two gradients always make an acute angle. Without loss of generality, we assume $\eta_1, \eta_2 \geq 1$, as if the conditions hold with $\eta_1, \eta_2 < 1$ they also hold with $\eta_1, \eta_2 = 1$. Conceptually, the condition allows the leader to improve its objective by acting to the best response from the followers. Prior works on SMFGs and the closely related major-minor MFGs need to make assumptions of a similar nature, but more restrictive. For example, Cui et al. [2024] by their Assumptions 4.c) and 4.d) assumes that the leader's reward is independent of the mean field and that the follower's reward and transition are independent of the leader's action. Under such conditions, the leader and follower are effectively decoupled, reducing the hierarchical structure of the problem to a near-independent setting. In contrast, with the gradient alignment assumption we permit our leader to depend on $\mu$, and follower rewards and transitions to depend on $b$. In Appendix E, we present a non-trivial MFG which satisfies the gradient alignment assumption but not the leader-follower independence condition in Cui et al. [2024].

## 4.1 Algorithm Complexity

Each variable in Algorithm 1 converges in the sense that an associated residual (measure of sub-optimality gap) decays to zero. The residual of the leader is the squared policy gradient norm. The mean field convergence is measured by its deviation from the (unique) equilibrium of a standard MFG induced by leader's latest policy iterate. The convergence of the follower is assessed by the objective function gap relative to the best response against $\pi_k, \hat{\mu}_k$. Finally, the value function estimates are evaluated by their $\ell_2$ distance to the true value functions under the latest policy and mean field iterates. The proof of Theorem 1 relies on analyzing the convergence of all variables jointly, through a coupled Lyapunov function.

$$\varepsilon_k^\phi = \|\nabla_\omega \Phi(\phi_{\omega_k})\|^2, \quad \varepsilon_k^\pi = J_f(\pi^\star(\phi_{\omega_k}, \hat{\mu}_k), \phi_{\omega_k}, \hat{\mu}_k) - J_f(\pi_{\theta_k}, \phi_{\omega_k}, \hat{\mu}_k),$$
$$\varepsilon_k^\mu = \|\hat{\mu}_k - \mu^\star(\phi_{\omega_k})\|^2, \quad \varepsilon_{l,k}^V = \|\hat{V}_{l,k} - V_l^{\phi_{\omega_k}, \hat{\mu}_k}\|^2, \quad \varepsilon_{f,k}^V = \|\hat{V}_{f,k} - V_f^{\pi_{\theta_k}, \phi_{\omega_k}, \hat{\mu}_k}\|^2. \tag{15}$$

**Theorem 1** *Consider the iterates of Algorithm 1 under the step sizes*

$$\zeta_k = \frac{c_\zeta}{(k+1)^{1/2}}, \quad \alpha_k = \frac{c_\alpha}{(k+1)^{1/2}}, \quad \xi_k = \frac{c_\xi}{(k+1)^{1/2}}, \quad \beta_k = \frac{c_\beta}{(k+1)^{1/2}},$$

*with the properly selected constants $c_\zeta, c_\alpha, c_\xi, c_\beta$. Under Assumptions 1-5, we have for all $k > 0$,*

$$\min_{t < k} \mathbb{E}[\varepsilon_t^\phi] \leq \widetilde{\mathcal{O}}\left(\frac{1}{(k+1)^{\frac{1}{2}}}\right).$$

As the residuals are all non-negative, Theorem 1 implies that the best iterate of Algorithm 1 converges to a stationary point of the leader's objective with rate $\widetilde{\mathcal{O}}(k^{-1/2})$, where $\widetilde{\mathcal{O}}$ hides constants and logarithmic factors of $k$. As exactly two samples are drawn in each iteration, this translates to a sample complexity of the same order. To our knowledge, this is the first result providing a finite-time and sample complexity for learning in Stackelberg/major-minor mean field games. In contrast to the most relevant prior work [Cui et al., 2024], which establishes the asymptotic convergence of a nested-loop fictitious play method, our paper analyzes a single-loop algorithm and provides non-asymptotic guarantees. This rate improves upon that of a two-time-scale algorithm for bi-level optimization with a non-convex upper-level objective and lower-level strong convexity [Hong et al., 2023], which achieves $\widetilde{\mathcal{O}}(1/k^{2/5})$. Our improvement is obtained in a more challenging setting – the lower-level problem is not a strongly convex function but a MFG involving two coupled variables: the follower's policy and the mean field. The MFG does not exhibit any convex structure, and only a non-uniform PL condition holds with respect to the follower's policy under regularization. A key technical innovation made in our work is to handle such non-convexity, which we sketch in Remark 2, and which we believe is of independent interest and applicable to the analysis of general bi-level optimization methods with lower-level PL condition.

Note that our analysis relies on a sufficiently large regularization weight $\tau$ to ensure that Assumption 4 is satisfied. As a result, our method does not find the equilibrium of the original (unregularized) problem. In fact, solving unregularized MFGs – even in the standard, non-hierarchical setting – remains an open problem. Our work inherits this fundamental limitation.

**Remark 1** *It is known in the standard RL literature that the cumulative reward observes a "gradient domination" condition with respect to the policy parameter. Using the notation of our paper, it means*

*that any stationary point (where $\nabla_\omega J_l(\phi_\omega, \mu) = 0$) is globally optimal under a fixed $\mu$. We are not able to extend the argument here and have to settle for convergence to a stationary point, because $\Phi$ is a composite function involving $\mu^\star$ and hence does not observe the gradient domination condition.*

**Remark 2** *Two techniques allow us to achieve the convergence rate of $\widetilde{\mathcal{O}}(k^{-1/2})$, surpassing that in Hong et al. [2023] under lower-level strong convexity.*

*To analyze a bi-level optimization algorithm, we need to bound the residual of the follower's policy and show its iteration-wise reduction. If the lower-level objective $J_f$ were strongly convex, we would select the residual to be $\tilde{\epsilon}_k^\pi = \|\pi_{\theta_k} - \pi^\star(\phi_{\omega_k}, \hat{\mu}_k)\|^2$ and show that $\tilde{\epsilon}_{k+1}^\pi - \tilde{\epsilon}_k^\pi$ is approximately negative. As the learning target shifts from iteration $k$ to $k + 1$, bounding $\tilde{\epsilon}_{k+1}^\pi - \tilde{\epsilon}_k^\pi$ requires controlling the drift $\|\pi^\star(\phi_{\omega_{k+1}}, \hat{\mu}_{k+1}) - \pi^\star(\phi_{\omega_k}, \hat{\mu}_k)\|^2$, which can be easily bounded by*

$$\mathcal{O}(\|\phi_{\omega_{k+1}} - \phi_{\omega_k}\|^2 + \|\hat{\mu}_{k+1} - \hat{\mu}_k\|^2) \tag{16}$$

*under the Lipschitz continuity of $\pi^\star$. However, without strong convexity, we cannot consider the residual $\tilde{\epsilon}_k^\pi$ (as $\pi^\star$ may not be unique in general) and instead employ $\epsilon_k^\pi$ defined in (15), a residual in the value function space. We would like to bound the learning target shift $J_f(\pi^\star(\phi_{\omega_{k+1}}, \hat{\mu}_{k+1}), \phi_{\omega_{k+1}}, \hat{\mu}_{k+1}) - J_f(\pi^\star(\phi_{\omega_k}, \hat{\mu}_k), \phi_{\omega_k}, \hat{\mu}_k)$, preferably still by (16) to avoid rate deterioration. However, the Lipschitz continuity of $J_f$ only yields a loose bound on the order of $\mathcal{O}(\|\phi_{\omega_{k+1}} - \phi_{\omega_k}\| + \|\hat{\mu}_{k+1} - \hat{\mu}_k\|)$, where the norms are not squared. Detailed in Proposition 3 (proof in Section C.3), we develop a careful error decomposition scheme to break down the residual difference $\epsilon_{k+1}^\pi - \epsilon_k^\pi$, and tightly bound the decomposed terms based on the PL condition, thereby avoiding the suboptimal bound implied by naive application of the Lipschitz property of $J_f$.*

*The technique highlighted above allows us to achieve the same convergence rate in SMFGs as in bi-level optimization with lower-level strong convexity. We further improve upon the rate in Hong et al. [2023], leveraging a recent advance in the multi-time-scale stochastic approximation literature. Specifically, Han et al. [2024] shows that a faster convergence rate can be achieved under lower-level strong convexity, if the lower-level learning target (equivalent to operators $\mu^\star, \pi^\star$ in our context) has Lipschitz gradients. Our work adapts and extends the argument to the case where the lower-level objective is nonconvex.*

## 5 Experiments

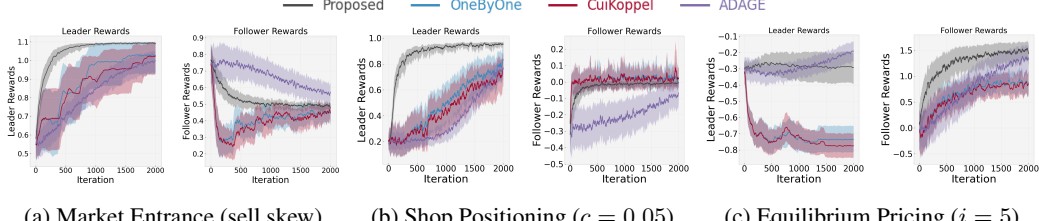

(a) Market Entrance (sell skew)     (b) Shop Positioning ($c = 0.05$)     (c) Equilibrium Pricing ($i = 5$)

Figure 2: Convergence across environments, demonstrating bootstrapped mean and 5% confidence interval across 30 runs for the leader reward (left) and follower rewards (right).

We conduct a comprehensive evaluation of the proposed methodology across a diverse set of canonical MFGs. Specifically, we extend three environments from MFGLib [Guo et al., 2023a], each exhibiting varying degrees of complexity. These environments, ordered by increasing difficulty, include Left-Right, Beach Bar, and Equilibrium pricing [Guo et al., 2023b], with descriptions in Appendix F. We also include in the appendix additional plots visualizing the evolution of the mean field as learning proceeds. All source code is available in the supplementary material.

**Comparisons.** The proposed approach is compared to the existing state-of-the-art methods for the discrete time setting: 1) a nested-loop MFG algorithm which alternates between training the leader and representative follower to convergence (OneByOne); 2) a weighted OneByOne update as presented in Algorithm 1 of [Cui et al., 2024] (CuiKoeppl); 3) a MARL implementation, where the leader and followers are updated simultaneously with PPO, as presented in [Evans et al., 2025] (ADAGE). In the MARL setting, there are two policies, one for the leader, and a shared policy [Vadori et al., 2020] for the followers, to keep the representation as fair as possible when compared to mean fields, and to prevent the need to learn $N$ independent follower policies.

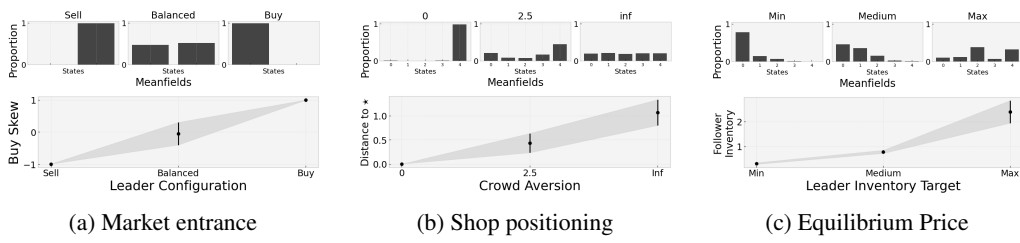

| (a) Market entrance | (b) Shop positioning | (c) Equilibrium Price |

Figure 3: Equilibria Analysis

**Market Entrance.** The first environment is a market entrance scenario inspired by the Left-Right game [Cui and Koeppl, 2021a], akin to minority games in physics and El Farol problems in complex systems, as detailed in Appendix F.1. Figure 3a illustrates the impact of various leader configurations, achieved by altering the leader's objective, demonstrating how the leader learns to execute optimal actions while followers adapt effectively, steering the mean field towards the target state (see also Figure 6). The top row of Figure 3a highlights the leader's influence on the resulting mean fields, while the bottom row depicts the market skew. The leader shifts the market dynamics to align with the desired behavior. In each scenario, we observe convergence towards a suitable MFE, with the MF adjusting appropriately to the task at hand. These findings underscore how follower behavior successfully adapts in response to the leader's actions, consistently achieving appropriate equilibria.

**Shop Positioning.** The shop positioning environment (Appendix F.2) is modeled after the beach bar MFG, where followers strive to position themselves near a desirable location $\star$ while avoiding crowded areas, and a leader is tasked with establishing a new location to compete with $\star$. We investigate how varying the crowd aversion parameter $c$ among followers influences both the resulting mean field and the leader's chosen location. When crowd aversion is absent ($c = 0$), the leader strategically places the new location at the same spot as $\star$, capitalizing on the existing customer base, aligning with known models of spatial competition where competitors cluster nearby (Figure 3b left). As followers' focus shifts entirely to crowd aversion ($c \to \infty$, Figure 3b right), the leader places the shop uniformly across the state space, preventing any single state from becoming overly crowded. For intermediate crowd aversions ($0 < c < \infty$), we observe a spectrum of behaviors, balancing between the two limiting equilibria. In each scenario, the MFE is appropriately adjusted. These findings demonstrate how the leader's actions adapt automatically in response to the followers' behavior.

**Equilibrium Price.** Finally, we examine an equilibrium pricing environment (Appendix F.3), as outlined in [Guo et al., 2023a,b, Cousin et al., 2011]. In this environment, followers are homogeneous firms producing an identical product, with prices determined by the endogenous supply-demand equilibrium. These firms must effectively manage their inventory, production, and replenishment processes, and the leaders role is to incentivize firms to maintain a specific target inventory level. We explore various target scenarios, ranging from promoting lean operations with minimal inventories (suitable for predictable periods with stable demand) to encouraging robust firms with substantial inventory reserves (to withstand crises or demand surges). As illustrated in Figure 3c, the leader can effectively shift the mean field of the followers to achieve the desired inventory levels. This demonstrates the leader's ability to implement appropriate penalties or bonuses to guide followers towards either building a buffer (Figure 3c right) or maintaining leaner operations (Figure 3c left).

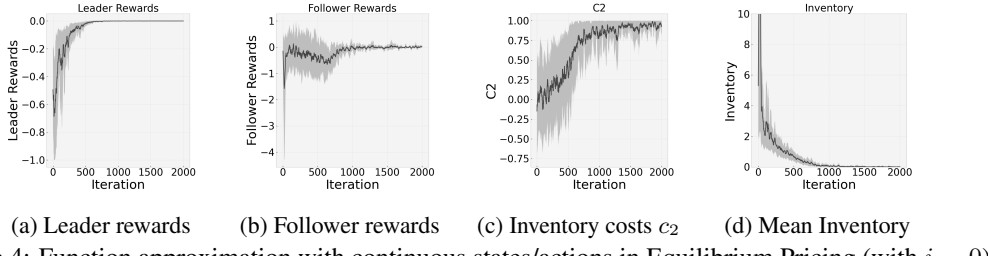

| (a) Leader rewards | (b) Follower rewards | (c) Inventory costs $c_2$ | (d) Mean Inventory |

Figure 4: Function approximation with continuous states/actions in Equilibrium Pricing (with $i = 0$)

**Key simulation findings.** The convergence of the leader and follower rewards is compared in Figure 2. Overall, the proposed approach affords significantly faster leader convergence, and smoother convergence trajectories, characterized by reduced variability across runs and diminished intra-run

volatility. Notably, the more the followers' behavior is influenced by the leader's actions, the greater the enhancement observed with the proposed approach. Breaking down the results per environment, we see significantly faster convergence for the proposed approach in the market entrance game than all the comparisons (Figure 2a), while also resulting in an improved final equilibria (here, governed by the leaders reward which attempts to shape the equilibrium). In the shop positioning game (Figure 2b), we see significantly faster convergence for the leader, albeit with slightly slower follower convergence, yet still showing substantial improvements over the MARL scenario for both leaders and followers. The slower follower convergence here is because the leader has a smaller impact on the followers behaviour, due to their being the fixed desirable location, so only a small part of the environment is changing (the additional location). In the equilibrium pricing game (Figure 2c), the proposed approach yields significantly better leader and follower rewards than existing SMFG algorithms. This environment is particularly challenging and exceeds the representational capacity of purely tabular methods due to the continuous state and action spaces. ADAGE leverages function approximation, parameterizing the leader's and followers' policies by a neural neural work. We discuss in Remark 3 how function approximation integrates with the proposed SMFG approach – we can parameterize the policies of the leader and representative follower by a neural network in a standard way, whereas for the mean field we make a distributional assumption and learn the parameters of the distribution. Specifically in the equilibrium pricing experiment, we assume that the mean field follows a Gaussian distribution and updates the mean and standard deviation of the mean field in AC-SMFG. The assumption may not be exactly accurate, which is a possible cause of the gap in leader's rewards between ADAGE and AC-SMFG. Nevertheless, the approximation suffices to yield competitive performance, and the proposed algorithm closely tracks the performance of ADAGE and outperforms other SMFG baselines.

**Remark 3 (Continuous State and Action Spaces & Function approximation)** *In Algorithm 1, we see that there are three key features that can be substituted with function approximation techniques rather than the exact update schemes presented: 1) the updates of the policies, 2) the updates of the value functions, 3) the updates of the mean field. Here, we show how each of these can be achieved. For the leader and follower, we can convert the tabular policy and value function into continuous representations with neural network function approximations (with an actor network and critic network) – this is relatively straightforward and similar to how continuous state and action spaces are handled in standard RL. What poses more challenge is the mean field update: in the tabular case we track the mean field by a probability vector of dimension $|\mathcal{S}|$, but this does not scale to the large/continuous state space setting. For simplicity, in the experiments of this paper, we consider a Gaussian mean field $\mathcal{N}(\nu, \sigma)$ and update the distribution parameters towards the sampled state $s$ with*

$$\nu_{k+1} = \nu_k + \xi_k(s - \nu_k), \quad \sigma_{k+1} = \sigma_k + \xi_k((s - \nu_{k+1})^2 - \sigma_k).$$

*However, more complex schemes could be utilized. For demonstration, we take the Equilibrium Price environment and assume that the leader takes a continuous action $c_2 \in [-1, 1]$ to set the cost term, and the follower's inventory (and thus the state space) is positive, ordinal, and unbounded. An overview of the resulting learning dynamics is displayed in Figure 4, which shows rapid learning and matches or even surpasses the results shown in Figure 8a of the appendix. This highlights the compatibility and effectiveness of our approach when integrated with function approximation in complex environments, underscoring the practical versatility and applicability of our method across diverse settings.*

## 6 Conclusion & Discussions

We proposed a single-loop actor-critic algorithm for learning equilibria in SMFGs, and provided the first known non-asymptotic analysis. Numerical simulations show the algorithm's superior performance in various realistic problems, highlighting its potential for practical applications in hierarchical decision-making environments. Our findings contribute to the growing body of research on SMFGs, offering a scalable and theoretically grounded solution for learning equilibria in complex SMFGs.

Finally, we acknowledge certain limitations of the SMFG framework in modeling practical problems, particularly when compared to agent-based modeling (ABM) approaches that simulate a finite number of followers directly. One limitation stems from the assumption of homogeneity among followers, which is embedded in the mean field formulation. Another key consideration is that MFGs serve as asymptotic approximations of $N$-agent Markov games, with the approximation error diminishing as $N$ increases. In scenarios involving a small number of follower agents, the approximation error may become non-negligible, and alternative modeling paradigms such as ABM may be more appropriate.

## Disclaimer

This paper was prepared for informational purposes by the Artificial Intelligence Research group of JPMorgan Chase & Co. and its affiliates ("JP Morgan") and is not a product of the Research Department of JP Morgan. JP Morgan makes no representation and warranty whatsoever and disclaims all liability, for the completeness, accuracy or reliability of the information contained herein. This document is not intended as investment research or investment advice, or a recommendation, offer or solicitation for the purchase or sale of any security, financial instrument, financial product or service, or to be used in any way for evaluating the merits of participating in any transaction, and shall not constitute a solicitation under any jurisdiction or to any person, if such solicitation under such jurisdiction or to such person would be unlawful.

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

# Appendix

## Contents

## A    Notations and Frequently Used Inequalities

We introduce some shorthand notations frequently used in the rest of the paper. The operators below abstract the (semi-)gradient of the leader, follower, mean field, and value functions.

$$D(\omega, \mu, V_l, s, b, s') = \nabla_\omega \log \phi_\omega(b \mid s)\Big(r_l(s, b, \mu) + \gamma V_l(s') - V_l(s)\Big),$$

$$F(\theta, \mu, V_f, s, a, b, s') = \nabla_\theta \log \pi_\theta(a \mid s)\Big(r_f(s, a, b, \mu) + \tau E(\pi_\theta, s) + \gamma V_f(s') - V_f(s)\Big),$$

$$H(\mu, s) = e_s - \mu, \tag{17}$$

$$G_l(\omega, \mu, V_l, s, a, b, s') = e_s\Big(r_l(s, b, \mu) + \gamma V_l(s') - V_l(s)\Big),$$

$$G_f(\theta, \mu, V_f, s, a, b, s') = e_s\Big(r_f(s, a, b, \mu) + \tau E(\pi_\theta, s) + \gamma V_f(s') - V_f(s)\Big).$$

Given the notations in (17), we can re-write the variable updates in Algorithm 1.

$$\omega_{k+1} = \omega_k + \zeta_k D(\omega_k, \hat{\mu}_k, \hat{V}_{l,k}, s_k, b_k, s'_k), \tag{18}$$

$$\theta_{k+1} = \theta_k + \alpha_k F(\theta_k, \hat{\mu}_k, \hat{V}_{f,k}, s_k, a_k, b_k, s'_k), \tag{19}$$

$$\hat{\mu}_{k+1} = \Pi_{\Delta_S}\Big(\hat{\mu}_k + \xi_k H(\hat{\mu}_k, \bar{s}_k)\Big), \tag{20}$$

$$\hat{V}_{l,k+1} = \Pi_{B_V}\Big(\hat{V}_{l,k} + \beta_k G_l(\omega_k, \hat{\mu}_k, \hat{V}_{l,k}, s_k, a_k, b_k, s'_k)\Big), \tag{21}$$

$$\hat{V}_{f,k+1} = \Pi_{B_V}\Big(\hat{V}_{f,k} + \beta_k G_f(\theta_k, \hat{\mu}_k, \hat{V}_{f,k}, s_k, a_k, b_k, s'_k)\Big). \tag{22}$$

We also define the expected versions of the (semi-)gradient operators, where the expectation is taken over the stochastic samples from the stationary distribution.

$$\bar{D}(\omega, \mu, V_l) \triangleq \mathbb{E}_{s \sim d_\rho^{\pi, \phi_\omega, \mu}, a \sim \pi(\cdot|s), b \sim \phi_\omega(\cdot|s), s' \sim \mathcal{P}^\mu(\cdot|s, a, b)}[D(\omega, \mu, V_l, s, b, s')], \tag{23}$$

$$\bar{F}(\theta, \omega, \mu, V_f) \triangleq \mathbb{E}_{s \sim d_\rho^{\pi_\theta, \phi_\omega, \mu}, a \sim \pi_\theta(\cdot|s), b \sim \phi_\omega(\cdot|s), s' \sim \mathcal{P}^\mu(\cdot|s, a, b)}[F(\theta, \mu, V_f, s, a, b, s')], \tag{24}$$

$$\bar{H}(\pi, \omega, \mu) \triangleq \mathbb{E}_{\bar{s} \sim \nu^{\pi, \phi_\omega, \mu}}[H(\mu, \bar{s})], \tag{25}$$

$$\bar{G}_l(\omega, \mu, V_l) \triangleq \mathbb{E}_{s \sim d_\rho^{\pi, \phi_\omega, \mu}, a \sim \pi(\cdot|s), b \sim \phi_\omega(\cdot|s), s' \sim \mathcal{P}^\mu(\cdot|s, a, b)}[G_l(\omega, \mu, V_l, s, a, b, s')], \tag{26}$$

$$\bar{G}_f(\theta, \omega, \mu, V_f) \triangleq \mathbb{E}_{s \sim d_\rho^{\pi_\theta, \phi_\omega, \mu}, a \sim \pi_\theta(\cdot|s), b \sim \phi_\omega(\cdot|s), s' \sim \mathcal{P}^\mu(\cdot|s, a, b)}[G_f(\theta, \mu, V_f, s, a, b, s')]. \tag{27}$$

We re-iterate that due to the state separation discussed in Section 2, $\bar{D}$ and $\bar{G}_l$ are independent of the single follower's policy $\pi$, though $\pi$ appears on the right hand side of (23) and (26).

We define the filtration $\mathcal{F}_k = \{s_0, a_0, b_0, s'_0 \cdots, s_k, a_k, b_k, s'_k, \bar{s}_0, \cdots, \bar{s}_k\}$.

Unless otherwise noted, we use $\| \cdot \|$ to denote the $\ell_2$ norm of a vector and the operator norm of a matrix.

We may use $\nabla_\omega \Phi(\phi_\omega)$ and $\nabla_\omega J_l(\phi_\omega, \mu^\star(\phi_\omega))$ interchangeably in the rest of the paper, which is the true gradient of the leader. The partial gradient is denoted by $\nabla_\omega J_l(\phi_\omega, \mu) \mid_{\mu = \mu^\star(\phi_\omega)}$.

We also introduce a few frequently used lemmas. The first is on the Lipschitz continuity and smoothness of value functions.

**Lemma 1** *We have for any $\theta, \theta', \omega, \omega', \mu, \mu'$*

$$|J_f(\pi_\theta, \phi_\omega, \mu) - J_f(\pi_{\theta'}, \phi_{\omega'}, \mu')| \leq L_V(\|\theta - \theta'\| + \|\phi_\omega - \phi_{\omega'}\| + \|\mu - \mu'\|), \qquad (28)$$

$$\|V_f^{\pi_\theta, \phi_\omega, \mu} - V_f^{\pi_{\theta'}, \phi_{\omega'}, \mu'}\| \leq L_V(\|\theta - \theta'\| + \|\phi_\omega - \phi_{\omega'}\| + \|\mu - \mu'\|), \qquad (29)$$

$$\|\nabla_\theta V_f^{\pi_\theta, \phi_\omega, \mu} - \nabla_\theta V_f^{\pi_{\theta'}, \phi_\omega, \mu}\| \leq L_{VV}\|\theta - \theta'\|, \ \|\nabla_\omega V_f^{\pi_\theta, \phi_\omega, \mu} - \nabla_\omega V_f^{\pi_\theta, \phi_{\omega'}, \mu}\| \leq L_{VV}\|\omega - \omega'\|,$$
$$(30)$$

$$\|\nabla_\mu V_f^{\pi_\theta, \phi_\omega, \mu} - \nabla_\mu V_f^{\pi_\theta, \phi_\omega, \mu'}\| \leq L_{VV}\|\mu - \mu'\|, \qquad (31)$$

$$|J_l(\phi_\omega, \mu) - J_l(\phi_{\omega'}, \mu')| \leq L_V(\|\omega - \omega'\| + \|\mu - \mu'\|), \qquad (32)$$

$$\|V_l^{\phi_\omega, \mu} - V_l^{\phi_{\omega'}, \mu'}\| \leq L_V(\|\omega - \omega'\| + \|\mu - \mu'\|), \qquad (33)$$

$$\|\nabla_\omega V_l^{\phi_\omega, \mu} - \nabla_\omega V_l^{\phi_{\omega'}, \mu}\| \leq L_{VV}\|\omega - \omega'\|, \ \|\nabla_\mu V_l^{\phi_\omega, \mu} - \nabla_\mu V_l^{\phi_\omega, \mu'}\| \leq L_{VV}\|\mu - \mu'\|, \quad (34)$$

*where $L_V = \max\left\{\frac{2}{(1-\gamma)^2}, \frac{L_P \gamma \sqrt{|\mathcal{S}|}}{(1-\gamma)^2} + \frac{L_r}{1-\gamma}\right\}$ and $L_{VV} = \max\left\{\frac{8}{(1-\gamma)^3}, \frac{2\gamma^2 \sqrt{|\mathcal{S}|}L_P^2}{(1-\gamma)^3} + \frac{2\gamma \sqrt{|\mathcal{S}|}L_P}{(1-\gamma)^2} + 2\gamma L_P L_r\right\}$.*

The second lemma guarantees the Lipschitz continuity of the stationary distribution (of the current and next states jointly) with respect to the policies and mean field.

**Lemma 2** *For any $\theta_1, \theta_2, \omega_1, \omega_2, \mu_1, \mu_2$, we have*

$$\left\| \sum_s \left( d_\rho^{\pi_{\theta_1}, \phi_{\omega_1} \mu_1}(s) P^{\pi_{\theta_1}, \phi_{\omega_1} \mu_1}(\cdot \mid s) - d_\rho^{\pi_{\theta_2}, \phi_{\omega_2} \mu_2}(s) P^{\pi_{\theta_2}, \phi_{\omega_2} \mu_2}(\cdot \mid s) \right) \right\|$$

$$\leq \frac{1}{1-\gamma}\left( L_P\|\mu_1 - \mu_2\| + \|\theta_1 - \theta_2\| + \|\omega_1 - \omega_2\| \right).$$

The next lemma establishes the boundedness of the (semi-)gradient operators introduced in (17).

**Lemma 3** *Recall that $B_V$ is the entry-wise upper bound on the magnitude of the value function. We define the constants*

$$B_D = 2(1+\gamma)B_V + 2, \quad B_F = 2(1+\gamma)B_V + 2\tau \log |\mathcal{A}| + 2,$$
$$B_H = 2, \quad B_G = (1+\gamma)B_V + \tau \log |\mathcal{A}| + 1. \tag{35}$$

*We have for all $\theta, \omega, \mu, V_l, V_f, s, a, b, s'$*

$$\|D(\omega, \mu, V_l, s, b, s')\| \leq B_D, \quad \|F(\theta, \mu, V_f, s, a, b, s')\| \leq B_F,$$
$$\|H(\mu, s)\| \leq B_H, \quad \|G_l(\omega, \mu, V_l, s, b, s')\| \leq B_G, \quad \|G_f(\theta, \mu, V_f, s, a, b, s')\| \leq B_G.$$

Lemma 4 further shows that the (semi-)gradient operators introduced in (17) are Lipschitz.

**Lemma 4** *We define the constants*

$$L_D = L_F = \max\left\{ 1 + \tau \log |\mathcal{A}| + (1+\gamma)B_V + \frac{(4 + 8\log|\mathcal{A}|)\tau}{(1-\gamma)^3} + \frac{B_F}{1-\gamma}, 2L_r + \frac{B_F L_P}{1-\gamma}, \frac{B_F}{1-\gamma}, 4 \right\},$$

$$L_H = \max\left\{ \frac{L_P}{1-\gamma}, 1 \right\}, \quad L_G = \max\left\{ \frac{(4 + 8\log|\mathcal{A}|)\tau}{(1-\gamma)^3} + \frac{B_G}{1-\gamma}, L_r + \frac{B_G L_P}{1-\gamma}, \frac{B_G}{1-\gamma}, 2 \right\}.$$
$$(36)$$

*We have for all $\theta_1, \theta_2, \pi_1, \pi_2, \omega_1, \omega_2, \mu_1, \mu_2, V_1, V_2$*

$$\|\bar{D}(\omega_1, \mu_1, V_1) - \bar{D}(\omega_2, \mu_2, V_2)\| \leq L_D \left( \|\omega_1 - \omega_2\| + \|\mu_1 - \mu_2\| + \|V_1 - V_2\| \right),$$

$$\|\bar{F}(\theta_1, \omega_1, \mu_1, V_1) - \bar{F}(\theta_2, \omega_2, \mu_2, V_2)\| \leq L_F \left( \|\theta_1 - \theta_2\| + \|\omega_1 - \omega_2\| + \|\mu_1 - \mu_2\| + \|V_1 - V_2\| \right),$$

$$\|\bar{H}(\pi_1, \omega_1, \mu_1) - \bar{H}(\pi_2, \omega_2, \mu_2)\| \leq L_H \left( \|\pi_1 - \pi_2\| + \|\omega_1 - \omega_2\| + \|\mu_1 - \mu_2\| \right),$$

$$\|\bar{G}_f(\theta_1, \omega_1, \mu_1, V_1) - \bar{G}_f(\theta_2, \omega_2, \mu_2, V_2)\| \leq L_G \left( \|\theta_1 - \theta_2\| + \|\omega_1 - \omega_2\| + \|\mu_1 - \mu_2\| + \|V_1 - V_2\| \right),$$

$$\|\bar{G}_l(\omega_1, \mu_1, V_1) - \bar{G}_l(\omega_2, \mu_2, V_2)\| \leq L_G \left( \|\omega_1 - \omega_2\| + \|\mu_1 - \mu_2\| + \|V_1 - V_2\| \right).$$

---

**Algorithm 2** Actor-Critic Algorithm for Hierarchical Mean Field Games (Simplified for Analysis)

---

1: **Initialize:** leader's policy parameter $\omega_0$, follower's policy parameter $\theta_0$, value function estimate $\hat{V}_{l,0}, \hat{J}_{l,0}, \hat{V}_{f,0}, \hat{J}_{f,0}$, mean field estimate $\hat{\mu}_0 \in \Delta_{\mathcal{S}}$, initial state $s_0, \bar{s}_0 \sim \rho$
2: **for** iteration $k = 0, 1, 2, ...$ **do**
3:   **Sample 1** for tracking the discounted occupancy measure:
     Get samples $s_k \sim d_\rho^{\pi_{\theta_k}, \phi_{\omega_k}, \hat{\mu}_k}$, $a_k \sim \phi_{\omega_k}(\cdot \mid s_k)$, $b_k \sim \phi_{\omega_k}(\cdot \mid s_k)$, $s_k' \sim \mathcal{P}^{\hat{\mu}_k}(\cdot \mid s_k, a_k, b_k)$
     and receive reward $r_f(s_k, a_k, b_k, \hat{\mu}_k), r_l(s_k, b_k, \hat{\mu}_k)$.
4:   **Sample 2** for tracking the stationary distribution (mean field):
     Get sample $\bar{s}_k \sim \nu^{\pi_{\theta_k}, \phi_{\omega_k}, \hat{\mu}_k}$.
5:   Leader's policy update:

$$\omega_{k+1} = \omega_k + \zeta_k \nabla_\omega \log \phi_{\omega_k}(b_k \mid s_k) \Big( r_l(s_k, b_k, \hat{\mu}_k) + \gamma \hat{V}_{l,k}(s_k') - \hat{V}_{l,k}(s_k) \Big) \qquad (37)$$

6:   Follower's policy (actor) update:

$$\theta_{k+1} = \theta_k + \alpha_k \nabla_\theta \log \pi_{\theta_k}(a_k \mid s_k) \Big( r_f(s_k, a_k, b_k, \hat{\mu}_k) + \tau E(\pi_{\theta_k}, s_k) + \gamma \hat{V}_{f,k}(s_k') - \hat{V}_{f,k}(s_k) \Big) \tag{38}$$

7:   Mean field update:

$$\hat{\mu}_{k+1} = \Pi_{\Delta_{\mathcal{S}}} \big( \hat{\mu}_k + \xi_k(e_{\bar{s}_k} - \hat{\mu}_k) \big) \qquad (39)$$

8:   Value function (critic) update:

$$\hat{V}_{l,k+1} = \Pi_{B_V} \left( \hat{V}_{l,k} + \beta_k e_{s_k} \big( r_l(s_k, b_k, \hat{\mu}_k) + \gamma \hat{V}_{l,k}(s_k') - \hat{V}_{l,k}(s_k) \big) \right) \qquad (40)$$

$$\hat{V}_{f,k+1} = \Pi_{B_V} \left( \hat{V}_{f,k} + \beta_k e_{s_k} \big( r_f(s_k, a_k, b_k, \hat{\mu}_k) + \tau E(\pi_{\theta_k}, s_k) + \gamma \hat{V}_{f,k}(s_k') - \hat{V}_{f,k}(s_k) \big) \right)$$

9: **end for**

---

# B  Proof of Main Results

We analyze a slightly simplified version of Algorithm 1, which we present in Algorithm 2. The only difference between them is that Algorithm 2 replaces continuously generated Markovian samples with i.i.d. samples from the stationary distribution. Note that stochastic approximation and RL algorithms have been extensively studied under Markovian samples [Zou et al., 2019, Wu et al., 2020], and it is well-known that Markovian sampling only incurs an additional logarithm factor in the convergence rates. This simplification allows us to focus the presentation on the innovations we develop for MFGs, without diverting attention to well-understood technical details on Markovian samples.

We state the theorem on the convergence of Algorithm 2 below, with the exact conditions that the step sizes need to satisfy.

**Theorem 2** *Consider the iterates of Algorithm 2 under the step sizes*

$$\zeta_k = \frac{c_\zeta}{(k+1)^{1/2}}, \quad \alpha_k = \frac{c_\alpha}{(k+1)^{1/2}}, \quad \xi_k = \frac{c_\xi}{(k+1)^{1/2}}, \quad \beta_k = \frac{c_\beta}{(k+1)^{1/2}},$$

*with the constants $c_\zeta, c_\alpha, c_\xi, c_\beta$ selected such that*

$$\zeta_l \le \xi_k \le \alpha_k \le \beta_k \le 1, \quad \xi_k \le \min\{\frac{1-\delta}{8L^2 L_H^2}, \frac{1}{1-\delta}\},$$

$$\alpha_k \le \min\{\frac{1-\gamma}{4L_{VV}(B_F + B_D + B_H)^2}, \frac{2}{B_D^2 + B_H^2}\}, \quad \beta_k \le \frac{1-\gamma}{L_G^2},$$

$$\frac{\zeta_k}{\xi_k} \le \min\Big\{\frac{1-\delta}{64L^2\eta_1\eta_2^2}, \frac{1-\delta}{24\eta_1 L_D^2 L_V^2}, \frac{1-\delta}{8\sqrt{6}LL_V L_D}, \frac{\eta_1(1-\delta)}{16L^2}, \frac{1-\delta}{16B_D^2 L}, \frac{B_H}{B_D L}, \frac{LL_H}{3L_V L_D}\Big\},$$

$$\frac{\zeta_k}{\alpha_k} \le \min\Big\{\frac{(1-\gamma)\tau^2\rho_{\min}^3 p_{\min}^2}{16|\mathcal{S}|L_V^2\eta_1\eta_2^2}, \frac{(1-\delta)(1-\gamma)\tau^2\rho_{\min}^3 p_{\min}^2}{192|\mathcal{S}|L^2 L_V^4 L_D^2 L_H^2}, \frac{(1-\gamma)\eta_1\tau^2\rho_{\min}^3 p_{\min}^2 L_D^2}{4|\mathcal{S}|L_V^2}\Big\},$$

$$\frac{\zeta_k}{\beta_k} \le \min\Big\{\frac{1-\gamma}{96L_V^2\eta_1\eta_2^2}, \frac{(1-\delta)(1-\gamma)}{384L^2 L_V^2 L_H^2}, \frac{1-\gamma}{32\eta_1 L_D^2}, \frac{1-\gamma}{2\sqrt{6}L_V^2}\Big\},$$

$$\frac{\xi_k}{\alpha_k} \le \min\Big\{\frac{(1-\gamma)(1-\delta)\tau\rho_{\min}^3 p_{\min}^2}{352|\mathcal{S}|L_H^2}, \frac{(1-\gamma)^2\tau\rho_{\min}^2 p_{\min}^2}{384|\mathcal{S}|L_V^2 L_H^2}, \frac{(1-\gamma)\tau^2\rho_{\min}^3 p_{\min}^2}{4|\mathcal{S}|L_V L_D L_H}, \frac{B_F(B_D + B_H)}{L_V(2B_H^2 + 3L_V)}\Big\},$$

$$\frac{\alpha_k}{\beta_k} \le \min\Big\{\frac{1-\gamma}{384L_V^2}, \frac{1-\gamma}{8\sqrt{6}L_V L_D}, \frac{2B_G}{\sqrt{6L_V^2 + L_{VV}}(B_F + B_D + B_H)}, \frac{1-\gamma}{4\sqrt{6}L_V L_D}\Big\}. \tag{41}$$

*Then, under Assumptions 1-4 and 6, we have for all $k > 0$,*

$$\min_{t<k}\mathbb{E}[\varepsilon_t^\phi] \le \frac{16}{c_\zeta(k+1)^{1/2}}\Big(J_l(\phi^\star, \mu^\star(\phi^\star)) - J(\phi_{\omega_0}, \mu^\star(\phi_{\omega_0})) + \varepsilon_0^\mu + \varepsilon_0^\pi + \varepsilon_{l,0}^V + \varepsilon_{f,0}^V\Big)$$

$$+ \frac{128c_\alpha\log(k+1)}{c_\zeta(k+1)^{1/2}}.$$

We note that the step sizes satisfying (41) always exist, and the conditions are met by making the step sizes sufficiently small. To find the appropriate step sizes, we first make $c_\beta$ small enough that $\beta_k$ satisfies all of its upper bounds, then $c_\xi$ small enough with respect to $\beta_k$, followed by $c_\alpha$ with respect to $\xi_k$ and $\beta_k$, and eventually $c_\zeta$. Note that $c_\zeta, c_\alpha, c_\xi, c_\beta$ are all chosen as polynomial functions of the structural parameters, and do not depend on $k$.

The proof of Theorem 1 relies on intermediate convergence bound on the iteration-wise convergence of the leader's policy parameter, mean field, follower's policy parameter, and value function estimates. We state the intermediate results in the propositions below and defer their proofs to Section C.

**Proposition 1 (Leader Convergence)** *Under Assumptions 1-4 and 6 and step sizes satisfying* (41), *we have for all $k \ge 0$,*

$$\mathbb{E}[J_l(\phi_{\omega_k}, \mu^\star(\phi_{\omega_k})) - J_l(\phi_{\omega_{k+1}}, \mu^\star(\phi_{\omega_{k+1}}))]$$

$$\le -\frac{\zeta_k}{2\eta_1}\mathbb{E}[\varepsilon_k^\phi] + 2\eta_1 L_D^2 L_V^2\zeta_k\mathbb{E}[\varepsilon_k^\mu] + 2\eta_1 L_D^2\zeta_k\mathbb{E}[\varepsilon_{l,k}^V] + \frac{L_V B_D^2\zeta_k^2}{2}.$$

**Proposition 2 (Mean Field Convergence)** *Under Assumptions 1-4 and 6 and step sizes satisfying* (41), *we have for all $k \ge 0$,*

$$\mathbb{E}[\varepsilon_{k+1}^\mu] \le (1 - \frac{1-\delta}{4}\xi_k)\mathbb{E}[\varepsilon_k^\mu] + \frac{11L_H^2\xi_k}{(1-\delta)\tau\rho_{\min}}\mathbb{E}[\varepsilon_k^\pi] + \frac{8L^2\eta_2^2\zeta_k^2}{(1-\delta)\xi_k}\mathbb{E}[\varepsilon_k^\phi]$$

$$+ \frac{32L^2 L_D^2\zeta_k^2}{(1-\delta)\xi_k}\mathbb{E}[\varepsilon_{l,k}^V] + 16B_H^2\xi_k^2.$$

**Proposition 3 (Follower Convergence)** *Under Assumptions 1-4 and 6 and step sizes satisfying* (41), *we have for all $k \ge 0$,*

$$\mathbb{E}[\varepsilon_{k+1}^\pi] \le \mathbb{E}[\varepsilon_k^\pi] - \frac{\alpha_k}{16}\mathbb{E}[\|\nabla_\theta J_f(\pi_{\theta_k}, \phi_{\omega_k}, \hat{\mu}_k)\|^2] + \frac{2|\mathcal{S}|L_V^2\eta_2^2\zeta_k^2}{(1-\gamma)\tau^2\rho_{\min}^3 p_{\min}^2\alpha_k}\mathbb{E}[\varepsilon_k^\phi]$$

$$+ \frac{16|\mathcal{S}|L^2 L_V^4 L_D^2 L_H^2\xi_k^2}{(1-\gamma)\tau^2\rho_{\min}^3 p_{\min}^2\alpha_k}\mathbb{E}[\varepsilon_k^\mu] + \frac{8|\mathcal{S}|L_V^2\zeta_k^2}{(1-\gamma)\tau^2\rho_{\min}^3 p_{\min}^2\alpha_k}\mathbb{E}[\varepsilon_{l,k}^V]$$

$$+ 2L_V^2\alpha_k\mathbb{E}[\varepsilon_{f,k}^V] + 3B_F(B_D + B_H)\alpha_k\xi_k.$$

**Proposition 4 (Value Function Convergence)** *Under Assumptions 1-4 and 6 and step sizes satisfying (41), we have for all $k \geq 0$,*

$$\mathbb{E}[\varepsilon_{f,k+1}^V] \leq (1 - \frac{(1-\gamma)\beta_k}{4})\mathbb{E}[\varepsilon_{f,k}^V] + \frac{6L_V^2\alpha_k^2}{(1-\gamma)\beta_k}\mathbb{E}[\|\nabla_\theta J_f(\pi_{\theta_k}, \phi_{\omega_k}, \hat{\mu}_k)\|^2] + \frac{6L_V^2 L_H^2 \xi_k^2}{(1-\gamma)\beta_k}\mathbb{E}[\varepsilon_k^\pi]$$

$$+ \frac{6L_V^2\eta_2^2\zeta_k^2}{(1-\gamma)\beta_k}\mathbb{E}[\varepsilon_k^\phi] + \frac{16L^2 L_V^2 L_H^2 \xi_k^2}{(1-\gamma)\beta_k}\mathbb{E}[\varepsilon_k^\mu] + \frac{24L_V^2 L_D^2 \alpha_k^2}{(1-\gamma)\beta_k}\mathbb{E}[\varepsilon_{l,k}^V] + 12B_G^2\beta_k^2,$$

$$\mathbb{E}[\varepsilon_{l,k+1}^V] \leq (1 - \frac{(1-\gamma)\beta_k}{4})\mathbb{E}[\varepsilon_{l,k}^V] + \frac{6L_V^2\alpha_k^2}{(1-\gamma)\beta_k}\mathbb{E}[\|\nabla_\theta J_f(\pi_{\theta_k}, \phi_{\omega_k}, \hat{\mu}_k)\|^2] + \frac{6L_V^2 L_H^2 \xi_k^2}{(1-\gamma)\beta_k}\mathbb{E}[\varepsilon_k^\pi]$$

$$+ \frac{6L_V^2\eta_2^2\zeta_k^2}{(1-\gamma)\beta_k}\mathbb{E}[\varepsilon_k^\phi] + \frac{16L^2 L_V^2 L_H^2 \xi_k^2}{(1-\gamma)\beta_k}\mathbb{E}[\varepsilon_k^\mu] + \frac{6L_V^4\zeta_k^2}{(1-\gamma)\beta_k}\mathbb{E}[\varepsilon_{f,k}^V] + 12B_G^2\beta_k^2.$$

## B.1 Proof of Theorem 2

Collecting the results from Propositions 1-4, we have

$$\mathbb{E}\left[\left(J_l(\phi_{\omega_k}, \mu^\star(\phi_{\omega_k})) - J_l(\phi_{\omega_{k+1}}, \mu^\star(\phi_{\omega_{k+1}}))\right) + \varepsilon_{k+1}^\mu + \varepsilon_{k+1}^\pi + \varepsilon_{f,k+1}^V + \varepsilon_{l,k+1}^V\right]$$

$$\leq -\frac{\zeta_k}{2\eta_1}\mathbb{E}[\varepsilon_k^\phi] + 2\eta_1 L_D^2 L_V^2 \zeta_k \mathbb{E}[\varepsilon_k^\mu] + 2\eta_1 L_D^2 \zeta_k \mathbb{E}[\varepsilon_{l,k}^V] + \frac{L_V B_D^2 \zeta_k^2}{2}$$

$$+ (1 - \frac{1-\delta}{4}\xi_k)\mathbb{E}[\varepsilon_k^\mu] + \frac{11L_H^2\xi_k}{(1-\delta)\tau\rho_{\min}}\mathbb{E}[\varepsilon_k^\pi] + \frac{8L^2\eta_2^2\zeta_k^2}{(1-\delta)\xi_k}\mathbb{E}[\varepsilon_k^\phi] + \frac{32L^2 L_D^2 \zeta_k^2}{(1-\delta)\xi_k}\mathbb{E}[\varepsilon_{l,k}^V] + 16B_H^2\xi_k$$

$$+ \mathbb{E}[\varepsilon_k^\pi] - \frac{\alpha_k}{16}\mathbb{E}[\|\nabla_\theta J_f(\pi_{\theta_k}, \phi_{\omega_k}, \hat{\mu}_k)\|^2] + \frac{2|\mathcal{S}|L_V^2\eta_2^2\zeta_k^2}{(1-\gamma)\tau^2\rho_{\min}^3 p_{\min}^2 \alpha_k}\mathbb{E}[\varepsilon_k^\phi] + \frac{16|\mathcal{S}|L^2 L_V^4 L_D^2 L_H^2 \xi_k^2}{(1-\gamma)\tau^2\rho_{\min}^3 p_{\min}^2 \alpha_k}\mathbb{E}[\varepsilon_k^\mu]$$

$$+ \frac{8|\mathcal{S}|L_V^2\zeta_k^2}{(1-\gamma)\tau^2\rho_{\min}^3 p_{\min}^2 \alpha_k}\mathbb{E}[\varepsilon_{l,k}^V] + 2L_V^2\alpha_k\mathbb{E}[\varepsilon_{f,k}^V] + 3B_F(B_D + B_H)\alpha_k\xi_k$$

$$+ (1 - \frac{(1-\gamma)\beta_k}{4})\mathbb{E}[\varepsilon_{f,k}^V] + \frac{6L_V^2\alpha_k^2}{(1-\gamma)\beta_k}\mathbb{E}[\|\nabla_\theta J_f(\pi_{\theta_k}, \phi_{\omega_k}, \hat{\mu}_k)\|^2] + \frac{6L_V^2 L_H^2 \xi_k^2}{(1-\gamma)\beta_k}\mathbb{E}[\varepsilon_k^\pi]$$

$$+ \frac{6L_V^2\eta_2^2\zeta_k^2}{(1-\gamma)\beta_k}\mathbb{E}[\varepsilon_k^\phi] + \frac{16L^2 L_V^2 L_H^2 \xi_k^2}{(1-\gamma)\beta_k}\mathbb{E}[\varepsilon_k^\mu] + \frac{24L_V^2 L_D^2 \alpha_k^2}{(1-\gamma)\beta_k}\mathbb{E}[\varepsilon_{l,k}^V] + 12B_G^2\beta_k^2$$

$$+ (1 - \frac{(1-\gamma)\beta_k}{4})\mathbb{E}[\varepsilon_{l,k}^V] + \frac{6L_V^2\alpha_k^2}{(1-\gamma)\beta_k}\mathbb{E}[\|\nabla_\theta J_f(\pi_{\theta_k}, \phi_{\omega_k}, \hat{\mu}_k)\|^2] + \frac{6L_V^2 L_H^2 \xi_k^2}{(1-\gamma)\beta_k}\mathbb{E}[\varepsilon_k^\pi]$$

$$+ \frac{6L_V^2\eta_2^2\zeta_k^2}{(1-\gamma)\beta_k}\mathbb{E}[\varepsilon_k^\phi] + \frac{16L^2 L_V^2 L_H^2 \xi_k^2}{(1-\gamma)\beta_k}\mathbb{E}[\varepsilon_k^\mu] + \frac{6L_V^4\zeta_k^2}{(1-\gamma)\beta_k}\mathbb{E}[\varepsilon_{f,k}^V] + 12B_G^2\beta_k^2$$

$$\leq \underbrace{\left(-\frac{\zeta_k}{2\eta_1} + \frac{8L^2\eta_2^2\zeta_k^2}{(1-\delta)\xi_k} + \frac{2|\mathcal{S}|L_V^2\eta_2^2\zeta_k^2}{(1-\gamma)\tau^2\rho_{\min}^3 p_{\min}^2 \alpha_k} + \frac{12L_V^2\eta_2^2\zeta_k^2}{(1-\gamma)\beta_k}\right)}_{T_1}\mathbb{E}[\varepsilon_k^\phi]$$

$$+ \mathbb{E}[\varepsilon_k^\mu] + \underbrace{\left(-\frac{1-\delta}{4}\xi_k + 2\eta_1 L_D^2 L_V^2 \zeta_k + \frac{16|\mathcal{S}|L^2 L_V^4 L_D^2 L_H^2 \xi_k^2}{(1-\gamma)\tau^2\rho_{\min}^3 p_{\min}^2 \alpha_k} + \frac{32L^2 L_V^2 L_H^2 \xi_k^2}{(1-\gamma)\beta_k}\right)}_{T_2}\mathbb{E}[\varepsilon_k^\mu]$$

$$+ \mathbb{E}[\varepsilon_k^\pi] + \underbrace{\left(-\frac{\alpha_k}{16} + \frac{12L_V^2\alpha_k^2}{(1-\gamma)\beta_k}\right)\mathbb{E}[\|\nabla_\theta J_f(\pi_{\theta_k}, \phi_{\omega_k}, \hat{\mu}_k)\|^2] + \left(\frac{11L_H^2\xi_k}{(1-\delta)\tau\rho_{\min}} + \frac{12L_V^2 L_H^2 \xi_k^2}{(1-\gamma)\beta_k}\right)\mathbb{E}[\varepsilon_k^\pi]}_{T_3}$$

$$+ \mathbb{E}[\varepsilon_{l,k}^V] + \underbrace{\left(-\frac{1-\gamma}{4}\beta_k + 2\eta_1 L_D^2 \zeta_k + \frac{32L^2 L_D^2 \zeta_k^2}{(1-\delta)\xi_k} + \frac{8|\mathcal{S}|L_V^2\zeta_k^2}{(1-\gamma)\tau^2\rho_{\min}^3 p_{\min}^2 \alpha_k} + \frac{24L_V^2 L_D^2 \alpha_k^2}{(1-\gamma)\beta_k}\right)}_{T_4}\mathbb{E}[\varepsilon_{l,k}^V]$$

$$+ \mathbb{E}[\varepsilon_{f,k}^V] + \underbrace{\left(-\frac{1-\gamma}{4}\beta_k + 2L_V^2\alpha_k + \frac{6L_V^4\zeta_k^2}{(1-\gamma)\beta_k}\right)}_{T_5}\mathbb{E}[\varepsilon_{f,k}^V]$$

$$+ \underbrace{\frac{B_D^2 L_V \zeta_k^2}{2} + 16 B_H^2 \xi_k^2 + 3 B_F (B_D + B_H) \alpha_k \xi_k + 24 B_G^2 \beta_k^2}_{T_6}. \tag{42}$$

We bound each term of (42) individually. First, we can ensure $T_1 \leq -\frac{\zeta_k}{8\eta_1}$ by choosing the step sizes such that every positive term is no larger than $\frac{\zeta_k}{8\eta_1}$. The required step size conditions are

$$\frac{\zeta_k}{\xi_k} \leq \frac{1-\delta}{64 L^2 \eta_1 \eta_2^2}, \quad \frac{\zeta_k}{\alpha_k} \leq \frac{(1-\gamma)\tau^2 \rho_{\min}^3 p_{\min}^2}{16|\mathcal{S}| L_V^2 \eta_1 \eta_2^2}, \quad \frac{\zeta_k}{\beta_k} \leq \frac{1-\gamma}{96 L_V^2 \eta_1 \eta_2^2}.$$

For $T_2$, we choose the step sizes such that each positive term is no larger than $\frac{1-\delta}{12}$, making $T_2$ non-positive. The step size conditions are

$$\frac{\zeta_k}{\xi_k} \leq \frac{1-\delta}{24\eta_1 L_D^2 L_V^2}, \quad \frac{\zeta_k}{\alpha_k} \leq \frac{(1-\delta)(1-\gamma)\tau^2 \rho_{\min}^3 p_{\min}^2}{192|\mathcal{S}| L^2 L_V^4 L_D^2 L_H^2}, \quad \frac{\zeta_k}{\beta_k} \leq \frac{(1-\delta)(1-\gamma)}{384 L^2 L_V^2 L_H^2}.$$

To bound $T_3$, note that under entropy regularization, $J_f$ satisfies a non-uniform PL condition with respect to the follower's policy [Mei et al., 2020]. Specifically, for any $\theta, \phi, \mu$,

$$\|\nabla_\theta J_f(\pi_\theta, \phi, \mu)\|^2 \geq \frac{2(1-\gamma)\tau \rho_{\min}^2 p_{\min}^2}{|\mathcal{S}|} \Big( J_f(\pi^\star(\phi, \mu), \phi, \mu) - J_f(\pi_\theta, \phi, \mu) \Big). \tag{43}$$

The inequality in this specific form is adapted from Zeng et al. [2022][Lemma 4].

Under the step size conditions $\frac{\alpha_k}{\beta_k} \leq \frac{1-\gamma}{384 L_V^2}$, $\xi_k \leq \beta_k$, and $\frac{\xi_k}{\alpha_k} \leq \min\{\frac{(1-\gamma)(1-\delta)\tau \rho_{\min}^3 p_{\min}^2}{352|\mathcal{S}| L_H^2}, \frac{(1-\gamma)^2 \tau \rho_{\min}^2 p_{\min}^2}{384|\mathcal{S}| L_V^2 L_H^2}\}$, (43) implies

$$T_3 \leq -\frac{\alpha_k}{32} \mathbb{E}[\|\nabla_\theta J_f(\pi_{\theta_k}, \phi_{\omega_k}, \hat{\mu}_k)\|^2] + \Big( \frac{11 L_H^2 \xi_k}{(1-\delta)\tau \rho_{\min}} + \frac{12 L_V^2 L_H^2 \xi_k^2}{(1-\gamma)\beta_k} \Big) \mathbb{E}[\varepsilon_k^\pi]$$

$$\leq \Big( -\frac{(1-\gamma)\tau \rho_{\min}^2 p_{\min}^2}{16|\mathcal{S}|} \alpha_k + \frac{11 L_H^2 \xi_k}{(1-\delta)\tau \rho_{\min}} + \frac{12 L_V^2 L_H^2 \xi_k^2}{(1-\gamma)\beta_k} \Big) \mathbb{E}[\varepsilon_k^\pi]$$

$$\leq 0.$$

For $T_4$, each positive term is no larger than $\frac{1-\gamma}{16}$ under the step size conditions

$$\frac{\zeta_k}{\beta_k} \leq \frac{1-\gamma}{32\eta_1 L_D^2}, \quad \frac{\zeta_k}{\xi_k} \leq \frac{\eta_1(1-\delta)}{16 L^2}, \quad \frac{\zeta_k}{\alpha_k} \leq \frac{(1-\gamma)\eta_1 \tau^2 \rho_{\min}^3 p_{\min}^2 L_D^2}{4|\mathcal{S}| L_V^2}, \quad \frac{\alpha_k}{\beta_k} \leq \frac{1-\gamma}{8\sqrt{6} L_V L_D}.$$

This ensures $T_4 \leq 0$.

Similarly, we have $T_5 \leq 0$ under the conditions

$$\frac{\alpha_k}{\beta_k} \leq \frac{1-\gamma}{16 L_V^2}, \quad \frac{\zeta_k}{\beta_k} \leq \frac{1-\gamma}{2\sqrt{6} L_V^2}.$$

Finally, we choose the step sizes such that each term in $T_6$ is no larger than $\alpha_k^2$

$$\frac{\zeta_k}{\alpha_k} \leq B_D \sqrt{\frac{L_V}{2}}, \quad \frac{\xi_k}{\alpha_k} \leq \min\{\frac{1}{4 B_H}, \frac{1}{B_F(B_D + B_H)}\}, \quad \frac{\beta_k}{\alpha_k} \leq \frac{1}{2\sqrt{6} B_G}.$$

Plugging the bounds on $T_1$-$T_6$ into (42), we have

$$\mathbb{E}\Big[\Big(J_l(\phi_{\omega_k}, \mu^\star(\phi_{\omega_k})) - J_l(\phi_{\omega_{k+1}}, \mu^\star(\phi_{\omega_{k+1}}))\Big) + \varepsilon_{k+1}^\mu + \varepsilon_{k+1}^\pi + \varepsilon_{f,k+1}^V + \varepsilon_{l,k+1}^V\Big]$$

$$\leq -\frac{\zeta_k}{8\eta_1} \mathbb{E}[\varepsilon_k^\phi] + \mathbb{E}[\varepsilon_k^\mu + \varepsilon_k^\pi + \varepsilon_{l,k}^V + \varepsilon_{f,k}^V] + 4\alpha_k^2. \tag{44}$$

Re-arranging the terms and plugging in the step size, we have

$$\frac{c_\zeta}{8(k+1)^{1/2}} \mathbb{E}[\varepsilon_k^\phi] \leq \mathbb{E}[-J_l(\phi_{\omega_k}, \mu^\star(\phi_{\omega_k})) + \varepsilon_k^\mu + \varepsilon_k^\pi + \varepsilon_{l,k}^V + \varepsilon_{f,k}^V]$$

$$- \mathbb{E}[-J_l(\phi_{\omega_{k+1}}, \mu^\star(\phi_{\omega_{k+1}})) + \varepsilon_{k+1}^\mu + \varepsilon_{k+1}^\pi + \varepsilon_{f,k+1}^V + \varepsilon_{l,k+1}^V] + \frac{4c_\alpha}{(k+1)}.$$

Recall that $\phi^\star$ is the optimal solution to the Stackelberg objective in (6). The inequality above implies

$$\sum_{t=0}^{k-1} \frac{c_\zeta}{8(t+1)^{1/2}} \mathbb{E}[\varepsilon_t^\phi]$$

$$\leq \sum_{t=0}^{k-1} \frac{c_\zeta}{8(t+1)^{1/2}} \mathbb{E}[\varepsilon_t^\phi] + \mathbb{E}[\varepsilon_{k+1}^\mu + \varepsilon_{k+1}^\pi + \varepsilon_{f,k+1}^V + \varepsilon_{l,k+1}^V]$$

$$\leq J_l(\phi^\star, \mu^\star(\phi^\star)) - J(\phi_{\omega_0}, \mu^\star(\phi_{\omega_0})) + \varepsilon_0^\mu + \varepsilon_0^\pi + \varepsilon_{l,0}^V + \varepsilon_{f,0}^V + \sum_{t=0}^{k-1} \frac{4c_\alpha}{(t+1)}. \tag{45}$$

The following relationships are standard results: for any $k \geq 0$, we have

$$\sum_{t=0}^{k-1} \frac{1}{(t+1)^{1/2}} \geq \frac{(k+1)^{1/2}}{2}, \quad \sum_{t=0}^{k-1} \frac{1}{t+1} \leq 2\log(k+1). \tag{46}$$

Combining (45) and (46),

$$\min_{t<k} \mathbb{E}[\varepsilon_t^\phi] \leq \frac{\sum_{t=0}^{k-1} \frac{c_\zeta}{8(t+1)^{1/2}} \mathbb{E}[\varepsilon_t^\phi]}{\sum_{t'=0}^{k-1} \frac{c_\zeta}{8(t'+1)^{1/2}}}$$

$$\leq \frac{16}{c_\zeta(k+1)^{1/2}} \left( J_l(\phi^\star, \mu^\star(\phi^\star)) - J(\phi_{\omega_0}, \mu^\star(\phi_{\omega_0})) + \varepsilon_0^\mu + \varepsilon_0^\pi + \varepsilon_{l,0}^V + \varepsilon_{f,0}^V \right)$$

$$+ \frac{128c_\alpha \log(k+1)}{c_\zeta(k+1)^{1/2}}.$$

∎

## C  Proof of Propositions

### C.1  Proof of Proposition 1

By the $L_V$-smoothness of the function $J_l$

$$J_l(\phi_{\omega_k}, \mu^\star(\phi_{\omega_k})) - J_l(\phi_{\omega_{k+1}}, \mu^\star(\phi_{\omega_{k+1}}))$$

$$\leq -\langle \nabla_\omega J_l(\phi_{\omega_k}, \mu^\star(\phi_{\omega_k})), \omega_{k+1} - \omega_k \rangle + \frac{L_V}{2} \|\omega_{k+1} - \omega_k\|^2$$

$$= -\zeta_k \langle \nabla_\omega J_l(\phi_{\omega_k}, \mu^\star(\phi_{\omega_k})), D(\omega_k, \hat{\mu}_k, \hat{V}_{l,k}, s_k, b_k, s_k') \rangle + \frac{L_V \zeta_k^2}{2} \|D(\omega_k, \hat{\mu}_k, \hat{V}_{l,k}, s_k, b_k, s_k')\|^2$$

$$= -\zeta_k \langle \nabla_\omega J_l(\phi_{\omega_k}, \mu^\star(\phi_{\omega_k})), D(\omega_k, \hat{\mu}_k, \hat{V}_{l,k}, s_k, b_k, s_k') - \bar{D}(\omega_k, \hat{\mu}_k, \hat{V}_{l,k}) \rangle$$

$$- \zeta_k \langle \nabla_\omega J_l(\phi_{\omega_k}, \mu^\star(\phi_{\omega_k})), \bar{D}(\omega_k, \hat{\mu}_k, \hat{V}_{l,k}) \rangle + \frac{L_V \zeta_k^2}{2} \|D(\omega_k, \hat{\mu}_k, \hat{V}_{l,k}, s_k, b_k, s_k')\|^2. \tag{47}$$

Taking the expectation and plugging in the bound on $D$ from Lemma 3, we have

$$\mathbb{E}[J_l(\phi_{\omega_k}, \mu^\star(\phi_{\omega_k})) - J_l(\phi_{\omega_{k+1}}, \mu^\star(\phi_{\omega_{k+1}}))]$$

$$\leq -\zeta_k \mathbb{E}[\langle \nabla_\omega J_l(\phi_{\omega_k}, \mu^\star(\phi_{\omega_k})), \bar{D}(\omega_k, \hat{\mu}_k, \hat{V}_{l,k}) \rangle] + \frac{L_V B_D^2 \zeta_k^2}{2}$$

$$\leq -\frac{\zeta_k}{\eta_1} \mathbb{E}[\|\nabla_\omega J_l(\phi_{\omega_k}, \mu^\star(\phi_{\omega_k}))\|^2] + \frac{L_V B_D^2 \zeta_k^2}{2}$$

$$+ \zeta_k \mathbb{E}[\langle \nabla_\omega J_l(\phi_{\omega_k}, \mu^\star(\phi_{\omega_k})), \bar{D} \left( \omega_k, \mu^\star(\phi_{\omega_k}), V_l^{\phi_{\omega_k}, \mu^\star(\phi_{\omega_k})} \right) - \bar{D}(\omega_k, \hat{\mu}_k, \hat{V}_{l,k}) \rangle], \tag{48}$$

where the first inequality follows from

$$\mathbb{E}[\langle \nabla_\omega J_l(\phi_{\omega_k}, \mu^\star(\phi_{\omega_k})), D(\omega_k, \hat{\mu}_k, \hat{V}_{l,k}, s_k, b_k, s_k') - \bar{D}(\omega_k, \hat{\mu}_k, \hat{V}_{l,k})\rangle]$$
$$= \mathbb{E}[\langle \nabla_\omega J_l(\phi_{\omega_k}, \mu^\star(\phi_{\omega_k})), \mathbb{E}[D(\omega_k, \hat{\mu}_k, \hat{V}_{l,k}, s_k, b_k, s_k') - \bar{D}(\omega_k, \hat{\mu}_k, \hat{V}_{l,k})] \mid \mathcal{F}_{k-1}\rangle] = 0.$$

and the second inequality follows from Assumption 6 and the relationship

$$\nabla_\omega J(\phi_\omega, \mu)\mid_{\mu=\mu^\star(\phi_\omega)} = \bar{D}(\omega, \mu^\star(\phi_\omega), V_l^{\phi_\omega, \mu^\star(\phi_\omega)}).$$

To bound the last term on the right hand side of (48), we apply Young's inequality

$$\zeta_k \langle \nabla_\omega J_l(\phi_{\omega_k}, \mu^\star(\phi_{\omega_k})), \bar{D}\left(\omega_k, \mu^\star(\phi_{\omega_k}), V_l^{\phi_{\omega_k}, \mu^\star(\phi_{\omega_k})}\right) - \bar{D}(\omega_k, \hat{\mu}_k, \hat{V}_{l,k})\rangle$$

$$\leq \frac{\zeta_k}{2\eta_1} \|\nabla_\omega J(\phi_{\omega_k}, \mu^\star(\phi_{\omega_k}))\|^2 + \frac{\eta_1 \zeta_k}{2} \left\|\bar{D}\left(\omega_k, \mu^\star(\phi_{\omega_k}), V_l^{\phi_{\omega_k}, \mu^\star(\phi_{\omega_k})}\right) - \bar{D}(\omega_k, \hat{\mu}_k, \hat{V}_{l,k})\right\|^2$$

$$\leq \frac{\zeta_k}{2\eta_1} \|\nabla_\omega J(\phi_{\omega_k}, \mu^\star(\phi_{\omega_k}))\|^2 + \eta_1 L_D^2 \zeta_k \|\hat{V}_{l,k} - V_l^{\phi_{\omega_k}, \mu^\star(\phi_{\omega_k})}\|^2 + \eta_1 L_D^2 \zeta_k \|\hat{\mu}_k - \mu^\star(\phi_{\omega_k})\|^2$$

$$\leq \frac{\zeta_k}{2\eta_1} \varepsilon_k^\phi + 2\eta_1 L_D^2 \zeta_k \|\hat{V}_{l,k} - V_l^{\phi_{\omega_k}, \hat{\mu}_k}\|^2 + \eta_1 L_D^2 (L_V^2 + 1) \zeta_k \|\hat{\mu}_k - \mu^\star(\phi_{\omega_k})\|^2$$

$$\leq \frac{\zeta_k}{2\eta_1} \varepsilon_k^\phi + 2\eta_1 L_D^2 L_V^2 \zeta_k \varepsilon_k^\mu + 2\eta_1 L_D^2 \zeta_k \varepsilon_{l,k}^V, \tag{49}$$

where the second and third inequalities follow from the Lipschitz continuity established in Lemma 4.

Substituting (49) into (48),

$$\mathbb{E}[J_l(\phi_{\omega_k}, \mu^\star(\phi_{\omega_k})) - J_l(\phi_{\omega_{k+1}}, \mu^\star(\phi_{\omega_{k+1}}))]$$

$$\leq -\frac{\zeta_k}{\eta_1} \mathbb{E}[\varepsilon_k^\phi] + \frac{L_V B_D^2 \zeta_k^2}{2} + \zeta_k \mathbb{E}[\langle \nabla_\omega J_l(\phi_{\omega_k}, \mu^\star(\phi_{\omega_k})), \bar{D}\left(\omega_k, \mu^\star(\phi_{\omega_k}), V_l^{\phi_{\omega_k}, \mu^\star(\phi_{\omega_k})}\right) - \bar{D}(\omega_k, \hat{\mu}_k, \hat{V}_{l,k})\rangle]$$

$$\leq -\frac{\zeta_k}{2\eta_1} \mathbb{E}[\varepsilon_k^\phi] + 2\eta_1 L_D^2 L_V^2 \zeta_k \mathbb{E}[\varepsilon_k^\mu] + 2\eta_1 L_D^2 \zeta_k \mathbb{E}[\varepsilon_{l,k}^V] + \frac{L_V B_D^2 \zeta_k^2}{2}.$$

∎

## C.2 Proof of Proposition 2

The analysis employs the following lemma, which bounds an intermediate cross term. We provide the proof of the lemma in Section D.5.

**Lemma 5** *Under the assumptions and step sizes of Proposition 2, we have for all $k \geq 0$,*

$$\mathbb{E}[\langle \hat{\mu}_k - \mu^\star(\phi_{\omega_k}) + \xi_k \bar{H}(\pi^\star(\phi_{\omega_k}, \hat{\mu}_k), \omega_k, \hat{\mu}_k), \mu^\star(\phi_{\omega_{k+1}}) - \mu^\star(\phi_{\omega_k})\rangle]$$

$$\leq \frac{(1-\delta)\xi_k}{8} \mathbb{E}[\|\hat{\mu}_k - \mu^\star(\phi_{\omega_k}) + \xi_k \bar{H}(\pi^\star(\phi_{\omega_k}\|^2] + \frac{48 L^2 L_V^2 L_D^2 \zeta_k^2}{(1-\delta)\xi_k} \mathbb{E}[\varepsilon_k^\mu]$$

$$+ \frac{8 L^2 \eta_2^2 \zeta_k^2}{(1-\delta)\xi_k} \mathbb{E}[\varepsilon_k^\phi] + \frac{32 L^2 L_D^2 \zeta_k^2}{(1-\delta)\xi_k} \mathbb{E}[\varepsilon_{l,k}^V] + B_D^2 L \zeta_k^2.$$

We define the shorthand notation $\Delta h_k \triangleq H(\hat{\mu}_k, \bar{s}_k) - \bar{H}(\pi_{\theta_k}, \omega_k, \hat{\mu}_k)$. Note that $\mathbb{E}[\Delta h_k] = 0$ and $\|\Delta h_k\|^2 \leq 4 B_H^2$.

By the definition of $\varepsilon_k^\mu$,

$$\varepsilon_{k+1}^\mu = \|\hat{\mu}_{k+1} - \mu^\star(\phi_{\omega_{k+1}})\|^2$$

$$= \|\Pi_{\Delta_S}(\hat{\mu}_k + \xi_k H(\hat{\mu}_k, \bar{s}_k)) - \mu^\star(\phi_{\omega_{k+1}})\|^2$$

$$\leq \|\hat{\mu}_k + \xi_k H(\hat{\mu}_k, \bar{s}_k) - \mu^\star(\phi_{\omega_{k+1}})\|^2$$

$$= \left\|\hat{\mu}_k - \mu^\star(\phi_{\omega_k}) + \xi_k \Delta h_k + \xi_k \bar{H}(\pi^\star(\phi_{\omega_k}, \hat{\mu}_k), \omega_k, \hat{\mu}_k)\right.$$

$$+ \xi_k \left( \bar{H}(\pi_{\theta_k}, \omega_k, \hat{\mu}_k) - \bar{H}(\pi^\star(\phi_{\omega_k}, \hat{\mu}_k), \omega_k, \hat{\mu}_k) \right) - \left( \mu^\star(\phi_{\omega_{k+1}}) - \mu^\star(\phi_{\omega_k}) \right) \Big\|^2$$

$$\leq \|\hat{\mu}_k - \mu^\star(\phi_{\omega_k}) + \xi_k \bar{H}(\pi^\star(\phi_{\omega_k}, \hat{\mu}_k), \omega_k, \hat{\mu}_k)\|^2 + 3\xi_k^2 \|\Delta h_k\|^2 + 3\|\mu^\star(\phi_{\omega_{k+1}}) - \mu^\star(\phi_{\omega_k})\|^2$$
$$+ 3\xi_k^2 \|\bar{H}(\pi_{\theta_k}, \omega_k, \hat{\mu}_k) - \bar{H}(\pi^\star(\phi_{\omega_k}, \hat{\mu}_k), \omega_k, \hat{\mu}_k)\|^2$$
$$+ 2\xi_k \langle \hat{\mu}_k - \mu^\star(\phi_{\omega_k}) + \xi_k \bar{H}(\pi^\star(\phi_{\omega_k}, \hat{\mu}_k), \omega_k, \hat{\mu}_k), \Delta h_k \rangle$$
$$+ 2\langle \hat{\mu}_k - \mu^\star(\phi_{\omega_k}) + \xi_k \bar{H}(\pi^\star(\phi_{\omega_k}, \hat{\mu}_k), \omega_k, \hat{\mu}_k), \mu^\star(\phi_{\omega_{k+1}}) - \mu^\star(\phi_{\omega_k}) \rangle$$
$$+ 2\xi_k \langle \hat{\mu}_k - \mu^\star(\phi_{\omega_k}) + \xi_k \bar{H}(\pi^\star(\phi_{\omega_k}, \hat{\mu}_k), \omega_k, \hat{\mu}_k), \bar{H}(\pi^\star(\phi_{\omega_k}, \hat{\mu}_k), \omega_k, \hat{\mu}_k) - \bar{H}(\pi_{\theta_k}, \omega_k, \hat{\mu}_k) \rangle,$$
$$\tag{50}$$

where the first inequality is due to the fact that projection to a convex set is a non-expansive operator.

As $\bar{H}(\pi^\star(\phi), \phi, \mu^\star(\phi)) = 0$ for any $\phi$, we have for the first term of (50),

$$\|\hat{\mu}_k - \mu^\star(\phi_{\omega_k}) + \xi_k \bar{H}(\pi^\star(\phi_{\omega_k}, \hat{\mu}_k), \omega_k, \hat{\mu}_k)\|^2$$
$$= \|\hat{\mu}_k - \mu^\star(\phi_{\omega_k}) + \xi_k \bar{H}(\pi^\star(\phi_{\omega_k}, \hat{\mu}_k), \omega_k, \hat{\mu}_k) - \xi_k \bar{H}(\pi^\star(\phi_{\omega_k}), \omega_k, \mu^\star(\phi_{\omega_k}))\|^2$$
$$= \|\hat{\mu}_k - \mu^\star(\phi_{\omega_k})\|^2 + \xi_k^2 \|\bar{H}(\pi^\star(\phi_{\omega_k}, \hat{\mu}_k), \omega_k, \hat{\mu}_k) - \bar{H}(\pi^\star(\phi_{\omega_k}), \omega_k, \mu^\star(\phi_{\omega_k}))\|^2$$
$$+ \xi_k \langle \hat{\mu}_k - \mu^\star(\phi_{\omega_k}), \nu^{\pi^\star(\phi_{\omega_k}, \hat{\mu}_k), \omega_k, \hat{\mu}_k} - \hat{\mu}_k - \nu^{\pi^\star(\phi_{\omega_k}), \omega_k, \mu^\star(\phi_{\omega_k})} + \mu^\star(\phi_{\omega_k}) \rangle$$
$$\leq (1 - \xi_k)\|\hat{\mu}_k - \mu^\star(\phi_{\omega_k})\|^2 + 2L_H^2 \xi_k^2 \|\hat{\mu}_k - \mu^\star(\phi_{\omega_k})\|^2 + 2L_H^2 \xi_k^2 \|\pi^\star(\phi_{\omega_k}, \hat{\mu}_k) - \pi^\star(\phi_{\omega_k})\|^2$$
$$+ \xi_k \|\hat{\mu}_k - \mu^\star(\phi_{\omega_k})\| \|\nu^{\pi^\star(\phi_{\omega_k}, \hat{\mu}_k), \omega_k, \hat{\mu}_k} - \nu^{\pi^\star(\phi_{\omega_k}), \omega_k, \mu^\star(\phi_{\omega_k})}\|$$
$$\leq \left( 1 - (1-\delta)\xi_k + 4L^2 L_H^2 \xi_k^2 \right) \|\hat{\mu}_k - \mu^\star(\phi_{\omega_k})\|^2$$
$$\leq (1 - \frac{1-\delta}{2}\xi_k)\varepsilon_k^\mu, \tag{51}$$

where the second equation plugs in the definition of $\bar{H}$ in (25), and the first inequality is due to the Lipschitz continuity of $\bar{H}$, the second inequality follows from Assumption 4 and the Lipschitz continuity of $\pi^\star$ from Assumption 2, and the final inequality is due to the step size condition $\xi_k \leq \frac{1-\delta}{8L^2 L_H^2}$.

For the fourth term of (50),

$$2\xi_k \mathbb{E}[\langle \hat{\mu}_k - \mu^\star(\phi_{\omega_k}) + \xi_k \bar{H}(\pi^\star(\phi_{\omega_k}, \hat{\mu}_k), \omega_k, \hat{\mu}_k), \Delta h_k \rangle]$$
$$\leq 2\xi_k \mathbb{E}[\langle \hat{\mu}_k - \mu^\star(\phi_{\omega_k}) + \xi_k \bar{H}(\pi^\star(\phi_{\omega_k}, \hat{\mu}_k), \omega_k, \hat{\mu}_k), \mathbb{E}[\Delta h_k \mid \mathcal{F}_{k-1}] \rangle]$$
$$= 0. \tag{52}$$

For the fifth term of (50), we have from Lemma 5

$$2\langle \hat{\mu}_k - \mu^\star(\phi_{\omega_k}) + \xi_k \bar{H}(\pi^\star(\phi_{\omega_k}, \hat{\mu}_k), \omega_k, \hat{\mu}_k), \mu^\star(\phi_{\omega_{k+1}}) - \mu^\star(\phi_{\omega_k}) \rangle$$
$$\leq \frac{(1-\delta)\xi_k}{8} \mathbb{E}[\|\hat{\mu}_k - \mu^\star(\phi_{\omega_k}) + \xi_k \bar{H}(\pi^\star(\phi_{\omega_k}, \hat{\mu}_k), \omega_k, \hat{\mu}_k)\|^2] + \frac{48L^2 L_V^2 L_D^2 \zeta_k^2}{(1-\delta)\xi_k} \mathbb{E}[\varepsilon_k^\mu]$$
$$+ \frac{8L^2 \eta_2^2 \zeta_k^2}{(1-\delta)\xi_k} \mathbb{E}[\varepsilon_k^\phi] + \frac{32L^2 L_D^2 \zeta_k^2}{(1-\delta)\xi_k} \mathbb{E}[\varepsilon_{l,k}^V] + B_D^2 L \zeta_k^2$$
$$\leq \frac{1-\delta}{8}\xi_k \mathbb{E}[\varepsilon_k^\mu] + \frac{48L^2 L_V^2 L_D^2 \zeta_k^2}{(1-\delta)\xi_k} \mathbb{E}[\varepsilon_k^\mu] + \frac{8L^2 \eta_2^2 \zeta_k^2}{(1-\delta)\xi_k} \mathbb{E}[\varepsilon_k^\phi] + \frac{32L^2 L_D^2 \zeta_k^2}{(1-\delta)\xi_k} \mathbb{E}[\varepsilon_{l,k}^V] + B_D^2 L \zeta_k^2$$
$$\leq \frac{1-\delta}{4}\xi_k \mathbb{E}[\varepsilon_k^\mu] + \frac{8L^2 \eta_2^2 \zeta_k^2}{(1-\delta)\xi_k} \mathbb{E}[\varepsilon_k^\phi] + \frac{32L^2 L_D^2 \zeta_k^2}{(1-\delta)\xi_k} \mathbb{E}[\varepsilon_{l,k}^V] + B_D^2 L \zeta_k^2, \tag{53}$$

where the second inequality plugs in (51), and the last inequality is due to the step size condition $\frac{\zeta_k}{\xi_k} \leq \frac{1-\delta}{8\sqrt{6}L L_V L_D}$.

Similarly, for the final term of (50),

$$2\xi_k \langle \hat{\mu}_k - \mu^\star(\phi_{\omega_k}) + \xi_k \bar{H}(\pi^\star(\phi_{\omega_k}, \hat{\mu}_k), \omega_k, \hat{\mu}_k), \bar{H}(\pi^\star(\phi_{\omega_k}, \hat{\mu}_k), \omega_k, \hat{\mu}_k) - \bar{H}(\pi_{\theta_k}, \omega_k, \hat{\mu}_k) \rangle$$
$$\leq \frac{1-\delta}{8}\xi_k \|\hat{\mu}_k - \mu^\star(\phi_{\omega_k}) + \xi_k \bar{H}(\pi^\star(\phi_{\omega_k}, \hat{\mu}_k), \omega_k, \hat{\mu}_k)\|^2$$

$$+ \frac{8}{1-\delta}\xi_k\|\bar{H}(\pi^\star(\phi_{\omega_k},\hat{\mu}_k),\omega_k,\hat{\mu}_k) - \bar{H}(\pi_{\theta_k},\omega_k,\hat{\mu}_k)\|^2$$

$$\leq \frac{1-\delta}{8}\xi_k\varepsilon_k^\mu + \frac{8}{1-\delta}\xi_k\|\bar{H}(\pi^\star(\phi_{\omega_k},\hat{\mu}_k),\omega_k,\hat{\mu}_k) - \bar{H}(\pi_{\theta_k},\omega_k,\hat{\mu}_k)\|^2. \tag{54}$$

Taking expectation and substituting (51)-(54) into (50),

$$\mathbb{E}[\varepsilon_{k+1}^\mu] \leq (1 - \frac{1-\delta}{2}\xi_k)\mathbb{E}[\varepsilon_k^\mu] + 3\xi_k^2 \cdot 4B_H^2 + 3\mathbb{E}[\|\mu^\star(\phi_{\omega_{k+1}}) - \mu^\star(\phi_{\omega_k})\|^2]$$

$$+ 3\xi_k^2\mathbb{E}[\|\bar{H}(\pi_{\theta_k},\omega_k,\hat{\mu}_k) - \bar{H}(\pi^\star(\phi_{\omega_k},\hat{\mu}_k),\omega_k,\hat{\mu}_k)\|^2]$$

$$+ \frac{1-\delta}{4}\xi_k\mathbb{E}[\varepsilon_k^\mu] + \frac{8L^2\eta_2^2\zeta_k^2}{(1-\delta)\xi_k}\mathbb{E}[\varepsilon_k^\phi] + \frac{32L^2L_D^2\zeta_k^2}{(1-\delta)\xi_k}\mathbb{E}[\varepsilon_{l,k}^V] + B_D^2L\zeta_k^2$$

$$+ \frac{1-\delta}{8}\xi_k\mathbb{E}[\varepsilon_k^\mu] + \frac{8}{1-\delta}\xi_k\mathbb{E}[\|\bar{H}(\pi^\star(\phi_{\omega_k},\hat{\mu}_k),\omega_k,\hat{\mu}_k) - \bar{H}(\pi_{\theta_k},\omega_k,\hat{\mu}_k)\|^2]$$

$$\leq (1 - \frac{1-\delta}{8}\xi_k)\mathbb{E}[\varepsilon_k^\mu] + 12B_H^2\xi_k^2 + 3L^2\mathbb{E}[\|\omega_{k+1} - \omega_k\|^2]$$

$$+ \frac{8L^2\eta_2^2\zeta_k^2}{(1-\delta)\xi_k}\mathbb{E}[\varepsilon_k^\phi] + \frac{32L^2L_D^2\zeta_k^2}{(1-\delta)\xi_k}\mathbb{E}[\varepsilon_{l,k}^V] + B_D^2L\zeta_k^2$$

$$+ \frac{11\xi_k}{1-\delta}\mathbb{E}[\|\bar{H}(\pi_{\theta_k},\omega_k,\hat{\mu}_k) - \bar{H}(\pi^\star(\phi_{\omega_k},\hat{\mu}_k),\omega_k,\hat{\mu}_k)\|^2]$$

$$\leq (1 - \frac{1-\delta}{8}\xi_k)\mathbb{E}[\varepsilon_k^\mu] + \frac{11L_H^2\xi_k}{1-\delta}\mathbb{E}[\|\pi_{\theta_k} - \pi^\star(\phi_{\omega_k},\hat{\mu}_k)\|^2] + 12B_H^2\xi_k^2 + 3B_D^2L^2\zeta_k^2$$

$$+ \frac{8L^2\eta_2^2\zeta_k^2}{(1-\delta)\xi_k}\mathbb{E}[\varepsilon_k^\phi] + \frac{32L^2L_D^2\zeta_k^2}{(1-\delta)\xi_k}\mathbb{E}[\varepsilon_{l,k}^V] + B_D^2L\zeta_k^2$$

$$\leq (1 - \frac{1-\delta}{8}\xi_k)\mathbb{E}[\varepsilon_k^\mu] + \frac{11L_H^2\xi_k}{1-\delta}\mathbb{E}[\|\pi_{\theta_k} - \pi^\star(\phi_{\omega_k},\hat{\mu}_k)\|^2]$$

$$+ \frac{8L^2\eta_2^2\zeta_k^2}{(1-\delta)\xi_k}\mathbb{E}[\varepsilon_k^\phi] + \frac{32L^2L_D^2\zeta_k^2}{(1-\delta)\xi_k}\mathbb{E}[\varepsilon_{l,k}^V] + 16B_H^2\xi_k^2, \tag{55}$$

where the second inequality follows from the step size condition $\xi_k \leq \frac{1}{1-\delta}$, and the third inequality is a result of the Lipschitz continuity of $\mu^\star$ and $\bar{H}$, and the last inequality applies the step size condition $\frac{\zeta_k}{\xi_k} \leq \frac{B_H}{B_D L}$.

Note that the following inequality holds for any $\pi, \phi, \mu$ as a result of the entropy regularization (adapted from Lemma 1 of Zeng et al. [2022])

$$\|\pi - \pi^\star(\phi,\mu)\|^2 \leq \frac{1}{\tau\rho_{\min}}\Big(J_f(\pi^\star(\phi,\mu),\phi,\mu) - J_f(\pi,\phi,\mu)\Big). \tag{56}$$

Combining (55) and (56),

$$\mathbb{E}[\varepsilon_{k+1}^\mu] \leq (1 - \frac{1-\delta}{8}\xi_k)\mathbb{E}[\varepsilon_k^\mu] + \frac{11L_H^2\xi_k}{(1-\delta)\tau\rho_{\min}}\mathbb{E}\Big[J_f(\pi^\star(\phi_{\omega_k},\hat{\mu}_k),\phi_{\omega_k},\hat{\mu}_k) - J_f(\pi_{\theta_k},\phi_{\omega_k},\hat{\mu}_k)\Big]$$

$$+ \frac{8L^2\eta_2^2\zeta_k^2}{(1-\delta)\xi_k}\mathbb{E}[\varepsilon_k^\phi] + \frac{32L^2L_D^2\zeta_k^2}{(1-\delta)\xi_k}\mathbb{E}[\varepsilon_{l,k}^V] + 16B_H^2\xi_k^2$$

$$= (1 - \frac{1-\delta}{4}\xi_k)\mathbb{E}[\varepsilon_k^\mu] + \frac{11L_H^2\xi_k}{(1-\delta)\tau\rho_{\min}}\mathbb{E}[\varepsilon_k^\pi] + \frac{8L^2\eta_2^2\zeta_k^2}{(1-\delta)\xi_k}\mathbb{E}[\varepsilon_k^\phi] + \frac{32L^2L_D^2\zeta_k^2}{(1-\delta)\xi_k}\mathbb{E}[\varepsilon_{l,k}^V] + 16B_H^2\xi_k^2.$$

∎

## C.3 Proof of Proposition 3

The proof of the proposition relies on the intermediate result below which bounds an important cross term.

**Lemma 6** *Under the assumptions and step sizes of Proposition 3, we have for all $k \geq 0$*

$$- \mathbb{E}[\langle \left[ \begin{array}{c} \nabla_\omega J_f(\pi_{\theta_k}, \phi_{\omega_k}, \hat{\mu}_k) - \nabla_\omega J_f(\pi, \phi_{\omega_k}, \hat{\mu}_k) \mid_{\pi=\pi^\star(\phi_{\omega_k}, \hat{\mu}_k)} \\ \nabla_\mu J_f(\pi_{\theta_k}, \phi_{\omega_k}, \hat{\mu}_k) - \nabla_\mu J_f(\pi, \phi_{\omega_k}, \hat{\mu}_k) \mid_{\pi=\pi^\star(\phi_{\omega_k}, \hat{\mu}_k)} \end{array} \right], \left[ \begin{array}{c} \omega_{k+1} - \omega_k \\ \hat{\mu}_{k+1} - \hat{\mu}_k \end{array} \right] \rangle]$$

$$\leq \frac{(1-\gamma)\tau^2 \rho_{\min}^3 p_{\min}^2 \alpha_k}{4|\mathcal{S}|} \mathbb{E}[\|\pi^\star(\phi_{\omega_k}, \hat{\mu}_k) - \pi_{\theta_k}\|^2] + \frac{2|\mathcal{S}|L_V^2 L_D^2 L_H^2 \xi_k^2}{(1-\gamma)\tau^2 \rho_{\min}^3 p_{\min}^2 \alpha_k} \mathbb{E}[\|\pi^\star(\phi_{\omega_k}, \hat{\mu}_k) - \pi_{\theta_k}\|^2]$$

$$+ \frac{2|\mathcal{S}|L_V^2 \eta_2^2 \zeta_k^2}{(1-\gamma)\tau^2 \rho_{\min}^3 p_{\min}^2 \alpha_k} \mathbb{E}[\varepsilon_k^\phi] + \frac{16|\mathcal{S}|L^2 L_V^4 L_D^2 L_H^2 \xi_k^2}{(1-\gamma)\tau^2 \rho_{\min}^3 p_{\min}^2 \alpha_k} \mathbb{E}[\varepsilon_k^\mu] + \frac{8|\mathcal{S}|L_V^2 \zeta_k^2}{(1-\gamma)\tau^2 \rho_{\min}^3 p_{\min}^2 \alpha_k} \mathbb{E}[\varepsilon_{l,k}^V].$$

We provide the proof of Lemma 6 in Section D.6.

As $J_f$ has $L_V$-Lipschitz gradients,

$$J_f(\pi_{\theta_k}, \phi_{\omega_{k+1}}, \hat{\mu}_{k+1}) - J_f(\pi_{\theta_{k+1}}, \phi_{\omega_{k+1}}, \hat{\mu}_{k+1})$$

$$\leq -\langle \nabla_\theta J_f(\pi_{\theta_k}, \phi_{\omega_{k+1}}, \hat{\mu}_{k+1}), \theta_{k+1} - \theta_k \rangle + \frac{L_V}{2} \|\theta_{k+1} - \theta_k\|^2$$

$$= -\alpha_k \langle \nabla_\theta J_f(\pi_{\theta_k}, \phi_{\omega_{k+1}}, \hat{\mu}_{k+1}), F(\theta_k, \omega_k, \hat{\mu}_k, \hat{V}_{f,k}, s_k, a_k, b_k, s_k') \rangle + \frac{L_V \alpha_k^2}{2} \|F(\theta_k, \omega_k, \hat{\mu}_k, \hat{V}_{f,k}, s_k, a_k, b_k, s_k')\|^2$$

$$= -\alpha_k \langle \nabla_\theta J_f(\pi_{\theta_k}, \phi_{\omega_{k+1}}, \hat{\mu}_{k+1}), F(\theta_k, \omega_k, \hat{\mu}_k, \hat{V}_{f,k}, s_k, a_k, b_k, s_k') - \bar{F}(\theta_k, \omega_k, \hat{\mu}_k, \hat{V}_{f,k}) \rangle$$

$$\quad - \alpha_k \langle \nabla_\theta J_f(\pi_{\theta_k}, \phi_{\omega_{k+1}}, \hat{\mu}_{k+1}), \bar{F}(\theta_k, \omega_k, \hat{\mu}_k, \hat{V}_{f,k}) - \bar{F}(\theta_k, \omega_k, \hat{\mu}_k, V_f^{\pi_{\theta_k}, \phi_{\omega_k}, \hat{\mu}_k}) \rangle$$

$$\quad - \alpha_k \langle \nabla_\theta J_f(\pi_{\theta_k}, \phi_{\omega_{k+1}}, \hat{\mu}_{k+1}), \nabla_\theta J_f(\pi_{\theta_k}, \phi_{\omega_k}, \hat{\mu}_k) \rangle + \frac{L_V \alpha_k^2}{2} \|F(\theta_k, \omega_k, \hat{\mu}_k, \hat{V}_{f,k}, s_k, a_k, b_k, s_k')\|^2, \tag{57}$$

where the final equation follows from $\nabla_\theta J_f(\pi_\theta, \phi_\omega, \mu) = \bar{F}(\theta, \omega, \mu, V_f^{\pi_\theta, \phi_\omega, \mu})$ for all $\theta, \omega, \mu$.

For the first term of (57),

$$- \alpha_k \mathbb{E}[\langle \nabla_\theta J_f(\pi_{\theta_k}, \phi_{\omega_{k+1}}, \hat{\mu}_{k+1}), F(\theta_k, \omega_k, \hat{\mu}_k, \hat{V}_{f,k}, s_k, a_k, b_k, s_k') - \bar{F}(\theta_k, \omega_k, \hat{\mu}_k, \hat{V}_{f,k}) \rangle]$$

$$= -\alpha_k \mathbb{E}[\langle \nabla_\theta J_f(\pi_{\theta_k}, \phi_{\omega_k}, \hat{\mu}_k), \mathbb{E}[F(\theta_k, \omega_k, \hat{\mu}_k, \hat{V}_{f,k}, s_k, a_k, b_k, s_k') - \bar{F}(\theta_k, \omega_k, \hat{\mu}_k, \hat{V}_{f,k}) \mid \mathcal{F}_{k-1}] \rangle]$$

$$\quad + \alpha_k \mathbb{E}[\langle \nabla_\theta J_f(\pi_{\theta_k}, \phi_{\omega_k}, \hat{\mu}_k) - \nabla_\theta J_f(\pi_{\theta_k}, \phi_{\omega_{k+1}}, \hat{\mu}_{k+1}), F(\theta_k, \omega_k, \hat{\mu}_k, \hat{V}_{f,k}, s_k, a_k, b_k, s_k') - \bar{F}(\theta_k, \omega_k, \hat{\mu}_k, \hat{V}_{f,k}) \rangle]$$

$$= \alpha_k \mathbb{E}[\langle \nabla_\theta J_f(\pi_{\theta_k}, \phi_{\omega_k}, \hat{\mu}_k) - \nabla_\theta J_f(\pi_{\theta_k}, \phi_{\omega_{k+1}}, \hat{\mu}_{k+1}), F(\theta_k, \omega_k, \hat{\mu}_k, \hat{V}_{f,k}, s_k, a_k, b_k, s_k') - \bar{F}(\theta_k, \omega_k, \hat{\mu}_k, \hat{V}_{f,k}) \rangle]$$

$$\leq 2B_F \alpha_k \cdot L_V \mathbb{E}[\|\phi_{\omega_{k+1}} - \phi_{\omega_k}\| + \|\hat{\mu}_{k+1} - \hat{\mu}_k\|]$$

$$\leq 2B_F L_V \alpha_k (B_D \zeta_k + B_H \xi_k)$$

$$\leq 2B_F (B_D + B_H) \alpha_k \xi_k, \tag{58}$$

where the last inequality follows from $\zeta_k \leq \xi_k$.

For the second term of (57),

$$- \alpha_k \langle \nabla_\theta J_f(\pi_{\theta_k}, \phi_{\omega_{k+1}}, \hat{\mu}_{k+1}), \bar{F}(\theta_k, \omega_k, \hat{\mu}_k, \hat{V}_{f,k}) - \bar{F}(\theta_k, \omega_k, \hat{\mu}_k, V_f^{\pi_{\theta_k}, \phi_{\omega_k}, \hat{\mu}_k}) \rangle$$

$$\leq \frac{\alpha_k}{8} \|\nabla_\theta J_f(\pi_{\theta_k}, \phi_{\omega_{k+1}}, \hat{\mu}_{k+1})\|^2 + 2\alpha_k \|\bar{F}(\theta_k, \omega_k, \hat{\mu}_k, \hat{V}_{f,k}) - \bar{F}(\theta_k, \omega_k, \hat{\mu}_k, V_f^{\pi_{\theta_k}, \phi_{\omega_k}, \hat{\mu}_k})\|^2$$

$$\leq \frac{\alpha_k}{4} \|\nabla_\theta J_f(\pi_{\theta_k}, \phi_{\omega_k}, \hat{\mu}_k)\|^2 + \frac{\alpha_k}{4} \|\nabla_\theta J_f(\pi_{\theta_k}, \phi_{\omega_{k+1}}, \hat{\mu}_{k+1}) - \nabla_\theta J_f(\pi_{\theta_k}, \phi_{\omega_k}, \hat{\mu}_k)\|^2 + 2L_V^2 \alpha_k \varepsilon_{f,k}^V$$

$$\leq \frac{\alpha_k}{4} \|\nabla_\theta J_f(\pi_{\theta_k}, \phi_{\omega_k}, \hat{\mu}_k)\|^2 + \frac{L_V^2 \alpha_k}{2} \left( \|\phi_{\omega_{k+1}} - \phi_{\omega_k}\|^2 + \|\hat{\mu}_{k+1} - \hat{\mu}_k\|^2 \right) + 2L_V^2 \alpha_k \varepsilon_{f,k}^V$$

$$\leq \frac{\alpha_k}{4} \|\nabla_\theta J_f(\pi_{\theta_k}, \phi_{\omega_k}, \hat{\mu}_k)\|^2 + \frac{L_V^2 \alpha_k}{2} (B_D^2 \zeta_k^2 + B_H^2 \xi_k^2) + 2L_V^2 \alpha_k \varepsilon_{f,k}^V$$

$$\leq \frac{\alpha_k}{4} \|\nabla_\theta J_f(\pi_{\theta_k}, \phi_{\omega_k}, \hat{\mu}_k)\|^2 + 2L_V^2 \alpha_k \varepsilon_{f,k}^V + L_V^2 \xi_k^2, \tag{59}$$

where the third inequality is a result of the Lipschitz continuity of the value function, and the last inequality follows from the step size condition $\alpha_k \leq \frac{2}{B_D^2 + B_H^2}$ and $\zeta_k \leq \xi_k$.

For the third term of (57),

$$- \alpha_k \langle \nabla_\theta J_f(\pi_{\theta_k}, \phi_{\omega_{k+1}}, \hat{\mu}_{k+1}), \nabla_\theta J_f(\pi_{\theta_k}, \phi_{\omega_k}, \hat{\mu}_k) \rangle$$

$$= -\alpha_k \|\nabla_\theta J_f(\pi_{\theta_k}, \phi_{\omega_k}, \hat{\mu}_k)\|^2 + \alpha_k \langle \nabla_\theta J_f(\pi_{\theta_k}, \phi_{\omega_k}, \hat{\mu}_k) - \nabla_\theta J_f(\pi_{\theta_k}, \phi_{\omega_{k+1}}, \hat{\mu}_{k+1}), \nabla_\theta J_f(\pi_{\theta_k}, \phi_{\omega_k}, \hat{\mu}_k) \rangle$$

$$\leq -\frac{\alpha_k}{2} \|\nabla_\theta J_f(\pi_{\theta_k}, \phi_{\omega_k}, \hat{\mu}_k)\|^2 + \frac{\alpha_k}{2} \|\nabla_\theta J_f(\pi_{\theta_k}, \phi_{\omega_k}, \hat{\mu}_k) - \nabla_\theta J_f(\pi_{\theta_k}, \phi_{\omega_{k+1}}, \hat{\mu}_{k+1})\|^2$$

$$\leq -\frac{\alpha_k}{2} \|\nabla_\theta J_f(\pi_{\theta_k}, \phi_{\omega_k}, \hat{\mu}_k)\|^2 + L_V^2 \alpha_k \Big( \|\phi_{\omega_{k+1}} - \phi_{\omega_k}\|^2 + \|\hat{\mu}_{k+1} - \hat{\mu}_k\|^2 \Big)$$

$$\leq -\frac{\alpha_k}{2} \|\nabla_\theta J_f(\pi_{\theta_k}, \phi_{\omega_k}, \hat{\mu}_k)\|^2 + L_V^2 \alpha_k (B_D^2 \zeta_k^2 + B_H^2 \xi_k^2)$$

$$\leq -\frac{\alpha_k}{2} \|\nabla_\theta J_f(\pi_{\theta_k}, \phi_{\omega_k}, \hat{\mu}_k)\|^2 + 2L_V^2 \xi_k^2, \tag{60}$$

where the last inequality again uses the step size condition $\alpha_k \leq \frac{2}{B_D^2 + B_H^2}$ and $\zeta_k \leq \xi_k$.

Substituting (58)-(60) into (57) and taking the expectation,

$$\mathbb{E}[J_f(\pi_{\theta_k}, \phi_{\omega_{k+1}}, \hat{\mu}_{k+1}) - J_f(\pi_{\theta_{k+1}}, \phi_{\omega_{k+1}}, \hat{\mu}_{k+1})]$$

$$\leq 2B_F(B_D + B_H)\alpha_k \xi_k + \frac{\alpha_k}{4} \mathbb{E}[\|\nabla_\theta J_f(\pi_{\theta_k}, \phi_{\omega_k}, \hat{\mu}_k)\|^2] + 2L_V^2 \alpha_k \mathbb{E}[\varepsilon_{f,k}^V] + L_V^2 \xi_k^2$$

$$\qquad - \frac{\alpha_k}{2} \mathbb{E}[\|\nabla_\theta J_f(\pi_{\theta_k}, \phi_{\omega_k}, \hat{\mu}_k)\|^2] + 2L_V^2 \xi_k^2$$

$$\leq -\frac{\alpha_k}{4} \mathbb{E}[\|\nabla_\theta J_f(\pi_{\theta_k}, \phi_{\omega_k}, \hat{\mu}_k)\|^2] + 2L_V^2 \alpha_k \mathbb{E}[\varepsilon_{f,k}^V] + 3L_V^2 \xi_k^2 + 2B_F(B_D + B_H)\alpha_k \xi_k. \tag{61}$$

Under entropy regularization, $J_f$ satisfies a non-uniform PL condition with respect to the follower's policy [Mei et al., 2020]. Specifically, for any $\theta, \phi, \mu$,

$$\|\nabla_\theta J_f(\pi_\theta, \phi, \mu)\|^2 \geq \frac{2(1-\gamma)\tau \rho_{\min}^2 p_{\min}^2}{|\mathcal{S}|} \Big( J_f(\pi^\star(\phi, \mu), \phi, \mu) - J_f(\pi_\theta, \phi, \mu) \Big). \tag{62}$$

The inequality in this specific form is adapted from Zeng et al. [2022][Lemma 4].

Combining (61) with (62) leads to

$$\mathbb{E}[J_f(\pi_{\theta_k}, \phi_{\omega_{k+1}}, \hat{\mu}_{k+1}) - J_f(\pi_{\theta_{k+1}}, \phi_{\omega_{k+1}}, \hat{\mu}_{k+1})]$$

$$\leq -\frac{\alpha_k}{16} \mathbb{E}[\|\nabla_\theta J_f(\pi_{\theta_k}, \phi_{\omega_k}, \hat{\mu}_k)\|^2] + 2L_V^2 \alpha_k \mathbb{E}[\varepsilon_{f,k}^V] + 3L_V^2 \xi_k^2 + 2B_F(B_D + B_H)\alpha_k \xi_k$$

$$\qquad - \frac{3(1-\gamma)\tau \rho_{\min}^2 p_{\min}^2 \alpha_k}{8|\mathcal{S}|} \Big( J_f(\pi^\star(\phi_{\omega_k}, \hat{\mu}_k), \phi_{\omega_k}, \hat{\mu}_k) - J_f(\pi_{\theta_k}, \phi_{\omega_k}, \hat{\mu}_k) \Big)$$

$$\leq -\frac{\alpha_k}{16} \mathbb{E}[\|\nabla_\theta J_f(\pi_{\theta_k}, \phi_{\omega_k}, \hat{\mu}_k)\|^2] + 2L_V^2 \alpha_k \mathbb{E}[\varepsilon_{f,k}^V] + 3L_V^2 \xi_k^2 + 2B_F(B_D + B_H)\alpha_k \xi_k$$

$$\qquad - \frac{3(1-\gamma)\tau^2 \rho_{\min}^3 p_{\min}^2 \alpha_k}{8|\mathcal{S}|} \|\pi^\star(\phi_{\omega_k}, \hat{\mu}_k) - \pi_{\theta_k}\|^2. \tag{63}$$

where the last inequality follows from (56).

Within this subsection, we denote $x_k = (\phi_{\omega_k}, \hat{\mu}_k)$. Due to the $L_V$-smoothness of the function $J_f$, we have

$$J_f(\pi_{\theta_k}, \phi_{\omega_k}, \hat{\mu}_k) - J_f(\pi_{\theta_k}, \phi_{\omega_{k+1}}, \hat{\mu}_{k+1})$$

$$\leq -\langle \nabla_x J_f(\pi_{\theta_k}, \phi_{\omega_k}, \hat{\mu}_k), x_{k+1} - x_k \rangle + \frac{L_V}{2} \|x_{k+1} - x_k\|^2$$

$$= -\langle \nabla_x J_f(\pi_{\theta_k}, \phi_{\omega_k}, \hat{\mu}_k) - \nabla_x J_f(\pi, \phi_{\omega_k}, \hat{\mu}_k)\mid_{\pi = \pi^\star(\phi_{\omega_k}, \hat{\mu}_k)}, x_{k+1} - x_k \rangle + \frac{L_V}{2} \|x_{k+1} - x_k\|^2$$

$$\qquad - \langle \nabla_x J_f(\pi, \phi_{\omega_k}, \hat{\mu}_k)\mid_{\pi = \pi^\star(\phi_{\omega_k}, \hat{\mu}_k)}, x_{k+1} - x_k \rangle$$

$$= -\langle \nabla_x J_f(\pi_{\theta_k}, \phi_{\omega_k}, \hat{\mu}_k) - \nabla_x J_f(\pi, \phi_{\omega_k}, \hat{\mu}_k)\mid_{\pi = \pi^\star(\phi_{\omega_k}, \hat{\mu}_k)}, x_{k+1} - x_k \rangle + \frac{L_V}{2} \|x_{k+1} - x_k\|^2$$

$$\qquad - \langle \nabla_x J_f(\pi^\star(\phi_{\omega_k}, \hat{\mu}_k), \phi_{\omega_k}, \hat{\mu}_k), x_{k+1} - x_k \rangle$$

$$\leq -\langle \nabla_x J_f(\pi_{\theta_k}, \phi_{\omega_k}, \hat{\mu}_k) - \nabla_x J_f(\pi, \phi_{\omega_k}, \hat{\mu}_k)\mid_{\pi = \pi^\star(\phi_{\omega_k}, \hat{\mu}_k)}, x_{k+1} - x_k \rangle + \frac{L_V}{2} \|x_{k+1} - x_k\|^2$$

$$\qquad + \Big( J_f(\pi^\star(\phi_{\omega_k}, \hat{\mu}_k), \phi_{\omega_k}, \hat{\mu}_k) - J_f(\pi^\star(\phi_{\omega_{k+1}}, \hat{\mu}_{k+1}), \phi_{\omega_{k+1}}, \hat{\mu}_{k+1}) \Big) + \frac{L_V}{2} \|x_{k+1} - x_k\|^2,$$

where the second inequality follows from the fact that for an $L$-smooth function $f : \mathbb{R}^d \to \mathbb{R}$ we have $f(x) - f(y) \geq -\langle \nabla f(x), y - x \rangle - \frac{L}{2}\|y - x\|^2$. The second equations above uses the relationship $\nabla_x J_f(\pi^\star(\phi_{\omega_k}, \hat{\mu}_k), \phi_{\omega_k}, \hat{\mu}_k) = \nabla_x J_f(\pi, \phi_{\omega_k}, \hat{\mu}_k) \mid_{\pi = \pi^\star(\phi_{\omega_k}, \hat{\mu}_k)}$, which holds as the partial gradient with respect to $\pi$ is zero at the optimizer (under entropy regularization $\tau > 0$, the policy $\pi^\star(\phi, \mu)$ is in the interior of the probability simplex for any $\phi, \mu$), i.e.

$$\frac{\partial J_f(\pi^\star(\phi_{\omega_k}, \hat{\mu}_k), \phi_{\omega_k}, \hat{\mu}_k)}{\partial \pi^\star(\phi_{\omega_k}, \hat{\mu}_k)} = 0.$$

Taking the expectation and plugging in the result from Lemma 6,

$$\mathbb{E}[J_f(\pi_{\theta_k}, \phi_{\omega_k}, \hat{\mu}_k) - J_f(\pi_{\theta_k}, \phi_{\omega_{k+1}}, \hat{\mu}_{k+1})]$$
$$\leq \frac{(1-\gamma)\tau^2 \rho_{\min}^3 p_{\min}^2 \alpha_k}{4|\mathcal{S}|} \mathbb{E}[\|\pi^\star(\phi_{\omega_k}, \hat{\mu}_k) - \pi_{\theta_k}\|^2] + \frac{2|\mathcal{S}|L_V^2 L_D^2 L_H^2 \xi_k^2}{(1-\gamma)\tau^2 \rho_{\min}^3 p_{\min}^2 \alpha_k} \mathbb{E}[\|\pi^\star(\phi_{\omega_k}, \hat{\mu}_k) - \pi_{\theta_k}\|^2]$$
$$+ \frac{2|\mathcal{S}|L_V^2 \eta_2^2 \zeta_k^2}{(1-\gamma)\tau^2 \rho_{\min}^3 p_{\min}^2 \alpha_k} \mathbb{E}[\varepsilon_k^\phi] + \frac{16|\mathcal{S}|L^2 L_V^4 L_D^2 L_H^2 \xi_k^2}{(1-\gamma)\tau^2 \rho_{\min}^3 p_{\min}^2 \alpha_k} \mathbb{E}[\varepsilon_k^\mu] + \frac{8|\mathcal{S}|L_V^2 \zeta_k^2}{(1-\gamma)\tau^2 \rho_{\min}^3 p_{\min}^2 \alpha_k} \mathbb{E}[\varepsilon_{l,k}^V]$$
$$+ \left( J_f(\pi^\star(\phi_{\omega_k}, \hat{\mu}_k), \phi_{\omega_k}, \hat{\mu}_k) - J_f(\pi^\star(\phi_{\omega_{k+1}}, \hat{\mu}_{k+1}), \phi_{\omega_{k+1}}, \hat{\mu}_{k+1}) \right) + L_V \mathbb{E}[\|x_{k+1} - x_k\|^2].$$
$$(64)$$

Now, we combine (63) and (64). Recall the definition of $\varepsilon_k^\pi$ in (15).

$$\mathbb{E}[\varepsilon_{k+1}^\pi - \varepsilon_k^\pi]$$
$$= \mathbb{E}[J_f(\pi_{\theta_k}, \phi_{\omega_k}, \hat{\mu}_k) - J_f(\pi_{\theta_k}, \phi_{\omega_{k+1}}, \hat{\mu}_{k+1})]$$
$$\quad + \mathbb{E}[J_f(\pi_{\theta_k}, \phi_{\omega_{k+1}}, \hat{\mu}_{k+1}) - J_f(\pi_{\theta_{k+1}}, \phi_{\omega_{k+1}}, \hat{\mu}_{k+1})]$$
$$\quad + \mathbb{E}[J_f(\pi^\star(\phi_{\omega_{k+1}}, \hat{\mu}_{k+1}), \phi_{\omega_{k+1}}, \hat{\mu}_{k+1}) - J_f(\pi^\star(\phi_{\omega_k}, \hat{\mu}_k), \phi_{\omega_k}, \hat{\mu}_k)]$$
$$\leq \frac{(1-\gamma)\tau^2 \rho_{\min}^3 p_{\min}^2 \alpha_k}{4|\mathcal{S}|} \mathbb{E}[\|\pi^\star(\phi_{\omega_k}, \hat{\mu}_k) - \pi_{\theta_k}\|^2] + \frac{2|\mathcal{S}|L_V^2 L_D^2 L_H^2 \xi_k^2}{(1-\gamma)\tau^2 \rho_{\min}^3 p_{\min}^2 \alpha_k} \mathbb{E}[\|\pi^\star(\phi_{\omega_k}, \hat{\mu}_k) - \pi_{\theta_k}\|^2]$$
$$+ \frac{2|\mathcal{S}|L_V^2 \eta_2^2 \zeta_k^2}{(1-\gamma)\tau^2 \rho_{\min}^3 p_{\min}^2 \alpha_k} \mathbb{E}[\varepsilon_k^\phi] + \frac{16|\mathcal{S}|L^2 L_V^4 L_D^2 L_H^2 \xi_k^2}{(1-\gamma)\tau^2 \rho_{\min}^3 p_{\min}^2 \alpha_k} \mathbb{E}[\varepsilon_k^\mu] + \frac{8|\mathcal{S}|L_V^2 \zeta_k^2}{(1-\gamma)\tau^2 \rho_{\min}^3 p_{\min}^2 \alpha_k} \mathbb{E}[\varepsilon_{l,k}^V]$$
$$+ L_V (\|\omega_{k+1} - \omega_k\|^2 + \|\hat{\mu}_{k+1} - \hat{\mu}_k\|^2)$$
$$- \frac{\alpha_k}{16} \mathbb{E}[\|\nabla_\theta J_f(\pi_{\theta_k}, \phi_{\omega_k}, \hat{\mu}_k)\|^2] + 2L_V^2 \alpha_k \mathbb{E}[\varepsilon_{f,k}^V] + 3L_V^2 \xi_k^2 + 2B_F(B_D + B_H)\alpha_k \xi_k$$
$$- \frac{3(1-\gamma)\tau^2 \rho_{\min}^3 p_{\min}^2 \alpha_k}{8|\mathcal{S}|} \|\pi^\star(\phi_{\omega_k}, \hat{\mu}_k) - \pi_{\theta_k}\|^2$$
$$\leq - \frac{\alpha_k}{16} \mathbb{E}[\|\nabla_\theta J_f(\pi_{\theta_k}, \phi_{\omega_k}, \hat{\mu}_k)\|^2] + 2L_V^2 \alpha_k \mathbb{E}[\varepsilon_{f,k}^V]$$
$$+ \frac{2|\mathcal{S}|L_V^2 \eta_2^2 \zeta_k^2}{(1-\gamma)\tau^2 \rho_{\min}^3 p_{\min}^2 \alpha_k} \mathbb{E}[\varepsilon_k^\phi] + \frac{16|\mathcal{S}|L^2 L_V^4 L_D^2 L_H^2 \xi_k^2}{(1-\gamma)\tau^2 \rho_{\min}^3 p_{\min}^2 \alpha_k} \mathbb{E}[\varepsilon_k^\mu] + \frac{8|\mathcal{S}|L_V^2 \zeta_k^2}{(1-\gamma)\tau^2 \rho_{\min}^3 p_{\min}^2 \alpha_k} \mathbb{E}[\varepsilon_{l,k}^V]$$
$$+ 3L_V^2 \xi_k^2 + 2B_F(B_D + B_H)\alpha_k \xi_k + L_V(B_D^2 \zeta_k^2 + B_H^2 \xi_k^2)$$
$$\leq - \frac{\alpha_k}{16} \mathbb{E}[\|\nabla_\theta J_f(\pi_{\theta_k}, \phi_{\omega_k}, \hat{\mu}_k)\|^2] + \frac{2|\mathcal{S}|L_V^2 \eta_2^2 \zeta_k^2}{(1-\gamma)\tau^2 \rho_{\min}^3 p_{\min}^2 \alpha_k} \mathbb{E}[\varepsilon_k^\phi] + \frac{16|\mathcal{S}|L^2 L_V^4 L_D^2 L_H^2 \xi_k^2}{(1-\gamma)\tau^2 \rho_{\min}^3 p_{\min}^2 \alpha_k} \mathbb{E}[\varepsilon_k^\mu]$$
$$+ \frac{8|\mathcal{S}|L_V^2 \zeta_k^2}{(1-\gamma)\tau^2 \rho_{\min}^3 p_{\min}^2 \alpha_k} \mathbb{E}[\varepsilon_{l,k}^V] + 2L_V^2 \alpha_k \mathbb{E}[\varepsilon_{f,k}^V] + 3B_F(B_D + B_H)\alpha_k \xi_k,$$

where the second inequality uses the step size condition $\frac{\xi_k}{\alpha_k} \leq \frac{(1-\gamma)\tau^2 \rho_{\min}^3 p_{\min}^2}{4|\mathcal{S}|L_V L_D L_H}$, and the last inequality is due to the step size condition $\frac{\zeta_k}{\xi_k} \leq \frac{B_H}{B_D}$ and $\frac{\xi_k}{\alpha_k} \leq \frac{B_F(B_D + B_H)}{L_V(2B_H^2 + 3L_V)}$.

∎

## C.4 Proof of Proposition 4

We introduce a technical lemma below, which bounds an important cross term in the proof of the proposition.

**Lemma 7** *Under the assumptions and step sizes of Proposition 4, we have for all $k \geq 0$*

$$\mathbb{E}[\langle \hat{V}_{f,k} - V_f^{\pi_{\theta_k},\phi_{\omega_k},\hat{\mu}_k} + \beta_k \bar{G}_f(\theta_k,\omega_k,\hat{\mu}_k,\hat{V}_{f,k}), V_f^{\pi_{\theta_k},\phi_{\omega_k},\hat{\mu}_k} - V_f^{\pi_{\theta_{k+1}},\phi_{\omega_{k+1}},\hat{\mu}_{k+1}}\rangle]$$

$$\leq \frac{(1-\gamma)\beta_k}{2}\mathbb{E}[\|\hat{V}_{f,k} - V_f^{\pi_{\theta_k},\phi_{\omega_k},\hat{\mu}_k} + \beta_k \bar{G}_f(\theta_k,\omega_k,\hat{\mu}_k,\hat{V}_{f,k})\|^2]$$

$$+ \frac{6L_V^2\alpha_k^2}{(1-\gamma)\beta_k}\mathbb{E}[\|\nabla_\theta J_f(\pi_{\theta_k},\phi_{\omega_k},\hat{\mu}_k)\|^2] + \frac{6L_V^2 L_H^2 \xi_k^2}{(1-\gamma)\beta_k}\mathbb{E}[\varepsilon_k^\pi] + \frac{6L_V^2 \eta_2^2 \zeta_k^2}{(1-\gamma)\beta_k}\mathbb{E}[\varepsilon_k^\phi]$$

$$+ \frac{16L^2 L_V^2 L_H^2 \xi_k^2}{(1-\gamma)\beta_k}\mathbb{E}[\varepsilon_k^\mu] + \frac{6L_V^4 \zeta_k^2}{(1-\gamma)\beta_k}\mathbb{E}[\varepsilon_{f,k}^V] + \frac{24L_V^2 L_D^2 \alpha_k^2}{(1-\gamma)\beta_k}\mathbb{E}[\varepsilon_{l,k}^V] + L_{VV}(B_F + B_D + B_H)^2\alpha_k^2.$$

The proof of the lemma can be found in Section D.7.

By the definition of $\varepsilon_{f,k}^V$ and the update rule $\hat{V}_{f,k}$,

$$\varepsilon_{f,k+1}^V = \|\hat{V}_{f,k+1} - V_f^{\pi_{\theta_{k+1}},\phi_{\omega_{k+1}},\hat{\mu}_{k+1}}\|^2$$

$$= \|\Pi_{B_V}\left(\hat{V}_{f,k} + \beta_k G_f(\theta_k,\hat{\mu}_k,\hat{V}_{f,k},s_k,a_k,b_k,s_k')\right) - V_f^{\pi_{\theta_{k+1}},\phi_{\omega_{k+1}},\hat{\mu}_{k+1}}\|^2$$

$$\leq \left\|\hat{V}_{f,k} + \beta_k G_f(\theta_k,\hat{\mu}_k,\hat{V}_{f,k},s_k,a_k,b_k,s_k') - V_f^{\pi_{\theta_{k+1}},\phi_{\omega_{k+1}},\hat{\mu}_{k+1}}\right\|^2$$

$$= \left\|\hat{V}_{f,k} - V_f^{\pi_{\theta_k},\phi_{\omega_k},\hat{\mu}_k} + \beta_k \bar{G}_f(\theta_k,\omega_k,\hat{\mu}_k,\hat{V}_{f,k}) + \beta_k\left(G_f(\theta_k,\hat{\mu}_k,\hat{V}_{f,k},s_k,a_k,b_k,s_k') - \bar{G}_f(\theta_k,\omega_k,\hat{\mu}_k,\hat{V}_{f,k})\right)\right.$$

$$\left. + V_f^{\pi_{\theta_k},\phi_{\omega_k},\hat{\mu}_k} - V_f^{\pi_{\theta_{k+1}},\phi_{\omega_{k+1}},\hat{\mu}_{k+1}}\right\|^2$$

$$\leq \left\|\hat{V}_{f,k} - V_f^{\pi_{\theta_k},\phi_{\omega_k},\hat{\mu}_k} + \beta_k \bar{G}_f(\theta_k,\omega_k,\hat{\mu}_k,\hat{V}_{f,k})\right\|^2$$

$$+ 2\beta_k^2\|G_f(\theta_k,\hat{\mu}_k,\hat{V}_{f,k},s_k,a_k,b_k,s_k') - \bar{G}_f(\theta_k,\omega_k,\hat{\mu}_k,\hat{V}_{f,k})\|^2$$

$$+ 2\left\|V_f^{\pi_{\theta_k},\phi_{\omega_k},\hat{\mu}_k} - V_f^{\pi_{\theta_{k+1}},\phi_{\omega_{k+1}},\hat{\mu}_{k+1}}\right\|^2$$

$$+ \beta_k\langle\hat{V}_{f,k} - V_f^{\pi_{\theta_k},\phi_{\omega_k},\hat{\mu}_k} + \beta_k \bar{G}_f(\theta_k,\omega_k,\hat{\mu}_k,\hat{V}_{f,k}), G_f(\theta_k,\hat{\mu}_k,\hat{V}_{f,k},s_k,a_k,b_k,s_k') - \bar{G}_f(\theta_k,\omega_k,\hat{\mu}_k,\hat{V}_{f,k})\rangle$$

$$+ \langle\hat{V}_{f,k} - V_f^{\pi_{\theta_k},\phi_{\omega_k},\hat{\mu}_k} + \beta_k \bar{G}_f(\theta_k,\omega_k,\hat{\mu}_k,\hat{V}_{f,k}), V_f^{\pi_{\theta_k},\phi_{\omega_k},\hat{\mu}_k} - V_f^{\pi_{\theta_{k+1}},\phi_{\omega_{k+1}},\hat{\mu}_{k+1}}\rangle,$$

$$(65)$$

where the first inequality follows from the fact that the projection to a convex set is non-expansive.

To bound the first term of (65),

$$\left\|\hat{V}_{f,k} - V_f^{\pi_{\theta_k},\phi_{\omega_k},\hat{\mu}_k} + \beta_k \bar{G}_f(\theta_k,\omega_k,\hat{\mu}_k,\hat{V}_{f,k})\right\|^2$$

$$= \|\hat{V}_{f,k} - V_f^{\pi_{\theta_k},\phi_{\omega_k},\hat{\mu}_k}\|^2 + \beta_k^2\|\bar{G}_f(\theta_k,\omega_k,\hat{\mu}_k,\hat{V}_{f,k})\|^2 + 2\beta_k\langle\hat{V}_{f,k} - V_f^{\pi_{\theta_k},\phi_{\omega_k},\hat{\mu}_k}, \bar{G}_f(\theta_k,\omega_k,\hat{\mu}_k,\hat{V}_{f,k})\rangle$$

$$= \|\hat{V}_{f,k} - V_f^{\pi_{\theta_k},\phi_{\omega_k},\hat{\mu}_k}\|^2 + \beta_k^2\|\bar{G}_f(\theta_k,\omega_k,\hat{\mu}_k,\hat{V}_{f,k}) - \bar{G}_f(\theta_k,\omega_k,\hat{\mu}_k,V_f^{\pi_{\theta_k},\phi_{\omega_k},\hat{\mu}_k})\|^2$$

$$+ 2\beta_k\langle\hat{V}_{f,k} - V_f^{\pi_{\theta_k},\phi_{\omega_k},\hat{\mu}_k}, \bar{G}_f(\theta_k,\omega_k,\hat{\mu}_k,\hat{V}_{f,k}) - \bar{G}_f(\theta_k,\omega_k,\hat{\mu}_k,V_f^{\pi_{\theta_k},\phi_{\omega_k},\hat{\mu}_k})\rangle$$

$$= \|\hat{V}_{f,k} - V_f^{\pi_{\theta_k},\phi_{\omega_k},\hat{\mu}_k}\|^2 + \beta_k^2\|\bar{G}_f(\theta_k,\omega_k,\hat{\mu}_k,\hat{V}_{f,k}) - \bar{G}_f(\theta_k,\omega_k,\hat{\mu}_k,V_f^{\pi_{\theta_k},\phi_{\omega_k},\hat{\mu}_k})\|^2$$

$$+ 2\beta_k\left(\hat{V}_{f,k} - V_f^{\pi_{\theta_k},\phi_{\omega_k},\hat{\mu}_k}\right)^\top(\gamma P^{\pi_{\theta_k},\phi_{\omega_k},\hat{\mu}_k} - I)\left(\hat{V}_{f,k} - V_f^{\pi_{\theta_k},\phi_{\omega_k},\hat{\mu}_k}\right)$$

$$\leq \|\hat{V}_{f,k} - V_f^{\pi_{\theta_k},\phi_{\omega_k},\hat{\mu}_k}\|^2 + L_G^2\beta_k^2\|\hat{V}_{f,k} - V_f^{\pi_{\theta_k},\phi_{\omega_k},\hat{\mu}_k}\|^2 + 2(\gamma-1)\beta_k\|\hat{V}_{f,k} - V_f^{\pi_{\theta_k},\phi_{\omega_k},\hat{\mu}_k}\|^2$$

$$\leq (1 - (1-\gamma)\beta_k)\varepsilon_{f,k}^V,$$

$$(66)$$

where the last inequality follows from the step size condition $\beta_k \leq \frac{1-\gamma}{L_G^2}$.

For the third term of (65),

$$2\|V_f^{\pi_{\theta_k},\phi_{\omega_k},\hat{\mu}_k} - V_f^{\pi_{\theta_{k+1}},\phi_{\omega_{k+1}},\hat{\mu}_{k+1}}\|^2 \leq 6L_V^2\left(\|\pi_{\theta_{k+1}} - \pi_{\theta_k}\|^2 + \|\phi_{\omega_{k+1}} - \phi_{\omega_k}\|^2 + \|\hat{\mu}_{k+1} - \hat{\mu}_k\|^2\right)2$$

$$\leq 6L_V^2\left(B_F^2\alpha_k^2 + B_D^2\zeta_k^2 + B_H^2\xi_k^2\right)$$
$$\leq 6L_V^2(B_F + B_D + B_H)^2\alpha_k^2. \tag{67}$$

Note that the fourth term of (65) vanishes in expectation. Collecting the bounds from (66), (67), and Lemma 7, we have

$$\mathbb{E}[\varepsilon_{f,k+1}^V] \leq (1 - (1-\gamma)\beta_k)\mathbb{E}[\varepsilon_{f,k}^V] + 2\beta_k^2\mathbb{E}[\|G_f(\theta_k, \hat{\mu}_k, \hat{V}_{f,k}, s_k, a_k, b_k, s_k') - \bar{G}_f(\theta_k, \omega_k, \hat{\mu}_k, \hat{V}_{f,k})\|^2]$$
$$+ 6L_V^2(B_F + B_D + B_H)^2\alpha_k^2$$
$$+ \frac{(1-\gamma)\beta_k}{2}\mathbb{E}[\|\hat{V}_{f,k} - V_f^{\pi_{\theta_k}, \phi_{\omega_k}, \hat{\mu}_k} + \beta_k\bar{G}_f(\theta_k, \omega_k, \hat{\mu}_k, \hat{V}_{f,k})\|^2]$$
$$+ \frac{6L_V^2\alpha_k^2}{(1-\gamma)\beta_k}\mathbb{E}[\|\nabla_\theta J_f(\pi_{\theta_k}, \phi_{\omega_k}, \hat{\mu}_k)\|^2] + \frac{6L_V^2L_H^2\xi_k^2}{(1-\gamma)\beta_k}\mathbb{E}[\varepsilon_k^\pi] + \frac{6L_V^2\eta_2^2\zeta_k^2}{(1-\gamma)\beta_k}\mathbb{E}[\varepsilon_k^\phi]$$
$$+ \frac{16L^2L_V^2L_H^2\xi_k^2}{(1-\gamma)\beta_k}\mathbb{E}[\varepsilon_k^\mu] + \frac{6L_V^4\zeta_k^2}{(1-\gamma)\beta_k}\mathbb{E}[\varepsilon_{f,k}^V] + \frac{24L_V^2L_D^2\alpha_k^2}{(1-\gamma)\beta_k}\mathbb{E}[\varepsilon_{l,k}^V] + L_{VV}(B_F + B_D + B_H)^2\alpha_k^2$$
$$\leq (1 - \frac{(1-\gamma)\beta_k}{2} + \frac{6L_V^4\zeta_k^2}{(1-\gamma)\beta_k})\mathbb{E}[\varepsilon_{f,k}^V] + 8B_G^2\beta_k^2 + (6L_V^2 + L_{VV})(B_F + B_D + B_H)^2\alpha_k^2$$
$$+ \frac{6L_V^2\alpha_k^2}{(1-\gamma)\beta_k}\mathbb{E}[\|\nabla_\theta J_f(\pi_{\theta_k}, \phi_{\omega_k}, \hat{\mu}_k)\|^2] + \frac{6L_V^2L_H^2\xi_k^2}{(1-\gamma)\beta_k}\mathbb{E}[\varepsilon_k^\pi] + \frac{6L_V^2\eta_2^2\zeta_k^2}{(1-\gamma)\beta_k}\mathbb{E}[\varepsilon_k^\phi]$$
$$+ \frac{16L^2L_V^2L_H^2\xi_k^2}{(1-\gamma)\beta_k}\mathbb{E}[\varepsilon_k^\mu] + \frac{24L_V^2L_D^2\alpha_k^2}{(1-\gamma)\beta_k}\mathbb{E}[\varepsilon_{l,k}^V]$$
$$\leq (1 - \frac{(1-\gamma)\beta_k}{4})\mathbb{E}[\varepsilon_{f,k}^V] + \frac{6L_V^2\alpha_k^2}{(1-\gamma)\beta_k}\mathbb{E}[\|\nabla_\theta J_f(\pi_{\theta_k}, \phi_{\omega_k}, \hat{\mu}_k)\|^2] + \frac{6L_V^2L_H^2\xi_k^2}{(1-\gamma)\beta_k}\mathbb{E}[\varepsilon_k^\pi]$$
$$+ \frac{6L_V^2\eta_2^2\zeta_k^2}{(1-\gamma)\beta_k}\mathbb{E}[\varepsilon_k^\phi] + \frac{16L^2L_V^2L_H^2\xi_k^2}{(1-\gamma)\beta_k}\mathbb{E}[\varepsilon_k^\mu] + \frac{24L_V^2L_D^2\alpha_k^2}{(1-\gamma)\beta_k}\mathbb{E}[\varepsilon_{l,k}^V] + 12B_G^2\beta_k^2,$$

where the second inequality follows from $\|\hat{V}_{f,k} - V_f^{\pi_{\theta_k}, \phi_{\omega_k}, \hat{\mu}_k} + \beta_k\bar{G}_f(\theta_k, \omega_k, \hat{\mu}_k, \hat{V}_{f,k})\|^2 \leq \varepsilon_{f,k}^V$ established in (66), and the last inequality follows from the step size conditions $\frac{\zeta_k}{\beta_k} \leq \frac{1-\gamma}{2\sqrt{6}L_V^2}$ and $\frac{\alpha_k}{\beta_k} \leq \frac{2B_G}{\sqrt{6L_V^2 + L_{VV}}(B_F + B_D + B_H)}$.

The bound on $\mathbb{E}[\varepsilon_{l,k}^V]$ can be established using an identical argument, under the step size condition $\frac{\alpha_k}{\beta_k} \leq \frac{1-\gamma}{4\sqrt{6}L_V L_D}$.

∎

# D   Proof of Lemmas

## D.1   Proof of Lemma 1

We focus on proving the inequalities under the follower's reward and note that the same argument can be used for the leader's cumulative reward and value function.

We show the Lipschitz continuity conditions first. Fixing the mean field and leader's policy, the follower's problem reduces to a standard infinite-horizon discounted-reward MDP, in which it is well-known that the value function is Lipschitz and smooth with respect to the follower's policy parameter. Specifically, adapting Zeng et al. [2021][Lemma B.5], we have

$$\|V_f^{\pi_\theta, \phi_\omega, \mu} - V_f^{\pi_{\theta'}, \phi_{\omega'}, \mu}\| \leq \frac{2}{(1-\gamma)^2}(\|\theta - \theta'\| + \|\phi_\omega - \phi_{\omega'}\|). \tag{68}$$

This means that to show (31) it suffices for us to show the Lipschitz continuity of $V_f^{\pi_\theta, \phi_\omega, \mu}$ with respect to $\mu$. The value function can be expressed as

$$V_f^{\pi, \phi, \mu}(s) = \frac{1}{1-\gamma}\sum_{s', a, b} d_\rho^{\pi, \phi, \mu}(s')\pi(a \mid s')\phi(b \mid s')r_f(s, a, b, \mu)$$

$$= \frac{1}{1-\gamma} \langle d_\rho^{\pi,\phi,\mu}, r_f^{\pi,\phi,\mu} \rangle$$

$$= e_s^\top (I - \gamma P^{\pi,\phi,\mu})^{-\top} r_f^{\pi,\phi,\mu}, \tag{69}$$

where $e_s \in \mathbb{R}^{|\mathcal{S}|}$ denotes a vector with 1 at entry $s$ and zero otherwise, and $r_f^{\pi,\phi,\mu} \in \mathbb{R}^{|\mathcal{S}|}$ denotes the marginalized reward such that

$$r_f^{\pi,\phi,\mu}(s) = \sum_{a,b} r_f(s,a,b,\mu) \pi(a \mid s) \phi(b \mid s).$$

We have from (69)

$$\|V_f^{\pi,\phi,\mu} - V_f^{\pi,\phi,\mu'}\|$$

$$= \left( (I - \gamma P^{\pi,\phi,\mu})^{-\top} r_f^{\pi,\phi,\mu} - (I - \gamma P^{\pi,\phi,\mu'})^{-\top} r_f^{\pi,\phi,\mu'} \right)$$

$$= \left( (I - \gamma P^{\pi,\phi,\mu})^{-\top} - (I - \gamma P^{\pi,\phi,\mu'})^{-\top} \right) r_f^{\pi,\phi,\mu} + (I - \gamma P^{\pi,\phi,\mu'})^{-\top} \left( r_f^{\pi,\phi,\mu} - r_f^{\pi,\phi,\mu'} \right)$$

$$= (I - \gamma P^{\pi,\phi,\mu})^{-\top} \left( (I - \gamma P^{\pi,\phi,\mu'})^\top - (I - \gamma P^{\pi,\phi,\mu})^\top \right) (I - \gamma P^{\pi,\phi,\mu'})^{-\top} r_f^{\pi,\phi,\mu}$$

$$\quad + (I - \gamma P^{\pi,\phi,\mu'})^{-\top} \left( r_f^{\pi,\phi,\mu} - r_f^{\pi,\phi,\mu'} \right)$$

$$\leq \|(I - \gamma P^{\pi,\phi,\mu})^{-\top}\| \left\| (I - \gamma P^{\pi,\phi,\mu'})^\top - (I - \gamma P^{\pi,\phi,\mu})^\top \right\| \|(I - \gamma P^{\pi,\phi,\mu'})^{-\top}\| \|r_f^{\pi,\phi,\mu}\|$$

$$\quad + \|(I - \gamma P^{\pi,\phi,\mu'})^{-\top}\| \left\| r_f^{\pi,\phi,\mu} - r_f^{\pi,\phi,\mu'} \right\|$$

$$\leq \frac{1}{1-\gamma} \cdot \gamma \|P^{\pi,\phi,\mu} - P^{\pi,\phi,\mu'}\| \cdot \frac{1}{1-\gamma} \cdot \sqrt{|\mathcal{S}|} + \frac{1}{1-\gamma} \left\| r_f^{\pi,\phi,\mu} - r_f^{\pi,\phi,\mu'} \right\|$$

$$\leq \frac{L_P \gamma \sqrt{|\mathcal{S}|}}{(1-\gamma)^2} \|\mu - \mu'\| + \frac{L_r}{1-\gamma} \|\mu - \mu'\|$$

$$\leq \left( \frac{L_P \gamma \sqrt{|\mathcal{S}|}}{(1-\gamma)^2} + \frac{L_r}{1-\gamma} \right) \|\mu - \mu'\|, \tag{70}$$

where the second last inequality follows from Assumption 1.

Combining (68) and (70) implies (29), which obviously leads to (28) as $J_f(\pi,\phi,\mu)$ is simply $\langle V_f^{\pi,\phi,\mu}, \rho \rangle$.

To show the smoothness conditions, we note that again adapting Zeng et al. [2021][Lemma B.5], we have

$$\|\nabla_\theta V_f^{\pi_\theta,\phi_\omega,\mu} - \nabla_\theta V_f^{\pi_{\theta'},\phi_\omega,\mu}\| \leq \frac{8}{(1-\gamma)^3} \|\theta - \theta'\|, \quad \|\nabla_\omega V_f^{\pi_\theta,\phi_\omega,\mu} - \nabla_\omega V_f^{\pi_\theta,\phi_{\omega'},\mu}\| \leq \frac{8}{(1-\gamma)^3} \|\omega - \omega'\|,$$

This implies (30).

To show (31), we differentiate $V_f^{\pi,\phi,\mu}$ with respect to $\mu$. It can be seen from (69) that

$$\nabla_\mu V_f^{\pi,\phi,\mu} = \gamma \left( I - \gamma P^{\pi,\phi,\mu} \right)^{-\top} \left( \nabla_\mu P^{\pi,\phi,\mu} \right)^\top \left( I - \gamma P^{\pi,\phi,\mu} \right)^{-\top} r_f^{\pi,\phi,\mu} + \left( I - \gamma P^{\pi,\phi,\mu} \right)^{-\top} \nabla_\mu r_f^{\pi,\phi,\mu}.$$

Therefore, we have

$$\|\nabla_\mu V_f^{\pi,\phi,\mu} - \nabla_\mu V_f^{\pi,\phi,\mu'}\|$$

$$\leq \gamma \| \left( I - \gamma P^{\pi,\phi,\mu} \right)^{-\top} \left( \nabla_\mu P^{\pi,\phi,\mu} \right)^\top \left( I - \gamma P^{\pi,\phi,\mu} \right)^{-\top} r_f^{\pi,\phi,\mu} - \left( I - \gamma P^{\pi,\phi,\mu'} \right)^{-\top} \left( \nabla_\mu P^{\pi,\phi,\mu'} \right)^\top \left( I - \gamma P^{\pi,\phi,\mu'} \right)^{-\top} r_f^{\pi,\phi,\mu'} \|$$

$$\quad + \gamma \| \left( I - \gamma P^{\pi,\phi,\mu} \right)^{-\top} \nabla_\mu r_f^{\pi,\phi,\mu} - \left( I - \gamma P^{\pi,\phi,\mu'} \right)^{-\top} \nabla_\mu r_f^{\pi,\phi,\mu'} \|$$

$$\leq \gamma \| \left( I - \gamma P^{\pi,\phi,\mu} \right)^{-\top} - \left( I - \gamma P^{\pi,\phi,\mu'} \right)^{-\top} \| \cdot L_P \cdot \frac{1}{1-\gamma} \cdot \sqrt{|\mathcal{S}|}$$

$$+ \gamma \frac{1}{1-\gamma} \cdot \| \left(\nabla_\mu P^{\pi,\phi,\mu}\right)^\top - \left(\nabla_\mu P^{\pi,\phi,\mu'}\right)^\top \| \cdot \frac{1}{1-\gamma} \cdot \sqrt{|\mathcal{S}|}$$

$$+ \gamma \frac{1}{1-\gamma} \cdot L_P \cdot \| \left(I - \gamma P^{\pi,\phi,\mu}\right)^{-\top} - \left(I - \gamma P^{\pi,\phi,\mu'}\right)^{-\top} \| \cdot \sqrt{|\mathcal{S}|}$$

$$+ \gamma \frac{1}{1-\gamma} \cdot L_P \cdot \frac{1}{1-\gamma} \cdot \|\nabla_\mu r_f^{\pi,\phi,\mu} - \nabla_\mu r_f^{\pi,\phi,\mu'}\|$$

$$+ \gamma \| \left(\nabla_\mu P^{\pi,\phi,\mu}\right)^\top - \left(\nabla_\mu P^{\pi,\phi,\mu'}\right)^\top \| \cdot L_r$$

$$+ \gamma L_P \cdot \|\nabla_\mu r_f^{\pi,\phi,\mu} - \nabla_\mu r_f^{\pi,\phi,\mu'}\|$$

$$\leq \frac{2\gamma L_P \sqrt{|\mathcal{S}|}}{1-\gamma} \| \left(I - \gamma P^{\pi,\phi,\mu}\right)^{-\top} - \left(I - \gamma P^{\pi,\phi,\mu'}\right)^{-\top} \| + \left( \frac{\gamma\sqrt{|\mathcal{S}|}}{(1-\gamma)^2} + \gamma L_r \right) \|\nabla_\mu P^{\pi,\phi,\mu} - \nabla_\mu P^{\pi,\phi,\mu'}\|$$

$$+ \left( \frac{\gamma L_P}{(1-\gamma)^2} + \gamma L_P \right) \|\nabla_\mu r_f^{\pi,\phi,\mu} - \nabla_\mu r_f^{\pi,\phi,\mu'}\|$$

$$\leq \frac{2\gamma L_P \sqrt{|\mathcal{S}|}}{1-\gamma} \| \left(I - \gamma P^{\pi,\phi,\mu}\right)^{-\top} \|\| \left(I - \gamma P^{\pi,\phi,\mu'}\right)^\top - \left(I - \gamma P^{\pi,\phi,\mu}\right)^\top \|\| \left(I - \gamma P^{\pi,\phi,\mu'}\right)^{-\top} \|$$

$$+ \left( \frac{\gamma\sqrt{|\mathcal{S}|}}{(1-\gamma)^2} + \gamma L_r \right) L_P \|\mu - \mu'\| + \left( \frac{\gamma L_P}{(1-\gamma)^2} + \gamma L_P \right) L_r \|\mu - \mu'\|$$

$$\leq \frac{2\gamma L_P \sqrt{|\mathcal{S}|}}{(1-\gamma)^3} \cdot \gamma \|\nabla_\mu P^{\pi,\phi,\mu} - \nabla_\mu P^{\pi,\phi,\mu'}\| + 2\left( \frac{\gamma\sqrt{|\mathcal{S}|}L_P}{(1-\gamma)^2} + \gamma L_P L_r \right) \|\mu - \mu'\|$$

$$\leq \left( \frac{2\gamma^2 \sqrt{|\mathcal{S}|}L_P^2}{(1-\gamma)^3} + \frac{2\gamma \sqrt{|\mathcal{S}|}L_P}{(1-\gamma)^2} + 2\gamma L_P L_r \right) \|\mu - \mu'\|,$$

where we have plugged in the smoothness bounds on $P$ and $r_f$ from Assumption 1.

∎

### D.2  Proof of Lemma 2

Note that the discounted occupancy measure can be expressed as

$$d_\rho^{\pi,\phi,\mu} = (1-\gamma)(I - \gamma P^{\pi,\phi,\mu})^{-1}\rho. \tag{71}$$

This implies

$$\left\| \sum_s \left( d_\rho^{\pi_{\theta_1},\phi_{\omega_1}\mu_1}(s) P^{\pi_{\theta_1},\phi_{\omega_1}\mu_1}(\cdot \mid s) - d_\rho^{\pi_{\theta_2},\phi_{\omega_2}\mu_2}(s) P^{\pi_{\theta_2},\phi_{\omega_2}\mu_2}(\cdot \mid s) \right) \right\|$$

$$= (1-\gamma) \| P^{\pi_{\theta_1},\phi_{\omega_1}\mu_1}(I - \gamma P^{\pi_{\theta_1},\phi_{\omega_1}\mu_1})^{-1}\rho - P^{\pi_{\theta_2},\phi_{\omega_2}\mu_2}(I - \gamma P^{\pi_{\theta_2},\phi_{\omega_2}\mu_2})^{-1}\rho \|$$

$$\leq (1-\gamma)\|\rho\|\| P^{\pi_{\theta_1},\phi_{\omega_1}\mu_1} - P^{\pi_{\theta_2},\phi_{\omega_2}\mu_2} \|\|(I - \gamma P^{\pi_{\theta_1},\phi_{\omega_1}\mu_1})^{-1}\|$$

$$+ (1-\gamma)\|\rho\|\| P^{\pi_{\theta_2},\phi_{\omega_2}\mu_2} \|\|(I - \gamma P^{\pi_{\theta_1},\phi_{\omega_1}\mu_1})^{-1} - (I - \gamma P^{\pi_{\theta_2},\phi_{\omega_2}\mu_2})^{-1}\|$$

$$\leq (1-\gamma)\left( L_P\|\mu_1 - \mu_2\| + \|\pi_{\theta_1} - \pi_{\theta_2}\| + \|\phi_{\omega_1} - \phi_{\omega_2}\| \right) \cdot \frac{1}{1-\gamma}$$

$$+ (1-\gamma)\|(I - \gamma P^{\pi_{\theta_2},\phi_{\omega_2}\mu_2})^{-1}\|\|(I - \gamma P^{\pi_{\theta_2},\phi_{\omega_2}\mu_2}) - (I - \gamma P^{\pi_{\theta_1},\phi_{\omega_1}\mu_1})\|\|(I - \gamma P^{\pi_{\theta_1},\phi_{\omega_1}\mu_1})^{-1}\|$$

$$\leq \left( L_P\|\mu_1 - \mu_2\| + \|\pi_{\theta_1} - \pi_{\theta_2}\| + \|\phi_{\omega_1} - \phi_{\omega_2}\| \right) + \frac{\gamma}{1-\gamma} \| P^{\pi_{\theta_1},\phi_{\omega_1}\mu_1} - P^{\pi_{\theta_2},\phi_{\omega_2}\mu_2} \|$$

$$\leq \frac{1}{1-\gamma}\left( L_P\|\mu_1 - \mu_2\| + \|\pi_{\theta_1} - \pi_{\theta_2}\| + \|\phi_{\omega_1} - \phi_{\omega_2}\| \right)$$

$$\leq \frac{1}{1-\gamma}\left( L_P\|\mu_1 - \mu_2\| + \|\theta_1 - \theta_2\| + \|\omega_1 - \omega_2\| \right),$$

where the second inequality follows from Assumption 1 and the fact that the state transition matrix is 1-Lipschitz in the policies, and the final inequality is due to the 1-Lipschitz continuity of the softmax function.

∎

### D.3 Proof of Lemma 3

Note that $\nabla_\theta \log \pi_\theta(a \mid s)$ can be expressed entry-wise as

$$\frac{\partial \log \pi_\theta(a \mid s)}{\partial \theta_{s',a'}} = \mathbf{1}\left[s = s'\right]\left(\mathbf{1}\left[a = a'\right] - \pi_\theta\left(a' \mid s\right)\right), \tag{72}$$

which implies

$$\|\nabla_\theta \log \pi_\theta(a \mid s)\|_2 \le \|\nabla_\theta \log \pi_\theta(a \mid s)\|_1 \le 1 + 1 = 2. \tag{73}$$

By the definition of $F$,

$$\|F(\theta, \mu, V_f, s, a, b, s')\| \le \|\nabla_\theta \log \pi_\theta(a \mid s)\|\left|r_f(s, a, b, \mu) + \tau E(\pi_\theta, s) + \gamma V_f(s') - V_f(s)\right|$$
$$\le 2(1 + \tau \log|\mathcal{A}| + \gamma B_V + B_V)$$
$$= 2(1 + \gamma)B_V + 2\tau \log|\mathcal{A}| + 2,$$

where the second inequality applies (73).

By a similar argument, we can show

$$\|D(\omega, \mu, V_l, s, b, s')\| \le 2(1 + \gamma)B_V + 2.$$

For $H$, we have

$$\|H(\mu, s)\| = \|e_s - \mu\| \le \|e_s\| + \|\mu\| \le 2.$$

For the $G$ operators,

$$\|G_f(\theta, \mu, V_f, s, a, b, s')\| \le \|e_s\|\left|r_f(s, a, b, \mu) + \tau E(\pi_\theta, s) + \gamma V_f(s') - V_f(s)\right|$$
$$\le 1 \cdot (1 + \tau \log|\mathcal{A}| + \gamma B_V + B_V)$$
$$= (1 + \gamma)B_V + \tau \log|\mathcal{A}| + 1.$$

Similarly,

$$\|G_l(\omega, \mu, V_l, s, b, s')\| \le (1 + \gamma)B_V + 1.$$

$\blacksquare$

### D.4 Proof of Lemma 4

By the definition of $\bar{F}$,

$$\|\bar{F}(\theta_1, \omega_1, \mu_1, V_1) - \bar{F}(\theta_2, \omega_2, \mu_2, V_2)\|$$
$$= \|\mathbb{E}_{s \sim d_\rho^{\pi_{\theta_1}, \phi_{\omega_1}, \mu_1}, a \sim \pi_{\theta_1}(\cdot|s), b \sim \phi_{\omega_1}(\cdot|s), s' \sim \mathcal{P}^{\mu_1}(\cdot|s,a,b)}[F(\theta_1, \mu_1, V_1, s, a, b, s')]$$
$$\quad - \mathbb{E}_{s \sim d_\rho^{\pi_{\theta_2}, \phi_{\omega_2}, \mu_2}, a \sim \pi_{\theta_2}(\cdot|s), b \sim \phi_{\omega_2}(\cdot|s), s' \sim \mathcal{P}^{\mu_2}(\cdot|s,a,b)}[F(\theta_2, \mu_2, V_2, s, a, b, s')]\|$$
$$= \Big\| \sum_{s,a,b,s'} \Big(d_\rho^{\pi_{\theta_1}, \phi_{\omega_1} \mu_1}(s)\pi_{\theta_1}(a \mid s)\phi_{\omega_1}(b \mid s)\mathcal{P}^{\mu_1}(s' \mid s, a, b)$$
$$\quad - d_\rho^{\pi_{\theta_2}, \phi_{\omega_2} \mu_2}(s)\pi_{\theta_2}(a \mid s)\phi_{\omega_2}(b \mid s)\mathcal{P}^{\mu_2}(s' \mid s, a)\Big)F(\theta_2, \mu_2, V_2, s, a, b, s')$$
$$\quad + \mathbb{E}_{s \sim d_\rho^{\pi_{\theta_1}, \phi_{\omega_1}, \mu_1}, a \sim \pi_{\theta_1}(\cdot|s), b \sim \phi_{\omega_1}(\cdot|s), s' \sim \mathcal{P}^{\mu_1}(\cdot|s,a,b)}[F(\theta_1, \mu_1, V_1, s, a, b, s') - F(\theta_2, \mu_2, V_2, s, a, b, s')]\Big\|$$
$$\le \Big\|\mathbb{E}_{s \sim d_\rho^{\pi_{\theta_1}, \phi_{\omega_1}, \mu_1}, a \sim \pi_{\theta_1}(\cdot|s), b \sim \phi_{\omega_1}(\cdot|s), s' \sim \mathcal{P}^{\mu_1}(\cdot|s,a,b)}[F(\theta_1, \mu_1, V_1, s, a, b, s') - F(\theta_2, \mu_2, V_2, s, a, b, s')]\Big\|$$
$$\quad + B_F\Big|\sum_{s,a,b,s'}\big(d_\rho^{\pi_{\theta_1}, \phi_{\omega_1} \mu_1}(s)\pi_{\theta_1}(a \mid s)\phi_{\omega_1}(b \mid s)\mathcal{P}^{\mu_1}(s' \mid s, a, b) - d_\rho^{\pi_{\theta_2}, \phi_{\omega_2} \mu_2}(s)\pi_{\theta_2}(a \mid s)\phi_{\omega_2}(b \mid s)\mathcal{P}^{\mu_2}(s' \mid s, a)\big)\Big|$$
$$\le \Big\|\mathbb{E}_{s \sim d_\rho^{\pi_{\theta_1}, \phi_{\omega_1}, \mu_1}, a \sim \pi_{\theta_1}(\cdot|s), b \sim \phi_{\omega_1}(\cdot|s), s' \sim \mathcal{P}^{\mu_1}(\cdot|s,a,b)}[F(\theta_1, \mu_1, V_1, s, a, b, s') - F(\theta_2, \mu_2, V_2, s, a, b, s')]\Big\|$$

$$+ B_F \Big| \sum_{s,s'} \big(d_\rho^{\pi_{\theta_1}, \phi_{\omega_1} \mu_1}(s) P^{\pi_{\theta_1}, \phi_{\omega_1} \mu_1}(s' \mid s) - d_\rho^{\pi_{\theta_2}, \phi_{\omega_2} \mu_2}(s) P^{\pi_{\theta_2}, \phi_{\omega_2} \mu_2}(s' \mid s)\big) \Big|. \quad (74)$$

To bound the first term of (74),

$$\|F(\theta_1, \mu_1, V_1, s, a, b, s') - F(\theta_2, \mu_2, V_2, s, a, b, s')\|$$
$$= \|\nabla_\theta \log \pi_{\theta_1}(a \mid s)(r_f(s, a, b, \mu_1) + \tau E(\pi_{\theta_1}, s) + \gamma V_1(s') - V_1(s))$$
$$- \nabla_\theta \log \pi_{\theta_2}(a \mid s)(r_f(s, a, b, \mu_2) + \tau E(\pi_{\theta_2}, s) + \gamma V_2(s') - V_2(s))\|$$
$$\leq \|\nabla_\theta \log \pi_{\theta_1}(a \mid s) - \nabla_\theta \log \pi_{\theta_2}(a \mid s)\| \Big| r_f(s, a, b, \mu_1) + \tau E(\pi_{\theta_1}, s) + \gamma V_1(s') - V_1(s) \Big|$$
$$+ \|\nabla_\theta \log \pi_{\theta_2}(a \mid s)\| \Big| r_f(s, a, b, \mu_1) - r_f(s, a, b, \mu_2) + \tau E(\pi_{\theta_1}, s) - \tau E(\pi_{\theta_2}, s)$$
$$+ \gamma V_1(s') - \gamma V_2(s') - V_1(s) + V_2(s)) \Big|$$
$$\leq \|\theta_1 - \theta_2\|(1 + \tau \log |\mathcal{A}| + (1 + \gamma) B_V)$$
$$+ 2 \Big( L_r |\mu_1 - \mu_2| + \frac{(4 + 8 \log |\mathcal{A}|)\tau}{(1 - \gamma)^3} \|\theta_1 - \theta_2\| + (1 + \gamma)\|V_1 - V_2\| \Big)$$
$$\leq \Big( 1 + \tau \log |\mathcal{A}| + (1 + \gamma) B_V + \frac{(4 + 8 \log |\mathcal{A}|)\tau}{(1 - \gamma)^3} \Big) \|\theta_1 - \theta_2\| + 2 L_r \|\mu_1 - \mu_2\| + 2(1 + \gamma)\|V_1 - V_2\|,$$
$$(75)$$

where the second inequality is due to Lipschitz continuity of the entropy function (see Zeng et al. [2022][Lemma 6]) and the boundedness and 1-Lipschitz continuity of $\nabla_\theta \log \pi_\theta$, which is obvious from (72) and (73).

By Lemma 2, we have for the second term of (74),

$$B_F \Big| \sum_{s,s'} \big(d_\rho^{\pi_{\theta_1}, \phi_{\omega_1} \mu_1}(s) P^{\pi_{\theta_1}, \phi_{\omega_1} \mu_1}(s' \mid s) - d_\rho^{\pi_{\theta_2}, \phi_{\omega_2} \mu_2}(s) P^{\pi_{\theta_2}, \phi_{\omega_2} \mu_2}(s' \mid s)\big) \Big|$$
$$\leq B_F \Big\| \sum_s \big(d_\rho^{\pi_{\theta_1}, \phi_{\omega_1} \mu_1}(s) P^{\pi_{\theta_1}, \phi_{\omega_1} \mu_1}(\cdot \mid s) - d_\rho^{\pi_{\theta_2}, \phi_{\omega_2} \mu_2}(s) P^{\pi_{\theta_2}, \phi_{\omega_2} \mu_2}(\cdot \mid s)\big) \Big\|$$
$$\leq \frac{B_F}{1 - \gamma} \Big( L_P \|\mu_1 - \mu_2\| + \|\theta_1 - \theta_2\| + \|\omega_1 - \omega_2\| \Big). \quad (76)$$

Substituting (75) and (76) into (74), we have

$$\|\bar{F}(\theta_1, \omega_1, \mu_1, V_1) - \bar{F}(\theta_2, \omega_2, \mu_2, V_2)\|$$
$$\leq \Big( 1 + \tau \log |\mathcal{A}| + (1 + \gamma) B_V + \frac{(4 + 8 \log |\mathcal{A}|)\tau}{(1 - \gamma)^3} \Big) \|\theta_1 - \theta_2\| + 2 L_r \|\mu_1 - \mu_2\| + 2(1 + \gamma)\|V_1 - V_2\|$$
$$+ \frac{B_F}{1 - \gamma} \Big( L_P \|\mu_1 - \mu_2\| + \|\theta_1 - \theta_2\| + \|\omega_1 - \omega_2\| \Big)$$
$$= \Big( 1 + \tau \log |\mathcal{A}| + (1 + \gamma) B_V + \frac{(4 + 8 \log |\mathcal{A}|)\tau}{(1 - \gamma)^3} + \frac{B_F}{1 - \gamma} \Big) \|\theta_1 - \theta_2\|$$
$$+ \Big( 2 L_r + \frac{B_F L_P}{1 - \gamma} \Big) \|\mu_1 - \mu_2\| + \frac{B_F}{1 - \gamma} \|\omega_1 - \omega_2\| + 4\|V_1 - V_2\|.$$

This obviously leads to the claimed result by setting $L_F = \max \Big\{ 1 + \tau \log |\mathcal{A}| + (1 + \gamma) B_V + \frac{(4 + 8 \log |\mathcal{A}|)\tau}{(1 - \gamma)^3} + \frac{B_F}{1 - \gamma}, 2 L_r + \frac{B_F L_P}{1 - \gamma}, \frac{B_F}{1 - \gamma}, 4 \Big\}$.

The Lipschitz continuity of $\bar{D}$ can be shown using almost identical steps, which we will skip.

For $\bar{H}$, we can adopt a similar argument

$$\|\bar{H}(\pi_1, \omega_1, \mu_1) - \bar{H}(\pi_2, \omega_2, \mu_2)\|$$
$$\leq \|\mathbb{E}_{\bar{s} \sim \nu^{\pi_1, \phi_{\omega_1}, \mu_1}}[H(\mu_1, \bar{s}) - H(\mu_2, \bar{s})]\| + B_H \Big| \sum_s \big(d_\rho^{\pi_{\theta_1}, \phi_{\omega_1} \mu_1}(s) - d_\rho^{\pi_{\theta_2}, \phi_{\omega_2} \mu_2}(s)\big) \Big|$$

$$\leq \|\mu_1 - \mu_2\| + \left\| \sum_s \left( d_\rho^{\pi_{\theta_1}, \phi_{\omega_1} \mu_1}(s) P^{\pi_{\theta_1}, \phi_{\omega_1} \mu_1}(\cdot \mid s) - d_\rho^{\pi_{\theta_2}, \phi_{\omega_2} \mu_2}(s) P^{\pi_{\theta_2}, \phi_{\omega_2} \mu_2}(\cdot \mid s) \right) \right\|$$

$$\leq \|\mu_1 - \mu_2\| + \frac{1}{1-\gamma} \left( L_P \|\mu_1 - \mu_2\| + \|\theta_1 - \theta_2\| + \|\omega_1 - \omega_2\| \right)$$

$$\leq L_H(\|\theta_1 - \theta_2\| + \|\omega_1 - \omega_2\| + \|\mu_1 - \mu_2\|),$$

with $L_H = \max\{\frac{L_P}{1-\gamma}, 1\}$.

For $\bar{G}_f$, we again split $\|\bar{G}_f(\theta_1, \omega_1, \mu_1, V_1) - \bar{G}_f(\theta_2, \omega_2, \mu_2, V_2)\|$ into two parts and invoke Lemma 2

$$\|\bar{G}_f(\theta_1, \omega_1, \mu_1, V_1) - \bar{G}_f(\theta_2, \omega_2, \mu_2, V_2)\|$$

$$\leq \left\| \mathbb{E}_{s \sim d_\rho^{\pi_{\theta_1}, \phi_{\omega_1} \mu_1}, a \sim \pi_{\theta_1}(\cdot|s), b \sim \phi_{\omega_1}(\cdot|s), s' \sim \mathcal{P}^{\mu_1}(\cdot|s,a,b)} [G_f(\theta_1, \mu_1, V_1, s, a, b, s') - G_f(\theta_2, \mu_2, V_2, s, a, b, s')] \right\|$$

$$+ B_G \left| \sum_{s,s'} \left( d_\rho^{\pi_{\theta_1}, \phi_{\omega_1} \mu_1}(s) P^{\pi_{\theta_1}, \phi_{\omega_1} \mu_1}(s' \mid s) - d_\rho^{\pi_{\theta_2}, \phi_{\omega_2} \mu_2}(s) P^{\pi_{\theta_2}, \phi_{\omega_2} \mu_2}(s' \mid s) \right) \right|$$

$$\leq \left| r_f(s, a, b, \mu_1) - r_f(s, a, b, \mu_2) + \tau E(\pi_{\theta_1}, s) - \tau E(\pi_{\theta_2}, s) + \gamma V_1(s') - \gamma V_2(s') - V_1(s) + V_2(s)) \right|$$

$$+ \frac{B_G}{1-\gamma} \left( L_P \|\mu_1 - \mu_2\| + \|\theta_1 - \theta_2\| + \|\omega_1 - \omega_2\| \right)$$

$$\leq \left( \frac{(4 + 8 \log |\mathcal{A}|)\tau}{(1-\gamma)^3} + \frac{B_G}{1-\gamma} \right) \|\theta_1 - \theta_2\| + (L_r + \frac{B_G L_P}{1-\gamma}) \|\mu_1 - \mu_2\| + \frac{B_G}{1-\gamma} \|\omega_1 - \omega_2\| + 2\|V_1 - V_2\|.$$

Similarly,

$$\|\bar{G}_l(\omega_1, \mu_1, V_1) - \bar{G}_l(\omega_2, \mu_2, V_2)\|$$

$$\leq \left\| \mathbb{E}_{s \sim d_\rho^{\pi_{\theta_1}, \phi_{\omega_1} \mu_1}, a \sim \pi_{\theta_1}(\cdot|s), b \sim \phi_{\omega_1}(\cdot|s), s' \sim \mathcal{P}^{\mu_1}(\cdot|s,a,b)} [G_l(\mu_1, V_1, s, a, b, s') - G_l(\mu_2, V_2, s, b, s')] \right\|$$

$$+ B_G \left| \sum_{s,s'} \left( d_\rho^{\pi_{\theta_1}, \phi_{\omega_1} \mu_1}(s) P^{\pi_{\theta_1}, \phi_{\omega_1} \mu_1}(s' \mid s) - d_\rho^{\pi_{\theta_2}, \phi_{\omega_2} \mu_2}(s) P^{\pi_{\theta_2}, \phi_{\omega_2} \mu_2}(s' \mid s) \right) \right|$$

$$\leq \left| r_l(s, b, \mu_1) - r_l(s, b, \mu_2) + \gamma V_1(s') - \gamma V_2(s') - V_1(s) + V_2(s)) \right|$$

$$+ \frac{B_G}{1-\gamma} \left( L_P \|\mu_1 - \mu_2\| + \|\theta_1 - \theta_2\| + \|\omega_1 - \omega_2\| \right)$$

$$\leq \frac{B_G}{1-\gamma} \|\theta_1 - \theta_2\| + (L_r + \frac{B_G L_P}{1-\gamma}) \|\mu_1 - \mu_2\| + \frac{B_G}{1-\gamma} \|\omega_1 - \omega_2\| + 2\|V_1 - V_2\|.$$

$\blacksquare$

### D.5 Proof of Lemma 5

Within the proof of Lemma 5, we use the shorthand notation

$$y_k = \hat{\mu}_k - \mu^\star(\phi_{\omega_k}) + \xi_k \bar{H}(\pi^\star(\phi_{\omega_k}, \hat{\mu}_k), \omega_k, \hat{\mu}_k).$$

As the operator $\mu^\star$ is Lipschitz, we know from the mean-value theorem that there exists $\omega_{k+1}^z = z\omega_k + (1-z)\omega_{k+1}$ for some scalar $z \in [0, 1]$ such that

$$\mu^\star(\phi_{\omega_{k+1}}) - \mu^\star(\phi_{\omega_k}) = \nabla_\omega \mu^\star(\phi_{\omega_{k+1}^z})^\top (\omega_{k+1} - \omega_k)$$

$$= \zeta_k \nabla_\omega \mu^\star(\phi_{\omega_{k+1}^z})^\top D(\omega_k, \hat{\mu}_k, \hat{V}_{l,k}, s_k, b_k, s_k')$$

$$= \zeta_k \nabla_\omega \mu^\star(\phi_{\omega_{k+1}^z})^\top \bar{D}(\omega_k, \hat{\mu}_k, \hat{V}_{l,k})$$

$$+ \zeta_k \nabla_\omega \mu^\star(\phi_{\omega_{k+1}^z})^\top \left( D(\omega_k, \hat{\mu}_k, \hat{V}_{l,k}, s_k, b_k, s_k') - \bar{D}(\omega_k, \hat{\mu}_k, \hat{V}_{l,k}) \right).$$

This implies

$$\mathbb{E}[\langle \hat{\mu}_k - \mu^\star(\phi_{\omega_k}) + \xi_k \bar{H}(\pi^\star(\phi_{\omega_k}, \hat{\mu}_k), \omega_k, \hat{\mu}_k), \mu^\star(\phi_{\omega_{k+1}}) - \mu^\star(\phi_{\omega_k}) \rangle]$$

$$
\begin{aligned}
&= \zeta_k \mathbb{E}[\langle y_k, \nabla_\omega \mu^\star(\phi_{\omega_{k+1}^z})^\top \bar{D}(\omega_k, \hat{\mu}_k, \hat{V}_{l,k})\rangle] \\
&\quad + \zeta_k \mathbb{E}[\langle y_k, \nabla_\omega \mu^\star(\phi_{\omega_{k+1}^z})^\top \Big(D(\omega_k, \hat{\mu}_k, \hat{V}_{l,k}, s_k, b_k, s_k') - \bar{D}(\omega_k, \hat{\mu}_k, \hat{V}_{l,k})\Big)\rangle] \\
&= \zeta_k \mathbb{E}[\langle y_k, \nabla_\omega \mu^\star(\phi_{\omega_{k+1}^z})^\top \bar{D}(\omega_k, \hat{\mu}_k, \hat{V}_{l,k})\rangle] \\
&\quad + \zeta_k \mathbb{E}[\langle y_k, \Big(\nabla_\omega \mu^\star(\phi_{\omega_{k+1}^z}) - \nabla_\omega \mu^\star(\phi_{\omega_k})\Big)^\top \Big(D(\omega_k, \hat{\mu}_k, \hat{V}_{l,k}, s_k, b_k, s_k') - \bar{D}(\omega_k, \hat{\mu}_k, \hat{V}_{l,k})\Big)\rangle] \\
&\quad + \zeta_k \mathbb{E}[\langle y_k, \nabla_\omega \mu^\star(\phi_{\omega_k})^\top \mathbb{E}[D(\omega_k, \hat{\mu}_k, \hat{V}_{l,k}, s_k, b_k, s_k') - \bar{D}(\omega_k, \hat{\mu}_k, \hat{V}_{l,k}) \mid \mathcal{F}_{k-1}]\rangle] \\
&= \zeta_k \mathbb{E}[\langle y_k, \nabla_\omega \mu^\star(\phi_{\omega_{k+1}^z})^\top \bar{D}(\omega_k, \hat{\mu}_k, \hat{V}_{l,k})\rangle] \\
&\quad + \zeta_k \mathbb{E}[\langle y_k, \Big(\nabla_\omega \mu^\star(\phi_{\omega_{k+1}^z}) - \nabla_\omega \mu^\star(\phi_{\omega_k})\Big)^\top \Big(D(\omega_k, \hat{\mu}_k, \hat{V}_{l,k}, s_k, b_k, s_k') - \bar{D}(\omega_k, \hat{\mu}_k, \hat{V}_{l,k})\Big)\rangle].
\end{aligned}
\tag{77}
$$

To bound the first term of (77), note that by Assumption 2 and the fact that the softmax operator is 1-Lipschitz

$$
\begin{aligned}
&\zeta_k \langle y_k, \nabla_\omega \mu^\star(\phi_{\omega_{k+1}^z})^\top \bar{D}(\omega_k, \hat{\mu}_k, \hat{V}_{l,k})\rangle \\
&\leq L\zeta_k \|y_k\| \|\bar{D}(\omega_k, \hat{\mu}_k, \hat{V}_{l,k})\| \\
&\leq \frac{(1-\delta)\xi_k}{16} \|y_k\|^2 + \frac{4L^2\zeta_k^2}{(1-\delta)\xi_k} \|\bar{D}(\omega_k, \hat{\mu}_k, \hat{V}_{l,k})\|^2 \\
&\leq \frac{(1-\delta)\xi_k}{16} \|y_k\|^2 + \frac{8L^2\zeta_k^2}{(1-\delta)\xi_k} \|\bar{D}(\omega_k, \mu^\star(\phi_{\omega_k}), V_l^{\phi_{\omega_k},\mu^\star(\phi_{\omega_k})})\|^2 \\
&\quad + \frac{8L^2\zeta_k^2}{(1-\delta)\xi_k} \|\bar{D}(\omega_k, \hat{\mu}_k, \hat{V}_{l,k}) - \bar{D}(\omega_k, \mu^\star(\phi_{\omega_k}), V_l^{\phi_{\omega_k},\mu^\star(\phi_{\omega_k})})\|^2 \\
&\leq \frac{(1-\delta)\xi_k}{16} \|y_k\|^2 + \frac{8L^2\zeta_k^2}{(1-\delta)\xi_k} \|\nabla_\omega J_l(\phi_{\omega_k}, \mu) \mid_{\mu=\mu^\star(\phi_{\omega_k})}\|^2 \\
&\quad + \frac{16L^2 L_D^2 \zeta_k^2}{(1-\delta)\xi_k} \|\hat{\mu}_k - \mu^\star(\phi_{\omega_k})\|^2 + \frac{16L^2 L_D^2 \zeta_k^2}{(1-\delta)\xi_k} \|\hat{V}_{l,k} - V_l^{\phi_{\omega_k},\mu^\star(\phi_{\omega_k})}\|^2 \\
&\leq \frac{(1-\delta)\xi_k}{16} \|y_k\|^2 + \frac{8L^2\eta_2^2\zeta_k^2}{(1-\delta)\xi_k} \|\nabla_\omega J_l(\phi_{\omega_k}, \mu^\star(\phi_{\omega_k}))\|^2 + \frac{16L^2 L_D^2 \zeta_k^2}{(1-\delta)\xi_k} \|\hat{\mu}_k - \mu^\star(\phi_{\omega_k})\|^2 \\
&\quad + \frac{32L^2 L_D^2 \zeta_k^2}{(1-\delta)\xi_k} \|V_l^{\phi_{\omega_k},\hat{\mu}_k} - V_l^{\phi_{\omega_k},\mu^\star(\phi_{\omega_k})}\|^2 + \frac{32L^2 L_D^2 \zeta_k^2}{(1-\delta)\xi_k} \|\hat{V}_{l,k} - V_l^{\phi_{\omega_k},\hat{\mu}_k}\|^2 \\
&\leq \frac{(1-\delta)\xi_k}{16} \|y_k\|^2 + \frac{8L^2\eta_2^2\zeta_k^2}{(1-\delta)\xi_k} \varepsilon_k^\phi + \frac{48L^2 L_V^2 L_D^2 \zeta_k^2}{(1-\delta)\xi_k} \varepsilon_k^\mu + \frac{32L^2 L_D^2 \zeta_k^2}{(1-\delta)\xi_k} \varepsilon_{l,k}^V,
\end{aligned}
\tag{78}
$$

where the fifth inequality follows from Assumption 6.

To bound the second term of (77),

$$
\begin{aligned}
&\zeta_k \mathbb{E}[\langle y_k, \Big(\nabla_\omega \mu^\star(\phi_{\omega_{k+1}^z}) - \nabla_\omega \mu^\star(\phi_{\omega_k})\Big)^\top \Big(D(\omega_k, \hat{\mu}_k, \hat{V}_{l,k}, s_k, b_k, s_k') - \bar{D}(\omega_k, \hat{\mu}_k, \hat{V}_{l,k})\Big)\rangle] \\
&\leq 2B_D \zeta_k \mathbb{E}[\|y_k\| \|\nabla_\omega \mu^\star(\phi_{\omega_{k+1}^z}) - \nabla_\omega \mu^\star(\phi_{\omega_k})\|] \\
&\leq 2B_D L \zeta_k \mathbb{E}[\|y_k\| \|\phi_{\omega_{k+1}^z} - \phi_{\omega_k}\|] \\
&\leq 2B_D L \zeta_k \mathbb{E}[\|y_k\| \|\phi_{\omega_{k+1}} - \phi_{\omega_k}\|] \\
&\leq 2B_D^2 L \zeta_k^2 \mathbb{E}[\|y_k\|] \\
&\leq B_D^2 L \zeta_k^2 \mathbb{E}[\|y_k\|^2] + B_D^2 L \zeta_k^2.
\end{aligned}
\tag{79}
$$

Substituting (78) and (79) into (77),

$$
\begin{aligned}
&\mathbb{E}[\langle \hat{\mu}_k - \mu^\star(\phi_{\omega_k}) + \xi_k \bar{H}(\pi^\star(\phi_{\omega_k}, \hat{\mu}_k), \omega_k, \hat{\mu}_k), \mu^\star(\phi_{\omega_{k+1}}) - \mu^\star(\phi_{\omega_k})\rangle] \\
&\leq \frac{(1-\delta)\xi_k}{16} \mathbb{E}[\|y_k\|^2] + \frac{8L^2\eta_2^2\zeta_k^2}{(1-\delta)\xi_k} \mathbb{E}[\varepsilon_k^\phi] + \frac{48L^2 L_V^2 L_D^2 \zeta_k^2}{(1-\delta)\xi_k} \mathbb{E}[\varepsilon_k^\mu] + \frac{32L^2 L_D^2 \zeta_k^2}{(1-\delta)\xi_k} \mathbb{E}[\varepsilon_{l,k}^V]
\end{aligned}
$$

$$+ B_D^2 L \zeta_k^2 \mathbb{E}[\|y_k\|^2] + B_D^2 L \zeta_k^2$$

$$\leq \frac{(1-\delta)\xi_k}{8} \mathbb{E}[\|y_k\|^2] + \frac{8L^2\eta_2^2\zeta_k^2}{(1-\delta)\xi_k}\mathbb{E}[\varepsilon_k^\phi] + \frac{48L^2L_V^2L_D^2\zeta_k^2}{(1-\delta)\xi_k}\mathbb{E}[\varepsilon_k^\mu] + \frac{32L^2L_D^2\zeta_k^2}{(1-\delta)\xi_k}\mathbb{E}[\varepsilon_{l,k}^V] + B_D^2 L\zeta_k^2,$$

where the last inequality follows from the step size conditions $\zeta_k \leq 1$ and $\frac{\zeta_k}{\xi_k} \leq \frac{1-\delta}{16B_D^2 L}$.

∎

## D.6 Proof of Lemma 6

Within the proof of Lemma 6, we denote $x_k = (\phi_{\omega_k}, \hat{\mu}_k)$. By the law of total expectation,

$$-\mathbb{E}[\langle \nabla_x J_f(\pi_{\theta_k}, \phi_{\omega_k}, \hat{\mu}_k) - \nabla_x J_f(\pi, \phi_{\omega_k}, \hat{\mu}_k) \mid_{\pi = \pi^\star(\phi_{\omega_k}, \hat{\mu}_k)}, x_{k+1} - x_k \rangle]$$

$$= -\mathbb{E}[\langle \nabla_x J_f(\pi_{\theta_k}, \phi_{\omega_k}, \hat{\mu}_k) - \nabla_x J_f(\pi, \phi_{\omega_k}, \hat{\mu}_k) \mid_{\pi = \pi^\star(\phi_{\omega_k}, \hat{\mu}_k)}, \mathbb{E}\left[ \begin{array}{c} \zeta_k D(\omega_k, \hat{\mu}_k, \hat{V}_{l,k}, s_k, b_k, s_k') \\ \xi_k H(\hat{\mu}_k, \bar{s}_k) \end{array} \middle| \mathcal{F}_{k-1} \right] \rangle]$$

$$= \mathbb{E}[\langle \nabla_x J_f(\pi_{\theta_k}, \phi_{\omega_k}, \hat{\mu}_k) - \nabla_x J_f(\pi, \phi_{\omega_k}, \hat{\mu}_k) \mid_{\pi = \pi^\star(\phi_{\omega_k}, \hat{\mu}_k)}, \left[ \begin{array}{c} \zeta_k \bar{D}(\omega_k, \hat{\mu}_k, \hat{V}_{l,k}) \\ \xi_k H(\pi_{\theta_k}, \omega_k, \hat{\mu}_k) \end{array} \right] \rangle]$$

$$\leq \frac{(1-\gamma)\tau^2\rho_{\min}^3 p_{\min}^2 \alpha_k}{4|\mathcal{S}|L_V^2} \mathbb{E}[\|\nabla_x J_f(\pi_{\theta_k}, \phi_{\omega_k}, \hat{\mu}_k) - \nabla_x J_f(\pi, \phi_{\omega_k}, \hat{\mu}_k) \mid_{\pi = \pi^\star(\phi_{\omega_k}, \hat{\mu}_k)} \|^2]$$

$$+ \frac{|\mathcal{S}|L_V^2\zeta_k^2}{(1-\gamma)\tau^2\rho_{\min}^3 p_{\min}^2 \alpha_k} \mathbb{E}[\|\bar{D}(\omega_k, \hat{\mu}_k, \hat{V}_{l,k})\|^2] + \frac{|\mathcal{S}|L_V^2\xi_k^2}{(1-\gamma)\tau^2\rho_{\min}^3 p_{\min}^2 \alpha_k} \mathbb{E}[\|\bar{H}(\pi_{\theta_k}, \omega_k, \hat{\mu}_k)\|^2]$$

$$\leq \frac{(1-\gamma)\tau^2\rho_{\min}^3 p_{\min}^2 \alpha_k}{4|\mathcal{S}|} \mathbb{E}[\|\pi^\star(\phi_{\omega_k}, \hat{\mu}_k) - \pi_{\theta_k}\|^2]$$

$$+ \frac{2|\mathcal{S}|L_V^2\zeta_k^2}{(1-\gamma)\tau^2\rho_{\min}^3 p_{\min}^2 \alpha_k} \mathbb{E}[\|\bar{D}(\omega_k, \mu^\star(\phi_{\omega_k}), V_l^{\phi_{\omega_k}, \mu^\star(\phi_{\omega_k})})\|^2]$$

$$+ \frac{2|\mathcal{S}|L_V^2\zeta_k^2}{(1-\gamma)\tau^2\rho_{\min}^3 p_{\min}^2 \alpha_k} \mathbb{E}[\|\bar{D}(\omega_k, \hat{\mu}_k, \hat{V}_{l,k}) - \bar{D}(\omega_k, \mu^\star(\phi_{\omega_k}), V_l^{\phi_{\omega_k}, \mu^\star(\phi_{\omega_k})})\|^2]$$

$$+ \frac{2|\mathcal{S}|L_V^2\zeta_k^2}{(1-\gamma)\tau^2\rho_{\min}^3 p_{\min}^2 \alpha_k} \mathbb{E}[\|\bar{H}(\pi^\star(\phi_{\omega_k}, \hat{\mu}_k), \omega_k, \hat{\mu}_k) - \bar{H}(\pi^\star(\phi_{\omega_k}), \omega_k, \mu^\star(\phi_{\omega_k}))\|^2]$$

$$+ \frac{2|\mathcal{S}|L_V^2\zeta_k^2}{(1-\gamma)\tau^2\rho_{\min}^3 p_{\min}^2 \alpha_k} \mathbb{E}[\|\bar{H}(\pi_{\theta_k}, \omega_k, \hat{\mu}_k) - \bar{H}(\pi^\star(\phi_{\omega_k}, \hat{\mu}_k), \omega_k, \hat{\mu}_k)\|^2]$$

$$\leq \frac{(1-\gamma)\tau^2\rho_{\min}^3 p_{\min}^2 \alpha_k}{4|\mathcal{S}|} \mathbb{E}[\|\pi^\star(\phi_{\omega_k}, \hat{\mu}_k) - \pi_{\theta_k}\|^2]$$

$$+ \frac{2|\mathcal{S}|L_V^2\zeta_k^2}{(1-\gamma)\tau^2\rho_{\min}^3 p_{\min}^2 \alpha_k} \mathbb{E}[\|\nabla_\omega J_l(\phi_{\omega_k}, \mu) \mid_{\mu=\mu^\star(\phi_{\omega_k})} \|^2]$$

$$+ \frac{4|\mathcal{S}|L_V^2L_D^2\zeta_k^2}{(1-\gamma)\tau^2\rho_{\min}^3 p_{\min}^2 \alpha_k} \mathbb{E}[\|\hat{\mu}_k - \mu^\star(\phi_{\omega_k})\|^2] + \frac{4|\mathcal{S}|L_V^2L_D^2\zeta_k^2}{(1-\gamma)\tau^2\rho_{\min}^3 p_{\min}^2 \alpha_k} \mathbb{E}[\|\hat{V}_{l,k} - V_l^{\phi_{\omega_k}, \mu^\star(\phi_{\omega_k})}\|^2]$$

$$+ \frac{2|\mathcal{S}|L_V^2L_D^2\xi_k^2}{(1-\gamma)\tau^2\rho_{\min}^3 p_{\min}^2 \alpha_k} \cdot L_H^2 \mathbb{E}[\|\hat{\mu}_k - \mu^\star(\phi_{\omega_k})\|^2 + L^2 \|\hat{\mu}_k - \mu^\star(\phi_{\omega_k})\|^2]$$

$$+ \frac{2|\mathcal{S}|L_V^2L_D^2\xi_k^2}{(1-\gamma)\tau^2\rho_{\min}^3 p_{\min}^2 \alpha_k} \cdot L_H^2 \mathbb{E}[\|\pi^\star(\phi_{\omega_k, \hat{\mu}_k}) - \pi_{\theta_k}\|^2]$$

$$\leq \frac{(1-\gamma)\tau^2\rho_{\min}^3 p_{\min}^2 \alpha_k}{4|\mathcal{S}|} \mathbb{E}[\|\pi^\star(\phi_{\omega_k}, \hat{\mu}_k) - \pi_{\theta_k}\|^2]$$

$$+ \frac{2|\mathcal{S}|L_V^2\eta_2^2\zeta_k^2}{(1-\gamma)\tau^2\rho_{\min}^3 p_{\min}^2 \alpha_k} \mathbb{E}[\|\nabla_\omega J_l(\phi_{\omega_k}, \mu^\star(\phi_{\omega_k}))\|^2] + \frac{4|\mathcal{S}|L_V^2L_D^2\zeta_k^2}{(1-\gamma)\tau^2\rho_{\min}^3 p_{\min}^2 \alpha_k} \mathbb{E}[\|\hat{\mu}_k - \mu^\star(\phi_{\omega_k})\|^2]$$

$$+ \frac{8|\mathcal{S}|L_V^2L_D^2\zeta_k^2}{(1-\gamma)\tau^2\rho_{\min}^3 p_{\min}^2 \alpha_k} \mathbb{E}[\|V_l^{\phi_{\omega_k}, \hat{\mu}_k} - V_l^{\phi_{\omega_k}, \mu^\star(\phi_{\omega_k})}\|^2] + \frac{8|\mathcal{S}|L_V^2L_D^2\zeta_k^2}{(1-\gamma)\tau^2\rho_{\min}^3 p_{\min}^2 \alpha_k} \mathbb{E}[\|\hat{V}_{l,k} - V_l^{\phi_{\omega_k}, \hat{\mu}_k}\|^2]$$

$$+ \frac{4|\mathcal{S}|L^2L_V^2L_D^2L_H^2\xi_k^2}{(1-\gamma)\tau^2\rho_{\min}^3 p_{\min}^2 \alpha_k} \mathbb{E}[\|\hat{\mu}_k - \mu^\star(\phi_{\omega_k})\|^2] + \frac{2|\mathcal{S}|L_V^2L_D^2L_H^2\xi_k^2}{(1-\gamma)\tau^2\rho_{\min}^3 p_{\min}^2 \alpha_k} \mathbb{E}[\|\pi^\star(\phi_{\omega_k, \hat{\mu}_k}) - \pi_{\theta_k}\|^2]$$

$$\leq \frac{(1-\gamma)\tau^2\rho_{\min}^3 p_{\min}^2 \alpha_k}{4|\mathcal{S}|}\mathbb{E}[\|\pi^\star(\phi_{\omega_k},\hat{\mu}_k) - \pi_{\theta_k}\|^2] + \frac{2|\mathcal{S}|L_V^2 L_D^2 L_H^2 \xi_k^2}{(1-\gamma)\tau^2\rho_{\min}^3 p_{\min}^2 \alpha_k}\mathbb{E}[\|\pi^\star(\phi_{\omega_k,\hat{\mu}_k}) - \pi_{\theta_k}\|^2]$$

$$+ \frac{2|\mathcal{S}|L_V^2 \eta_2^2 \zeta_k^2}{(1-\gamma)\tau^2\rho_{\min}^3 p_{\min}^2 \alpha_k}\mathbb{E}[\varepsilon_k^\phi] + \frac{16|\mathcal{S}|L^2 L_V^4 L_D^2 L_H^2 \xi_k^2}{(1-\gamma)\tau^2\rho_{\min}^3 p_{\min}^2 \alpha_k}\mathbb{E}[\varepsilon_k^\mu] + \frac{8|\mathcal{S}|L_V^2 \zeta_k^2}{(1-\gamma)\tau^2\rho_{\min}^3 p_{\min}^2 \alpha_k}\mathbb{E}[\varepsilon_{l,k}^V],$$

$$\tag{80}$$

where the second inequality follows from the fact that $\bar{H}(\pi^\star(\phi),\omega_k,\mu^\star(\phi)) = 0$, and the last inequality simplifies and combines terms under the step size condition $\frac{\zeta_k}{\xi_k} \leq LL_H$.

∎

### D.7 Proof of Lemma 7

Within the proof of this lemma, we employ the shorthand notation $x_k = [\theta_k, \omega_k, \hat{\mu}_k]$, $\ell(x_k) = V_f^{\pi_{\theta_k},\phi_{\omega_k},\hat{\mu}_k}$, and

$$y_k = \hat{V}_{f,k} - V_f^{\pi_{\theta_k},\phi_{\omega_k},\hat{\mu}_k} + \beta_k \bar{G}_f(\theta_k,\omega_k,\hat{\mu}_k,\hat{V}_{f,k}).$$

The value function is smooth, i.e. has Lipschitz gradients. As a result, we have from the mean-value theorem that there exists $x_{k+1}^z = zx_k + (1-z)x_{k+1}$ for some scalar $z \in [0,1]$ such that

$$\ell(x_k) - \ell(x_{k+1})$$

$$= \nabla_x \ell(x_{k+1}^z)^\top \left(x_k - x_{k+1}\right)$$

$$= \left(\nabla_\theta V_f^{\pi_{\theta_{k+1}^z},\phi_{\omega_{k+1}^z},\hat{\mu}_{k+1}^z}\right)^\top \left(\theta_k - \theta_{k+1}\right) + \left(V_f^{\pi_{\theta_{k+1}^z},\phi_{\omega_{k+1}^z},\hat{\mu}_{k+1}^z}\right)^\top \left(\omega_k - \omega_{k+1}\right)$$

$$+ \left(V_f^{\pi_{\theta_{k+1}^z},\phi_{\omega_{k+1}^z},\hat{\mu}_{k+1}^z}\right)^\top \left(\hat{\mu}_k - \hat{\mu}_{k+1}\right)$$

$$= \alpha_k \left(\nabla_\theta V_f^{\pi_{\theta_{k+1}^z},\phi_{\omega_{k+1}^z},\hat{\mu}_{k+1}^z}\right)^\top \bar{F}(\theta_k,\omega_k,\hat{\mu}_k,\hat{V}_{f,k})$$

$$+ \alpha_k \left(\nabla_\theta V_f^{\pi_{\theta_{k+1}^z},\phi_{\omega_{k+1}^z},\hat{\mu}_{k+1}^z}\right)^\top \left(F(\theta_k,\omega_k,\hat{\mu}_k,\hat{V}_{f,k},s_k,a_k,b_k,s_k') - \bar{F}(\theta_k,\omega_k,\hat{\mu}_k,\hat{V}_{f,k})\right)$$

$$+ \zeta_k \left(\nabla_\omega V_f^{\pi_{\theta_{k+1}^z},\phi_{\omega_{k+1}^z},\hat{\mu}_{k+1}^z}\right)^\top \bar{D}(\omega_k,\hat{\mu}_k,\hat{V}_{l,k})$$

$$+ \zeta_k \left(\nabla_\theta V_f^{\pi_{\theta_{k+1}^z},\phi_{\omega_{k+1}^z},\hat{\mu}_{k+1}^z}\right)^\top \left(D(\omega_k,\hat{\mu}_k,\hat{V}_{l,k},s_k,b_k,s_k') - \bar{D}(\omega_k,\hat{\mu}_k,\hat{V}_{l,k})\right)$$

$$+ \xi_k \left(\nabla_\mu V_f^{\pi_{\theta_{k+1}^z},\phi_{\omega_{k+1}^z},\hat{\mu}_{k+1}^z}\right)^\top \bar{H}(\pi_{\theta_k},\omega_k,\hat{\mu}_k)$$

$$+ \xi_k \left(\nabla_\theta V_f^{\pi_{\theta_{k+1}^z},\phi_{\omega_{k+1}^z},\hat{\mu}_{k+1}^z}\right)^\top \left(H(\hat{\mu}_k,\bar{s}_k) - \bar{H}(\pi_{\theta_k},\omega_k,\hat{\mu}_k)\right), \tag{81}$$

where we denote $\theta_{k+1}^z = z\theta_k + (1-z)\theta_{k+1}$, $\omega_{k+1}^z = z\omega_k + (1-z)\omega_{k+1}$, $\hat{\mu}_{k+1}^z = z\hat{\mu}_k + (1-z)\hat{\mu}_{k+1}$. Plugging (81) into the cross term of interest, we have

$$\langle \hat{V}_{f,k} - V_f^{\pi_{\theta_k},\phi_{\omega_k},\hat{\mu}_k} + \beta_k \bar{G}_f(\theta_k,\omega_k,\hat{\mu}_k,\hat{V}_{f,k}), V_f^{\pi_{\theta_k},\phi_{\omega_k},\hat{\mu}_k} - V_f^{\pi_{\theta_{k+1}},\phi_{\omega_{k+1}},\hat{\mu}_{k+1}}\rangle$$

$$= \langle y_k, \ell(x_k) - \ell(x_{k+1})\rangle$$

$$= \alpha_k \langle y_k, \left(\nabla_\theta V_f^{\pi_{\theta_{k+1}^z},\phi_{\omega_{k+1}^z},\hat{\mu}_{k+1}^z}\right)^\top \bar{F}(\theta_k,\omega_k,\hat{\mu}_k,\hat{V}_{f,k})\rangle$$

$$+ \alpha_k \langle y_k, \left(\nabla_\theta V_f^{\pi_{\theta_{k+1}^z},\phi_{\omega_{k+1}^z},\hat{\mu}_{k+1}^z}\right)^\top \left(F(\theta_k,\omega_k,\hat{\mu}_k,\hat{V}_{f,k}),s_k,a_k,b_k,s_k') - \bar{F}(\theta_k,\omega_k,\hat{\mu}_k,\hat{V}_{f,k})\right)\rangle$$

$$+ \zeta_k \langle y_k, \left(\nabla_\omega V_f^{\pi_{\theta_{k+1}^z},\phi_{\omega_{k+1}^z},\hat{\mu}_{k+1}^z}\right)^\top \bar{D}(\omega_k,\hat{\mu}_k,\hat{V}_{l,k})\rangle$$

$$+ \zeta_k \langle y_k, \left(\nabla_\omega V_f^{\pi_{\theta_{k+1}^z},\phi_{\omega_{k+1}^z},\hat{\mu}_{k+1}^z}\right)^\top \left(D(\omega_k,\hat{\mu}_k,\hat{V}_{l,k},s_k,b_k,s_k') - \bar{D}(\omega_k,\hat{\mu}_k,\hat{V}_{l,k})\right)\rangle$$

$$+ \xi_k \langle y_k, \left(\nabla_\mu V_f^{\pi_{\theta_{k+1}^z},\phi_{\omega_{k+1}^z},\hat{\mu}_{k+1}^z}\right)^\top \bar{H}(\pi_{\theta_k},\omega_k,\hat{\mu}_k)\rangle$$

$$+ \xi_k \langle y_k, \left(\nabla_\mu V_f^{\pi_{\theta_{k+1}^z}, \phi_{\omega_{k+1}^z}, \hat{\mu}_{k+1}^z}\right)^\top \left(H(\hat{\mu}_k, \bar{s}_k) - \bar{H}(\pi_{\theta_k}, \omega_k, \hat{\mu}_k)\right)\rangle. \tag{82}$$

We bound each term of (82) individually. First, by Young's inequality

$$\alpha_k \langle y_k, \left(\nabla_\theta V_f^{\pi_{\theta_{k+1}^z}, \phi_{\omega_{k+1}^z}, \hat{\mu}_{k+1}^z}\right)^\top \bar{F}(\theta_k, \omega_k, \hat{\mu}_k, \hat{V}_{f,k})\rangle$$

$$\leq L_V \alpha_k \|y_k\| \|\bar{F}(\theta_k, \omega_k, \hat{\mu}_k, \hat{V}_{f,k})\|$$

$$\leq \frac{(1-\gamma)\beta_k}{12}\|y_k\|^2 + \frac{3L_V^2 \alpha_k^2}{(1-\gamma)\beta_k}\|\bar{F}(\theta_k, \omega_k, \hat{\mu}_k, \hat{V}_{f,k})\|^2$$

$$\leq \frac{(1-\gamma)\beta_k}{12}\|y_k\|^2 + \frac{6L_V^2 \alpha_k^2}{(1-\gamma)\beta_k}\|\bar{F}(\theta_k, \omega_k, \hat{\mu}_k, V_f^{\pi_{\theta_k}, \phi_{\omega_k}, \hat{\mu}_k})\|^2$$

$$\qquad + \frac{6L_V^2 \alpha_k^2}{(1-\gamma)\beta_k}\|\bar{F}(\theta_k, \omega_k, \hat{\mu}_k, \hat{V}_{f,k}) - \bar{F}(\theta_k, \omega_k, \hat{\mu}_k, V_f^{\pi_{\theta_k}, \phi_{\omega_k}, \hat{\mu}_k})\|^2$$

$$\leq \frac{(1-\gamma)\beta_k}{12}\|y_k\|^2 + \frac{6L_V^2 \alpha_k^2}{(1-\gamma)\beta_k}\|\nabla_\theta J_f(\pi_{\theta_k}, \phi_{\omega_k}, \hat{\mu}_k)\|^2 + \frac{6L_V^4 \alpha_k^2}{(1-\gamma)\beta_k}\varepsilon_{f,k}^V. \tag{83}$$

For the second term of (82), we take the expectation

$$\alpha_k \mathbb{E}\left[\langle y_k, \left(\nabla_\theta V_f^{\pi_{\theta_{k+1}^z}, \phi_{\omega_{k+1}^z}, \hat{\mu}_{k+1}^z}\right)^\top \left(F(\theta_k, \omega_k, \hat{\mu}_k, \hat{V}_{f,k}, s_k, a_k, b_k, s_k') - \bar{F}(\theta_k, \omega_k, \hat{\mu}_k, \hat{V}_{f,k})\right)\rangle\right]$$

$$= \alpha_k \mathbb{E}[\langle y_k, \left(\nabla_\theta V_f^{\pi_{\theta_{k+1}^z}, \phi_{\omega_{k+1}^z}, \hat{\mu}_{k+1}^z} - \nabla_\theta V_f^{\pi_{\theta_k}, \phi_{\omega_k}, \hat{\mu}_k}\right)^\top \left(F(\theta_k, \omega_k, \hat{\mu}_k, \hat{V}_{f,k}, s_k, a_k, b_k, s_k') - \bar{F}(\theta_k, \omega_k, \hat{\mu}_k, \hat{V}_{f,k})\right)\rangle]$$

$$\qquad + \alpha_k \mathbb{E}[\langle y_k, \left(\nabla_\theta V_f^{\pi_{\theta_k}, \phi_{\omega_k}, \hat{\mu}_k}\right)^\top \left(F(\theta_k, \omega_k, \hat{\mu}_k, \hat{V}_{f,k}, s_k, a_k, b_k, s_k') - \bar{F}(\theta_k, \omega_k, \hat{\mu}_k, \hat{V}_{f,k})\right)\rangle]$$

$$= \alpha_k \mathbb{E}[\langle y_k, \left(\nabla_\theta V_f^{\pi_{\theta_{k+1}^z}, \phi_{\omega_{k+1}^z}, \hat{\mu}_{k+1}^z} - \nabla_\theta V_f^{\pi_{\theta_k}, \phi_{\omega_k}, \hat{\mu}_k}\right)^\top \left(F(\theta_k, \omega_k, \hat{\mu}_k, \hat{V}_{f,k}, s_k, a_k, b_k, s_k') - \bar{F}(\theta_k, \omega_k, \hat{\mu}_k, \hat{V}_{f,k})\right)\rangle]$$

$$\leq 2B_F \alpha_k \mathbb{E}[\|y_k\| \|\nabla_\theta V_f^{\pi_{\theta_{k+1}^z}, \phi_{\omega_{k+1}^z}, \hat{\mu}_{k+1}^z} - \nabla_\theta V_f^{\pi_{\theta_k}, \phi_{\omega_k}, \hat{\mu}_k}\|]$$

$$\leq 2B_F L_{VV} \alpha_k \mathbb{E}[\|y_k\| \left(\|\pi_{\theta_{k+1}^z} - \pi_{\theta_k}\| + \|\phi_{\omega_{k+1}^z} - \phi_{\omega_k}\| + \|\hat{\mu}_{k+1}^z - \hat{\mu}_k\|\right)]$$

$$\leq 2B_F L_{VV} \alpha_k \mathbb{E}[\|y_k\| \left(\|\pi_{\theta_{k+1}} - \pi_{\theta_k}\| + \|\phi_{\omega_{k+1}} - \phi_{\omega_k}\| + \|\hat{\mu}_{k+1} - \hat{\mu}_k\|\right)]$$

$$\leq 2B_F L_{VV} \alpha_k \mathbb{E}[\|y_k\|] \left(\alpha_k B_F + \zeta_k B_D + \xi_k B_H\right)$$

$$\leq 2B_F L_{VV}(B_F + B_D + B_H)\alpha_k^2 \mathbb{E}[\|y_k\|]$$

$$\leq B_F L_{VV}(B_F + B_D + B_H)\alpha_k^2 \mathbb{E}[\|y_k\|^2] + B_F L_{VV}(B_F + B_D + B_H)\alpha_k^2, \tag{84}$$

where the fifth inequality follows from the step size condition $\zeta_k \leq \xi_l \leq \alpha_k$, and the second equation follows from

$$\mathbb{E}[\langle y_k, \left(\nabla_\theta V_f^{\pi_{\theta_k}, \phi_{\omega_k}, \hat{\mu}_k}\right)^\top \left(F(\theta_k, \omega_k, \hat{\mu}_k, \hat{V}_{f,k}, s_k, a_k, b_k, s_k') - \bar{F}(\theta_k, \omega_k, \hat{\mu}_k, \hat{V}_{f,k})\right)]$$

$$= \mathbb{E}[\langle y_k, \left(\nabla_\theta V_f^{\pi_{\theta_k}, \phi_{\omega_k}, \hat{\mu}_k}\right)^\top \mathbb{E}\left[F(\theta_k, \omega_k, \hat{\mu}_k, \hat{V}_{f,k}, s_k, a_k, b_k, s_k') - \bar{F}(\theta_k, \omega_k, \hat{\mu}_k, \hat{V}_{f,k}) \mid \mathcal{F}_{k-1}\right]]$$

$$= 0.$$

The third term of (82) can be bounded similar to the first term,

$$\zeta_k \langle y_k, \left(\nabla_\omega V_f^{\pi_{\theta_{k+1}^z}, \phi_{\omega_{k+1}^z}, \hat{\mu}_{k+1}^z}\right)^\top \bar{D}(\omega_k, \hat{\mu}_k, \hat{V}_{l,k})\rangle$$

$$\leq L_V \zeta_k \|y_k\| \|\bar{D}(\omega_k, \hat{\mu}_k, \hat{V}_{l,k})\|$$

$$\leq \frac{(1-\gamma)\beta_k}{12}\|y_k\|^2 + \frac{3L_V^2 \zeta_k^2}{(1-\gamma)\beta_k}\|\bar{D}(\omega_k, \hat{\mu}_k, \hat{V}_{l,k})\|^2$$

$$\leq \frac{(1-\gamma)\beta_k}{12}\|y_k\|^2 + \frac{6L_V^2 \zeta_k^2}{(1-\gamma)\beta_k}\|\bar{D}(\omega_k, \mu^\star(\phi_{\omega_k}), V_l^{\phi_{\omega_k}, \mu^\star(\phi_{\omega_k})})\|^2$$

$$+ \frac{6L_V^2\zeta_k^2}{(1-\gamma)\beta_k} \|\bar{D}(\omega_k, \hat{\mu}_k, \hat{V}_{l,k}) - \bar{D}(\omega_k, \mu^\star(\phi_{\omega_k}), V_l^{\phi_{\omega_k}, \mu^\star(\phi_{\omega_k})})\|^2$$

$$\leq \frac{(1-\gamma)\beta_k}{12}\|y_k\|^2 + \frac{6L_V^2\zeta_k^2}{(1-\gamma)\beta_k} \|\nabla_\omega J_l(\phi_{\omega_k}, \mu) \mid_{\mu=\mu^\star(\phi_{\omega_k})}\|^2$$

$$+ \frac{12L_V^2 L_D^2 \zeta_k^2}{(1-\gamma)\beta_k}\|\hat{\mu}_k - \mu^\star(\phi_{\omega_k})\|^2 + \frac{12L_V^2 L_D^2 \zeta_k^2}{(1-\gamma)\beta_k}\|\hat{V}_{l,k} - V_l^{\phi_{\omega_k}, \mu^\star(\phi_{\omega_k})}\|^2$$

$$\leq \frac{(1-\gamma)\beta_k}{12}\|y_k\|^2 + \frac{6L_V^2\eta_2^2\zeta_k^2}{(1-\gamma)\beta_k}\|\nabla_\omega J_l(\phi_{\omega_k}, \mu^\star(\phi_{\omega_k}))\|^2 + \frac{12L_V^2 L_D^2 \zeta_k^2}{(1-\gamma)\beta_k}\|\hat{\mu}_k - \mu^\star(\phi_{\omega_k})\|^2$$

$$+ \frac{24L_V^2 L_D^2 \zeta_k^2}{(1-\gamma)\beta_k}\|V_l^{\phi_{\omega_k}, \hat{\mu}_k} - V_l^{\phi_{\omega_k}, \mu^\star(\phi_{\omega_k})}\|^2 + \frac{24L_V^2 L_D^2 \zeta_k^2}{(1-\gamma)\beta_k}\|\hat{V}_{l,k} - V_l^{\phi_{\omega_k}, \hat{\mu}_k}\|^2$$

$$\leq \frac{(1-\gamma)\beta_k}{12}\|y_k\|^2 + \frac{6L_V^2\eta_2^2\zeta_k^2}{(1-\gamma)\beta_k}\varepsilon_k^\phi + \frac{36L_V^4 L_D^2 \zeta_k^2}{(1-\gamma)\beta_k}\varepsilon_k^\mu + \frac{24L_V^2 L_D^2 \zeta_k^2}{(1-\gamma)\beta_k}\varepsilon_{l,k}^V, \tag{85}$$

where the fifth inequality follows from Assumption 6.

For the fourth term of (82), we again take the expectation and use the technique in (84)

$$\zeta_k\mathbb{E}\langle y_k, \left(\nabla_\omega V_f^{\pi_{\theta_{k+1}^z}, \phi_{\omega_{k+1}^z}, \hat{\mu}_{k+1}^z}\right)^\top \left(D(\omega_k, \hat{\mu}_k, \hat{V}_{f,k}, s_k, b_k, s'_k) - \bar{D}(\omega_k, \hat{\mu}_k, \hat{V}_{f,k})\right)\rangle$$

$$= \zeta_k\mathbb{E}[\langle y_k, \left(\nabla_\theta V_f^{\pi_{\theta_{k+1}^z}, \phi_{\omega_{k+1}^z}, \hat{\mu}_{k+1}^z} - \nabla_\theta V_f^{\pi_{\theta_k}, \phi_{\omega_k}, \hat{\mu}_k}\right)^\top \left(D(\omega_k, \hat{\mu}_k, \hat{V}_{l,k}, s_k, b_k, s'_k) - \bar{D}(\omega_k, \hat{\mu}_k, \hat{V}_{l,k})\right)\rangle]$$

$$+ \zeta_k\mathbb{E}[\langle y_k, \left(\nabla_\theta V_f^{\pi_{\theta_k}, \phi_{\omega_k}, \hat{\mu}_k}\right)^\top \left(D(\omega_k, \hat{\mu}_k, \hat{V}_{l,k}, s_k, b_k, s'_k) - \bar{D}(\omega_k, \hat{\mu}_k, \hat{V}_{l,k})\right)\rangle]$$

$$= \zeta_k\mathbb{E}[\langle y_k, \left(\nabla_\theta V_f^{\pi_{\theta_{k+1}^z}, \phi_{\omega_{k+1}^z}, \hat{\mu}_{k+1}^z} - \nabla_\theta V_f^{\pi_{\theta_k}, \phi_{\omega_k}, \hat{\mu}_k}\right)^\top \left(D(\omega_k, \hat{\mu}_k, \hat{V}_{l,k}, s_k, b_k, s'_k) - \bar{D}(\omega_k, \hat{\mu}_k, \hat{V}_{l,k})\right)\rangle]$$

$$\leq 2B_D\zeta_k\mathbb{E}[\|y_k\|\|\nabla_\theta V_f^{\pi_{\theta_{k+1}^z}, \phi_{\omega_{k+1}^z}, \hat{\mu}_{k+1}^z} - \nabla_\theta V_f^{\pi_{\theta_k}, \phi_{\omega_k}, \hat{\mu}_k}\|]$$

$$\leq 2B_D L_{VV}\zeta_k\mathbb{E}[\|y_k\|\left(\|\pi_{\theta_{k+1}^z} - \pi_{\theta_k}\| + \|\phi_{\omega_{k+1}^z} - \phi_{\omega_k}\| + \|\hat{\mu}_{k+1}^z - \hat{\mu}_k\|\right)]$$

$$\leq 2B_D L_{VV}\zeta_k\mathbb{E}[\|y_k\|\left(\|\pi_{\theta_{k+1}} - \pi_{\theta_k}\| + \|\phi_{\omega_{k+1}} - \phi_{\omega_k}\| + \|\hat{\mu}_{k+1} - \hat{\mu}_k\|\right)]$$

$$\leq 2B_D L_{VV}\zeta_k\mathbb{E}[\|y_k\|]\left(\alpha_k B_F + \zeta_k B_D + \xi_k B_H\right)$$

$$\leq B_D L_{VV}(B_F + B_D + B_H)\alpha_k^2\mathbb{E}[\|y_k\|^2] + B_F L_{VV}(B_F + B_D + B_H)\alpha_k^2. \tag{86}$$

For the fifth term of (82), recall the definition of $\pi^\star$ in (2) and (4)

$$\xi_k\langle y_k, \left(\nabla_\mu V_f^{\pi_{\theta_{k+1}^z}, \phi_{\omega_{k+1}^z}, \hat{\mu}_{k+1}^z}\right)^\top \bar{H}(\pi_{\theta_k}, \omega_k, \hat{\mu}_k)\rangle$$

$$\leq L_V\xi_k\|y_k\|\|\bar{H}(\pi_{\theta_k}, \omega_k, \hat{\mu}_k)\|$$

$$\leq \frac{(1-\gamma)\beta_k}{12}\|y_k\|^2 + \frac{3L_V^2\xi_k^2}{(1-\gamma)\beta_k}\|\bar{H}(\pi_{\theta_k}, \omega_k, \hat{\mu}_k)\|^2$$

$$\leq \frac{(1-\gamma)\beta_k}{12}\|y_k\|^2 + \frac{6L_V^2\xi_k^2}{(1-\gamma)\beta_k}\|\bar{H}(\pi^\star(\phi_{\omega_k}, \hat{\mu}_k), \omega_k, \hat{\mu}_k) - \bar{H}(\pi^\star(\phi_{\omega_k}), \omega_k, \mu^\star(\phi_{\omega_k}))\|^2$$

$$+ \frac{6L_V^2\xi_k^2}{(1-\gamma)\beta_k}\|\bar{H}(\pi_{\theta_k}, \omega_k, \hat{\mu}_k) - \bar{H}(\pi^\star(\phi_{\omega_k}, \hat{\mu}_k), \omega_k, \hat{\mu}_k)\|^2$$

$$\leq \frac{(1-\gamma)\beta_k}{12}\|y_k\|^2 + \frac{6L_V^2\xi_k^2}{(1-\gamma)\beta_k} \cdot L_H^2\left(\|\hat{\mu}_k - \mu^\star(\phi_{\omega_k})\|^2 + L^2\|\hat{\mu}_k - \mu^\star(\phi_{\omega_k})\|^2\right)$$

$$+ \frac{6L_V^2 L_H^2\xi_k^2}{(1-\gamma)\beta_k}\|\pi_{\theta_k} - \pi^\star(\phi_{\omega_k}, \hat{\mu}_k)\|^2$$

$$\leq \frac{(1-\gamma)\beta_k}{12}\|y_k\|^2 + \frac{6L_V^2 L_H^2\xi_k^2}{(1-\gamma)\beta_k}\varepsilon_k^\pi + \frac{12L^2 L_V^2 L_H^2\xi_k^2}{(1-\gamma)\beta_k}\varepsilon_k^\mu, \tag{87}$$

where the third inequality follows from the fact that $\bar{H}(\pi^\star(\phi), \omega_k, \mu^\star(\phi)) = 0$.

Finally, for the last term of (82), we have in expectation

$$
\xi_k \mathbb{E}[\langle y_k, \left(\nabla_\mu V_f^{\pi_{\theta^z_{k+1}}, \phi_{\omega^z_{k+1}}, \hat{\mu}^z_{k+1}}\right)^\top \left(H(\hat{\mu}_k, \bar{s}_k) - \bar{H}(\pi_{\theta_k}, \omega_k, \hat{\mu}_k)\right)\rangle]
$$

$$
= \xi_k \mathbb{E}[\langle y_k, \left(\nabla_\theta V_f^{\pi_{\theta^z_{k+1}}, \phi_{\omega^z_{k+1}}, \hat{\mu}^z_{k+1}} - \nabla_\theta V_f^{\pi_{\theta_k}, \phi_{\omega_k}, \hat{\mu}_k}\right)^\top \left(H(\hat{\mu}_k, \bar{s}_k) - \bar{H}(\pi_{\theta_k}, \omega_k, \hat{\mu}_k)\right)\rangle]
$$

$$
\quad + \xi_k \mathbb{E}[\langle y_k, \left(\nabla_\theta V_f^{\pi_{\theta_k}, \phi_{\omega_k}, \hat{\mu}_k}\right)^\top \left(H(\hat{\mu}_k, \bar{s}_k) - \bar{H}(\pi_{\theta_k}, \omega_k, \hat{\mu}_k)\right)\rangle]
$$

$$
= \xi_k \mathbb{E}[\langle y_k, \left(\nabla_\theta V_f^{\pi_{\theta^z_{k+1}}, \phi_{\omega^z_{k+1}}, \hat{\mu}^z_{k+1}} - \nabla_\theta V_f^{\pi_{\theta_k}, \phi_{\omega_k}, \hat{\mu}_k}\right)^\top \left(H(\hat{\mu}_k, \bar{s}_k) - \bar{H}(\pi_{\theta_k}, \omega_k, \hat{\mu}_k)\right)\rangle]
$$

$$
\leq 2B_H \xi_k \mathbb{E}[\|y_k\| \|\nabla_\theta V_f^{\pi_{\theta^z_{k+1}}, \phi_{\omega^z_{k+1}}, \hat{\mu}^z_{k+1}} - \nabla_\theta V_f^{\pi_{\theta_k}, \phi_{\omega_k}, \hat{\mu}_k}\|]
$$

$$
\leq 2B_H L_{VV} \xi_k \mathbb{E}[\|y_k\| \left(\|\pi_{\theta^z_{k+1}} - \pi_{\theta_k}\| + \|\phi_{\omega^z_{k+1}} - \phi_{\omega_k}\| + \|\hat{\mu}^z_{k+1} - \hat{\mu}_k\|\right)]
$$

$$
\leq 2B_H L_{VV} \xi_k \mathbb{E}[\|y_k\| \left(\|\pi_{\theta_{k+1}} - \pi_{\theta_k}\| + \|\phi_{\omega_{k+1}} - \phi_{\omega_k}\| + \|\hat{\mu}_{k+1} - \hat{\mu}_k\|\right)]
$$

$$
\leq 2B_H L_{VV} \xi_k \mathbb{E}[\|y_k\|] \left(\alpha_k B_F + \zeta_k B_D + \xi_k B_H\right)
$$

$$
\leq B_H L_{VV} (B_F + B_D + B_H) \alpha_k^2 \mathbb{E}[\|y_k\|^2] + B_F L_{VV} (B_F + B_D + B_H) \alpha_k^2. \tag{88}
$$

Substituting (83)-(88) into (82), we have

$$
\mathbb{E}[\langle \hat{V}_{f,k} - V_f^{\pi_{\theta_k}, \phi_{\omega_k}, \hat{\mu}_k} + \beta_k \bar{G}_f(\theta_k, \omega_k, \hat{\mu}_k, \hat{V}_{f,k}), V_f^{\pi_{\theta_k}, \phi_{\omega_k}, \hat{\mu}_k} - V_f^{\pi_{\theta_{k+1}}, \phi_{\omega_{k+1}}, \hat{\mu}_{k+1}}\rangle]
$$

$$
\leq \frac{(1-\gamma)\beta_k}{12} \mathbb{E}[\|y_k\|^2] + \frac{6L_V^2 \alpha_k^2}{(1-\gamma)\beta_k} \mathbb{E}[\|\nabla_\theta J_f(\pi_{\theta_k}, \phi_{\omega_k}, \hat{\mu}_k)\|^2] + \frac{6L_V^4 \alpha_k^2}{(1-\gamma)\beta_k} \mathbb{E}[\varepsilon_{f,k}^V]
$$

$$
\quad + B_F L_{VV}(B_F + B_D + B_H)\alpha_k^2 \mathbb{E}[\|y_k\|^2] + B_F L_{VV}(B_F + B_D + B_H)\alpha_k^2
$$

$$
\quad + \frac{(1-\gamma)\beta_k}{12} \mathbb{E}[\|y_k\|^2] + \frac{6L_V^2 \eta_2^2 \zeta_k^2}{(1-\gamma)\beta_k} \mathbb{E}[\varepsilon_k^\phi] + \frac{36L_V^4 L_D^2 \zeta_k^2}{(1-\gamma)\beta_k} \mathbb{E}[\varepsilon_k^\mu] + \frac{24L_V^2 L_D^2 \zeta_k^2}{(1-\gamma)\beta_k} \mathbb{E}[\varepsilon_{l,k}^V]
$$

$$
\quad + B_D L_{VV}(B_F + B_D + B_H)\alpha_k^2 \mathbb{E}[\|y_k\|^2] + B_F L_{VV}(B_F + B_D + B_H)\alpha_k^2
$$

$$
\quad + \frac{(1-\gamma)\beta_k}{12} \mathbb{E}[\|y_k\|^2] + \frac{6L_V^2 L_H^2 \xi_k^2}{(1-\gamma)\beta_k} \mathbb{E}[\varepsilon_k^\pi] + \frac{12L^2 L_V^2 L_H^2 \xi_k^2}{(1-\gamma)\beta_k} \mathbb{E}[\varepsilon_k^\mu]
$$

$$
\quad + B_H L_{VV}(B_F + B_D + B_H)\alpha_k^2 \mathbb{E}[\|y_k\|^2] + B_F L_{VV}(B_F + B_D + B_H)\alpha_k^2
$$

$$
\leq \frac{(1-\gamma)\beta_k}{4} \mathbb{E}[\|y_k\|^2] + \frac{6L_V^2 \alpha_k^2}{(1-\gamma)\beta_k} \mathbb{E}[\|\nabla_\theta J_f(\pi_{\theta_k}, \phi_{\omega_k}, \hat{\mu}_k)\|^2] + \frac{6L_V^2 L_H^2 \xi_k^2}{(1-\gamma)\beta_k} \mathbb{E}[\varepsilon_k^\pi] + \frac{6L_V^2 \eta_2^2 \zeta_k^2}{(1-\gamma)\beta_k} \mathbb{E}[\varepsilon_k^\phi]
$$

$$
\quad + \left(\frac{36L_V^4 L_D^2 \zeta_k^2}{(1-\gamma)\beta_k} + \frac{12L^2 L_V^2 L_H^2 \xi_k^2}{(1-\gamma)\beta_k}\right) \mathbb{E}[\varepsilon_k^\mu] + \frac{6L_V^4 \zeta_k^2}{(1-\gamma)\beta_k} \mathbb{E}[\varepsilon_{f,k}^V] + \frac{24L_V^2 L_D^2 \alpha_k^2}{(1-\gamma)\beta_k} \mathbb{E}[\varepsilon_{l,k}^V]
$$

$$
\quad + L_{VV}(B_F + B_D + B_H)^2 \alpha_k^2 \mathbb{E}[\|y_k\|^2] + L_{VV}(B_F + B_D + B_H)^2 \alpha_k^2
$$

$$
\leq \frac{(1-\gamma)\beta_k}{2} \mathbb{E}[\|y_k\|^2] + \frac{6L_V^2 \alpha_k^2}{(1-\gamma)\beta_k} \mathbb{E}[\|\nabla_\theta J_f(\pi_{\theta_k}, \phi_{\omega_k}, \hat{\mu}_k)\|^2] + \frac{6L_V^2 L_H^2 \xi_k^2}{(1-\gamma)\beta_k} \mathbb{E}[\varepsilon_k^\pi] + \frac{6L_V^2 \eta_2^2 \zeta_k^2}{(1-\gamma)\beta_k} \mathbb{E}[\varepsilon_k^\phi]
$$

$$
\quad + \frac{16L^2 L_V^2 L_H^2 \xi_k^2}{(1-\gamma)\beta_k} \mathbb{E}[\varepsilon_k^\mu] + \frac{6L_V^4 \zeta_k^2}{(1-\gamma)\beta_k} \mathbb{E}[\varepsilon_{f,k}^V] + \frac{24L_V^2 L_D^2 \alpha_k^2}{(1-\gamma)\beta_k} \mathbb{E}[\varepsilon_{l,k}^V] + L_{VV}(B_F + B_D + B_H)^2 \alpha_k^2,
$$

where the last inequality is due to the step size condition $\frac{\zeta_k}{\xi_k} \leq \frac{LL_H}{3L_V L_D}$ and $\alpha_k \leq \min\{1, \frac{1-\gamma}{4L_{VV}(B_F+B_D+B_H)^2}\}$.

∎

# E  Details on Example Satisfying Assumption 6

**Example 1** *Consider a hierarchical mean field game in which the leader is stateless ($\mathcal{S}_l = \emptyset$), and the follower's state space $\mathcal{S}_f$, the follower's action space $\mathcal{A}$, and the leader's action space $\mathcal{B}$ all have equal cardinality $n$. Let the actions and states be indexed as $a = 1, \cdots, n \in \mathcal{A}, b = 1, \cdots, n \in$*

$\mathcal{B}, s_f = 1, \cdots, n \in \mathcal{S}$. *Suppose the transition kernel and reward function $r_l$ satisfy the conditions*

$$\mathcal{P}_f^\mu(s_f' \mid s_f, a, b) = \mathbf{1}(s_f' = 1), \qquad r_l(b, \mu) = \mathbf{1}(b = 1) + \mu(b),$$

*where $\mathbf{1}(\cdot)$ denotes the indicator function (1 if the condition holds, 0 otherwise). We drop $s_l$ as an argument from $r_l$, since the leader's state is vacuous. The follower's reward $r_f$ can be any function of $s_f, a, b$. This MFG satisfies Assumption 6 with $\eta_1 = 2, \eta_2 = 1/2$. Clearly, the leader-follower independence assumption in Cui et al. [2024] is violated as the leader's reward depends on the mean field and the follower's reward may depend on the leader's action.*

Below we justify that Example 1 indeed satisfies Assumption 6 but not the leader-follower independence assumption in Cui et al. [2024].

The mean field induced by any leader's policy is the same and satisfies the following

$$\mu^\star(\phi)(s) = \mathbf{1}(s = 1).$$

As the leader is stateless, we have

$$\Phi(\phi) = \sum_{b \in \mathcal{B}} \phi(b) r_l(b, \mu^\star(\phi)) = \sum_{b \in \mathcal{B}} \phi(b) \Big( \mathbf{1}(b = 1) + \mu^\star(\phi)(b) \Big)$$
$$= \sum_{b \in \mathcal{B}} \phi(b) \Big( \mathbf{1}(b = 1) + \mathbf{1}(b = 1) \Big) = 2\phi(1).$$

As a result,

$$\nabla_\omega \Phi(\phi_\omega) = 2 \nabla_\omega \phi_\omega(1).$$

Similarly, we have

$$J_l(\phi, \mu) = \sum_{b \in \mathcal{B}} \phi(b) \Big( \mathbf{1}(b = 1) + \mu(b) \Big) = \phi(1) + \sum_b \phi(b)\mu(b),$$

which leads to

$$\nabla_\omega J_l(\phi_\omega, \mu) \mid_{\mu = \mu^\star(\phi_\omega)} = \nabla_\omega \phi_\omega(1).$$

Since $\nabla_\omega \Phi(\phi_\omega) = 2 \nabla_\omega J_l(\phi_\omega, \mu) \mid_{\mu = \mu^\star(\phi_\omega)}$, the inequalities in Assumption 6 obviously hold with $\eta_1 = 2$ and $\eta_2 = 1/2$.

# F  Appendix: Environment Definitions

## F.1  Market Entrance

*Followers* Followers decide whether or not to buy (left) or sell (right) $A = \{\text{buy, sell}\}$ a good at each timestep, leading to the corresponding state $s_{k+1} = a_k, S = A$. The payoff for followers is based on the proportion of other agents choosing the same action:

$$r_f(a_k, b_k, s_k, \mu_k) = -\mu_k(a_k) + \text{bonus} * \mathbb{1}_{a_k = b_k},$$

where a bonus is awarded for choosing the same action as the leader, as discussed below. The transition kernel is independent of the mean field and a deterministic function of the action of the representative agent:

$$\mathcal{P}(s_{k+1} = s' \mid s_k = s, a_k = a) = \mathbb{1}_{s' = a}.$$

*Leader* We introduce a leader into this market, where the leader is trying to steer the mean field towards some desirable market state goal, for example, buying (goal = buy), selling (goal = sell), or balanced market (goal $\in_R \{\text{buy, sell}\}$), by taking actions to bonus these states $b_k \in B = \{\text{buy, sell}\}$. This can be seen as, for example, a market maker skewing prices to make certain trades more desirable, e.g., for managing their inventory. The reward for the leader is:

$$r_l(s_k, b_k, \mu_k) = \mu_k(\text{goal}) + \upsilon(b_k - \text{goal}).$$

where $0 < \upsilon \ll 1$ is a small reward shaping term, providing more immediate feedback to the leader than the delayed feedback from the mean field.

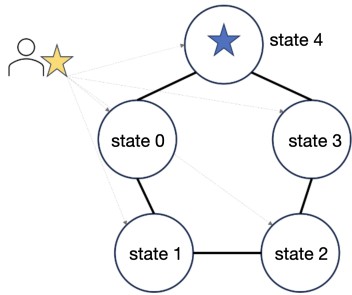

Figure 5: Beach Bar Environment. The blue stars indicates the fixed bar position, and the golden star shows the leader can add a new bar anywhere within the existing state.

### F.2 Shop position

The state space $S$ covers all the potential locations in a 1-dimensional ring, where there is a (fixed) desirable location $\star$ in one of these states, as visualized in Fig. 5.

*Followers* The followers are rewarded based on their proximity to the nearest desirable location (either $b_k$ or $\star$), as well as the crowdedness of the state they are currently in:

$$r_f(s_k, a_k, b_k, \mu_k) = \min(\text{dist}(s_k, \star), \text{dist}(s_k, b_k)) - c \times \log(\mu_k(s_k))$$

where $c \geq 0$ is a coefficient specifying the follower's preference to avoid the crowd.

*Leader* The leader is rewarded based on the business of their chosen location: $r_l(s_k, b_k, \mu_k) = \mu_k(b_k)$ e.g., more customers visiting a store can lead to higher profits, making the positioning more desirable.

### F.3 Equilibrium price

In this environment, the followers are homogeneous firms producing an identical good, where the price is determined by the (endogenous) supply-demand equilibrium around some baseline demand $d$. Each timestep, the representative firm chooses a production quantity $q_k$ and replenishment quantity $h_k$ based on their inventory level captured by the state $s_k$. The state/inventory transitions according to: $s_{k+1} = s_k - \min(q_k, s_k) + h_k$.

The followers are rewarded based on the resulting (endogenous) price and their chosen quantities as:

$$r_f(s, a, b, \mu) = (p_k - c_0)q_k - c_1 q_k^2 - c_2 h_k - (c_2 + c_3)\max(q_k - s_k, 0) - c_4 s_k \tag{89}$$

where $c_x$ are cost terms penalizing different inventory management and production capacities, and the price is determined based on the supply-demand equilibrium as $p_k = \frac{d}{E[q]}^{1/\sigma}$ where

$$E[q] = \sum_{\text{inv}} \sum_q \sum_h \mu(\text{inv}) \cdot \pi[(q, h), \text{inv}] \cdot \text{inv}$$

is the expected inventory of the mean field.

We introduce a leader into this environment, where the leader attempts to encourage firms to maintain some target inventory level $i$ by taking actions to set the inventory holding cost $c_2$. For example, a large target inventory level would encourage firms to build buffers to withstand future supply shocks, or an inventory of zero would encourage optimal production under known and predictable demands. The leader is rewarded by

$$r_F = -(E[q] - i)^2 + \upsilon[-c_2 \cdot (E[q] - i)] \tag{90}$$

where the first term is based on the inventory gap, and the second term is again a reward shaping term for more immediate feedback.

## G  Additional Experimental Results

We provide the evolution of the mean fields and resulting metrics across the environments in Figs. 6 to 8, for market entrance, shop positioning, and equilibrium price respectively.

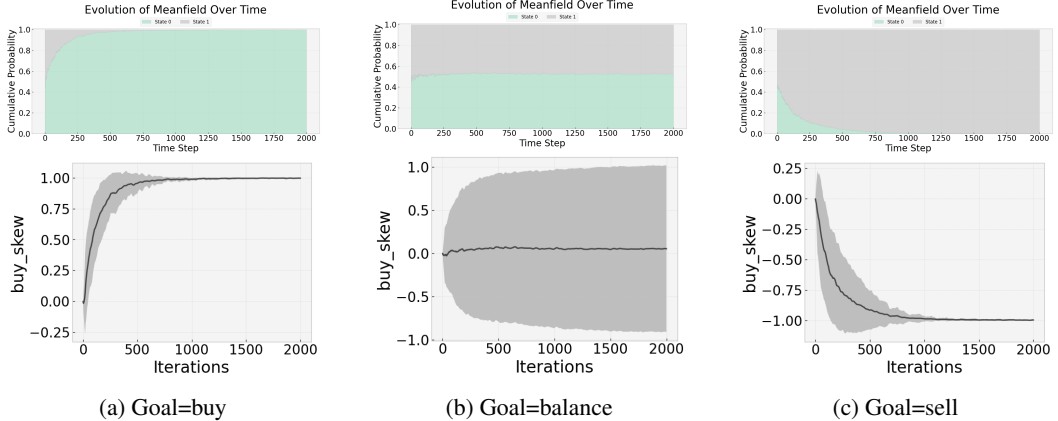

Figure 6: Market Entrance: Evolution of follower mean fields $\mu$ (top row) and resulting skew (bottom row) under different leader goals.

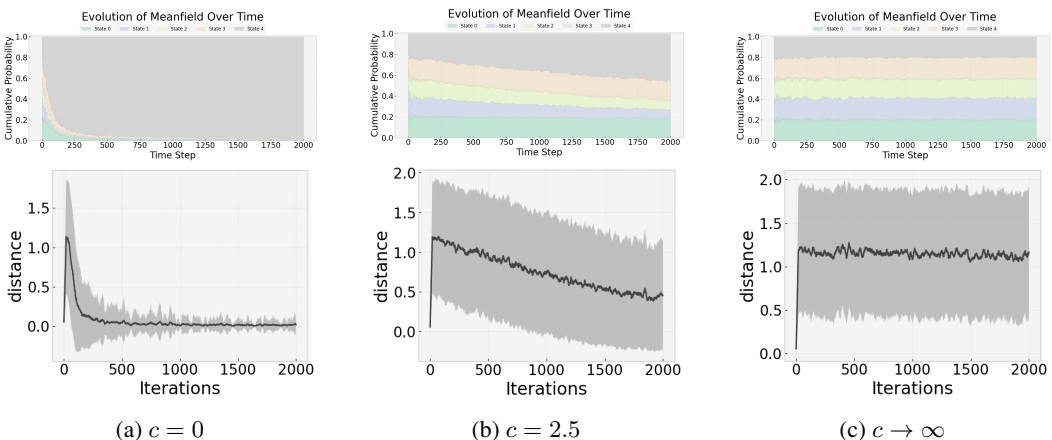

Figure 7: Beach bar: Evolution of follower mean field $\mu$ (top row) and distance between the chosen bar position $b$ (leaders action) and the fixed bar position $\star$ (bottom row) under varying crowd aversions $c$.

## H   Experimental Details

We run each approach for $20,000$ iterations (plots show every 10th iteration). Each run is repeated 30 times, with seeds set based on the run index for reproducibility.

**Proposed.** The proposed is run with $\zeta_0 = 0.5, \alpha_0 = 0.25, \beta_0 = 0.02, \xi_0 = 0.25$.

**PPO.** For the PPO implementation, we base the implementation off CleanRL (in Torch), using a batch size of 256, hidden layer shape of $(64, 64)$, learning rate of $3e - 4$, TanH activation functions for the hidden layers, and a clipping epsilon of $0.2$. ADAM is used as the optimiser (as implemented in torch).

**Nested.** To ensure fair comparisons, the nested comparisons are run with the same hyperparamaters as the proposed. However, follower (resp. leader) specific parameters are decayed at a rate based only on the number of follower (resp. leader) updates.

All approaches are run on a CPU, with Python3, and on an Amazon EC2 with R6i.large.

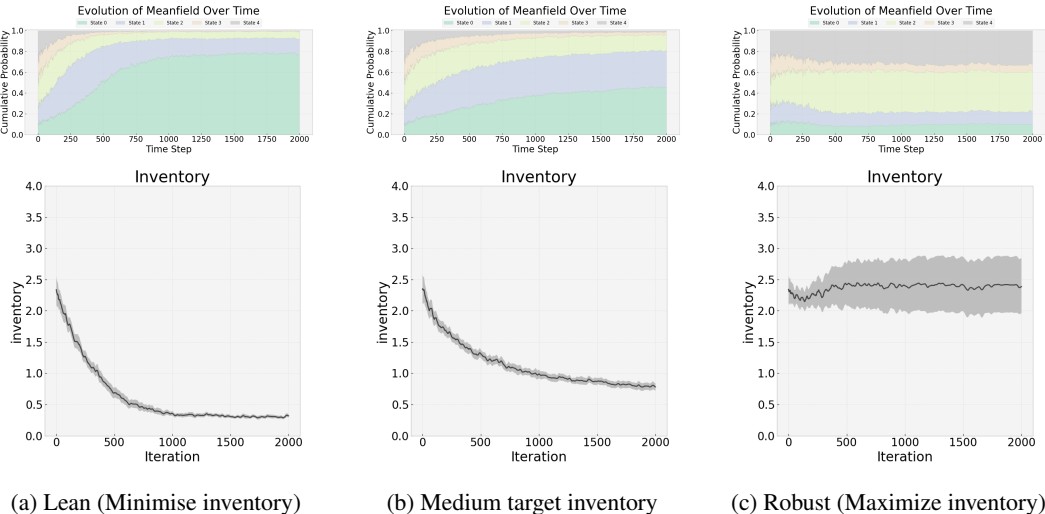

(a) Lean (Minimise inventory)    (b) Medium target inventory    (c) Robust (Maximize inventory)

Figure 8: Equilibrium pricing: Evolution of follower mean fields $\mu$ (top row) and resulting inventories (bottom row) under different leader goals.

