# OpenReview forum: "Learning in Stackelberg Mean Field Games: A Non-Asymptotic Analysis"
_NeurIPS.cc/2025/Conference — NeurIPS 2025 poster_

### Official Review · Reviewer_CXoA · 2025-07-02

**Clarity:** 4
**Significance:** 3
**Originality:** 3
**Rating:** 5
**Confidence:** 4

**Summary:**

Stackelberg mean field games (SFMGs) provide a model for studying settings in which a single "leader" takes an action and then "follower" agents react in a way modeled by a mean field capturing a collective response. This paper introduces AC-SMFG which is a single-loop actor critical algorithm that converges to a stationary point of a Stackelberg objective under a "gradient alignment" assumption that essentially implies the leader is able to improve their objective value by acting in response to the best response of the followers. This is a looser assumption than prior works which more so decouple the leaders and followers. Even with these looser assumptions, the authors establish the first non-asymptotic convergence guarantees for SMFGs. Additionally the authors provide numerical experiments demonstrating improved performance compared to baselines in mean field games such as market entrance and shop positioning.

**Questions:**

Could the authors provide an example of SMFG for which they expect the gradient alignment condition would not hold? To understand its limitations.

Could the authors clarify whether followers would be incentivized to actually use this algorithm to update their policies or do there exist settings where this would not be in their best interest?

The authors make clear their advantages to previous agent-based modeling approaches, but could they discuss what they believe their drawbacks are compared to those approaches? What would be the line in practice between using AC-SMFG and such previous models?

**Ethical Concerns:**

["NO or VERY MINOR ethics concerns only"]

**Final Justification:**

The authors clearly addressed my original questions, which increased my original confidence in my already positive review. Additionally, the authors appear to have sufficiently addressed the concerns of other reviewers and in the process raised the overall sentiment towards the paper. From my own perspective, the potential limited generality and applicability of the model setting is still a drawback, though within this assumed setting the work provides a clear contribution. Thus I will maintain my original score with increased confidence.

**Limitations:**

Limitations are mostly addressed, lingering ones are related to my questions above.

**Paper Formatting Concerns:**

None noticed.

**Quality:**

3

**Strengths And Weaknesses:**

The authors provide compelling motivating examples and an introduction to Stackelberg mean field games. Algorithm 1 AC-SMFG is well-explained and presented. Their theoretical analysis is thorough, building all the necessary steps to demonstrate their theoretical contributions of obtaining non-asymptotic convergence guarantees for this setting. Additionally the authors are careful to compare their algorithm to rates that could be achieves by considering the policy optimization problem inherent to a SFMG as a bi-level optimization problem, showing that their convergence rate improves over the baseline comparable single-loop algorithm. This involves a key technical contribution regarding the MFG satisfying the PL condition with respect to the follower's softmax policy under regularization and discuss the potential individual interest of this technical contribution. This highlights the technical difficulty inherent to proving theoretical results of their algorithm. Additionally the authors provide relatively extensive numerical results in multiple MFG settings with detailed results and provided code. Their numerical results demonstrate faster leader convergence and smoother convergence trajectories compared to baselines. For each setting they compare to both the existing mean field approaches and a MARL approach, discussing and explaining the reasons for the relatively differences of how AC-SMFG compares in each setting.

As the authors mention in Section 1.1, many previous agent-based modeling approaches to settings such as SMFGs simulate agents under complex interactions which causes them to suffer from computational inefficiencies and lack of provable guarantees. But the tradeoff this paper makes is the homogeneity of the mean field response, which although inherent to such modeling approaches is a drawback when considering real world significance. Additionally, results that impose a certain learning algorithm for both parties do come with some questions as to applicability especially for some of the mentioned settings such as optimal liquidation.

---

> ### Author Rebuttal · Authors · 2025-07-29
>
> We thank the reviewer for the time, effort, and valuable feedback on our paper. Please find our response below, which we have incorporated into the revised paper.
>
> > ...the tradeoff this paper makes is the homogeneity of the mean field response, which although inherent to such modeling approaches is a drawback when considering real world significance.
>
> **Response:** The reviewer is correct that it is a requirement of the setting of MFGs that the agents have homogeneous reward functions and exert symmetric effect on the environment. While this indeed implies that the standard (Stackelberg) MFG framework may not be directly applicable to real-world problems with heterogeneous agents, there exist many applications where the homogeneity assumption holds true (such as the ones in the economics space discussed in introduction and experiment sections) in which the MFG framework has yielded insights and a good approximation.
>
> We further point out that for problems involving heterogeneous agents, the graphon MFG framework offers a natural extension, where each agent's dynamics and reward depend on local interactions defined by a structured graph. While graphon MFG adds another layer of complexity on top of standard MFG and is beyond the scope of this paper, we believe that many components of our algorithm and analysis can carry over to the graphon setting as well.
>
>
> > Additionally, results that impose a certain learning algorithm for both parties do come with some questions as to applicability especially for some of the mentioned settings such as optimal liquidation.
>
> **Response:** We appreciate this insightful comment and have revised the paper to provide more clarification. Please find our response below.
>
> It is true that the specific algorithm proposed and analyzed in our paper is one in which the leader and follower both run actor-critic-type updates. However, we clarify that in general our framework does not require the two parties to adopt the same learning algorithm. The follower may use any algorithm (e.g. actor-critic algorithm considered in our paper, Q learning, or others) to optimize its own objectives given the leader's policy. The only requirement is that the chosen algorithm leads the follower to an (approximate) best response.
>
> To reflect this flexibility, we have deliberately structured our analysis in a modular way -- we individually establish the leader's policy convergence, mean field convergence, and follower's convergence in Propositions 1, 2, and 3, respectively. These components are then connected through a Lyapunov function that ensures joint convergence. If we switch the follower's algorithm, Propositions 1 and 2 remain unaffected, and we simply need to redo the convergence analysis of the follower. This modularity allows our framework to accommodate alternative follower dynamics and highlights the generality of our approach.
>
>
> > Could the authors provide an example of SMFG for which they expect the gradient alignment condition would not hold? To understand its limitations.
>
> **Response:** Generally speaking, the gradient alignment assumption breaks when there is competition between the leader and followers. For the assumption to hold, we need the SMFG to have at least one of the two following characteristics: 1) the leader's reward is such that $\nabla_ \mu J_ l(\phi,\mu)$ has a much smaller magnitude than $\nabla_ \phi J_ l(\phi,\mu)$, which allows $\nabla_ \phi\Phi(\phi)$ and the direct partial gradient $\nabla_ \phi J_ l(\phi,\mu)|_ {\mu=\mu^*(\phi)}$ to always form an acute angle, and 2) the indirect gradient is always roughly "aligned" with the direct partial gradient. We provide a non-trivial example (Example 1 in Appendix E) that falls into the second case and satisfies the gradient alignment assumption. Note that the leader's reward function in Example 1 has two components $r_ l(b,\mu)=\textbf{1}(b=1)+\mu(b)$. The first component is associated with the direct portion of the gradient $\nabla_ \phi\Phi(\phi)$ and the second component with the indirect portion. If we flip the second component and consider $r_ l(b,\mu)=\textbf{1}(b=1)-2\mu(b)$, the example will become one violating the gradient alignment assumption, because of the conflict between the direct and indirect partial gradients.
>
> We note that in prior work, the assumption is that the leader's reward is independent of the mean field, and that the follower's reward and transition are independent of the leader's action. Therefore, the notion of gradient alignment we consider here is more general in that it permits these quantities to be correlated. Though Assumption 6 is a relaxation, we agree that it may not hold in many practical problems, and even checking its validity can be challenging. This provides an important motivation for us to carry out extensive numerical simulations in environments that potentially violate the assumption. The environments in Section 5 cannot be verified to satisfy the gradient alignment condition, but the proposed algorithm still outperforms existing methods, demonstrating empirical effectiveness even in situations where the current analysis does not strictly apply.
>
>
> > Could the authors clarify whether followers would be incentivized to actually use this algorithm to update their policies or do there exist settings where this would not be in their best interest?
>
> **Response:** The follower's part of the proposed algorithm is that it uses an actor-critic algorithm to maximize its objective, in response to the shifting leader's policy and mean field, and this is indeed in the best interest of the follower. However, we note that a Stackelberg MFG, by definition, is a problem in which the goal is to maximize the leader’s objective. The most natural application of this framework is one where a central authority (e.g. government) uses an SMFG algorithm to determine a taxation rule, or where a large institutional investor uses an SMFG algorithm to determine a liquidation strategy, while accounting for the responses of the followers. The follower's objective is not what the SMFG formulation ultimately aims to optimize.
>
>
> > The authors make clear their advantages to previous agent-based modeling approaches, but could they discuss what they believe their drawbacks are compared to those approaches? What would be the line in practice between using AC-SMFG and such previous models?
>
> **Response:** We thank the reviewer for suggesting a discussion of the potential drawbacks, which we will incorporate into the revised version of the paper. We are aware of two types of practical scenarios where an agent-based modeling (ABM) approach may be more naturally suited.
>
> First, as the reviewer noted in an earlier comment, ABM may offer greater flexibility in modeling problems involving heterogeneous agents. The standard SMFG framework assumes symmetry among followers by design. While graphon-based extensions can capture heterogeneity, they come with the cost of additional complexity and the need to specify the interactions between followers by a graph.
>
> Second, MFGs are approximations of $N$-player symmetric Markov games, with the approximation error diminishing as $N$ increases. If the problem of interest involves only a small number of followers (e.g. $N<10$), the approximation error may be non-negligible, making an ABM approach more appropriate in such cases.
>
> Both ABM and SMFG approaches have their distinct advantages and challenges. We believe they should be seen as one complimenting the other in providing solutions and insights into the analysis of the challenging framework of multi-agent decision making.

---

> > ### Comment · Reviewer_CXoA · 2025-08-04
> >
> > I thank the reviewers for their thorough response. I feel more confident in my originally positive assessment, especially the clarifying distinction between which learning algorithm the followers may use and I thank the reviewers for the example of the gradient alignment condition failing. I also appreciate the discussion as to potential drawbacks compared to other approaches and for mentioning they would include in the next draft. I will stick with my original positive assessment.

---

> > > ### Author Response · Authors · 2025-08-04
> > >
> > > We thank the reviewer once again for the feedback and questions, especially those regarding the followers' incentive and learning dynamics. Your input has greatly helped us improve the quality of the work.

---

### Official Review · Reviewer_pcnH · 2025-07-02

**Clarity:** 4
**Significance:** 3
**Originality:** 3
**Rating:** 5
**Confidence:** 2

**Summary:**

This paper proposes a learning algorithm for Stackelberg mean field games where a large number of followers follow a single leader. The proposed method is an actor-critic model that utilizes two trajectories to estimate the occupancy measure and the mean field distribution. Under the key gradient alignment that allows the improving gradients updates for the leader, the proposed method has a finite-sample convergence rate that improves upon the existing results for general bilevel optimization under convexity assumptions.

**Questions:**

-Please explain the necessity of the entropy term. Is this term common in literature? Especially for the papers compared for the convergence rates. If the compared papers, does not have this term, I suggest the authors to make this an assumption or greatly emphasize that their setting is different than the others’.

-When does the gradient alignment assumption hold? Is it possible to make a conclusion on the validity of the assumption based on the objective functions?

**Ethical Concerns:**

["NO or VERY MINOR ethics concerns only"]

**Final Justification:**

The authors have addressed my concerns in their rebuttal. I will keep my original score of 5 (Accept).

**Limitations:**

Yes, but the discussion on limitations could be extended.

**Quality:**

3

**Strengths And Weaknesses:**

Strengths:
- Despite its technically dense content, the paper is written excellently. The existing methods, their weaknesses, and the proposed solution are explained well.
- Under some assumptions (see the relevant weakness and question), the theoretical result improves upon the existing SMFG methods that only provide asymptotic convergence guarantees. Additionally, the proposed method improves upon the bilivel optimization methods that are applicable to the SMFG setting with additional assumptions.
-Theoretical findings are supported by various numerical experiments.

Weaknesses:
-The problem setting has an unexplained addition compared to the existing formulations: The followers optimize an entropy regulated objective. To the best of my understanding, this term does not exist in the compared works (for example [1] and [2]). Furthermore, the followers are called ‘rational’, given that entropy-regularized objectives are very common in representing boundedly rational agents. I think that ‘rational’ does not accurately describe the followers considered in this paper.
- The assumption on the gradient alignment may be required to derive the theoretical result. However, the authors should discuss in more detail when this assumption does not hold. I believe that when the leader and followers have conflicting objectives, for example a zero-sum setting, this assumption will break.


[1] Cui, Kai, et al. "Learning discrete-time major-minor mean field games." Proceedings of the AAAI Conference on Artificial Intelligence. Vol. 38. No. 9. 2024.

[2] Guo, Xin, Anran Hu, and Jiacheng Zhang. "Optimization frameworks and sensitivity analysis of Stackelberg mean-field games." arXiv preprint arXiv:2210.04110 (2022).

---

> ### Author Rebuttal · Authors · 2025-07-29
>
> We thank the reviewer for the time, effort, and valuable feedback on our paper. Please find our response below, which we have incorporated into the revised paper.
>
> 1)  >* The problem setting has an unexplained addition compared to the existing formulations: The followers optimize an entropy regulated objective. To the best of my understanding, this term does not exist in the compared works (for example [1] and [2]).
> >* Please explain the necessity of the entropy term. Is this term common in literature? Especially for the papers compared for the convergence rates. If the compared papers, does not have this term, I suggest the authors to make this an assumption or greatly emphasize that their setting is different than the others’.
>
> **Response:** Our work adds regularization to ensure that Assumption 4 (optimality-consistency operator contraction condition) holds. The regularization (or the related Assumption 4) is not introduced in our paper to study Stackelberg MFGs, but rather is a common assumption inherited from the (standard, non-hierarchical) MFG literature. Below we explain how and why we adopt this assumption, and clarify its relationship to prior work.
>
> In general not all MFGs (even without the leader) admit polynomial time solutions, a fact proved in Yardim et al. [2024]. This implies that only certain special subclasses of MFGs are tractable. Currently, three subclasses are known to admit efficient solutions. The first, and most widely studied, consists of MFGs that satisfy the contraction condition (guaranteed by a sufficient regularization). Papers making the contraction assumption includes our work, as well as Mao et al. [2022], Zaman et al. [2023], and others. The second solvable subclass includes MFGs satisfying the Lasry-Lions monotonicity condition, which can also be guaranteed if a sufficient regularization is added. Existing works making the Lasry-Lions monotonicity assumption include the paper [2] (Cui et al. [2024]) that the reviewer mentioned. The third subclass is the herding class introduced in Zeng et al. [2024]. We provide a comparison on the subclasses of solvable MFGs and the corresponding papers in the table below.
> | Existing Work     | Assumption | Achievable Through Entropy Regularization |
> |----------|-----------|-----------|
> | Guo et. al [2019]| Contraction | Yes |
> | Xie et. al [2021]| Contraction | Yes |
> | Mao et. al [2022]| Contraction | Yes |
> | Zaman et. al [2022]| Contraction | Yes |
> | Yardim et. al [2023]| Contraction | Yes |
> | Zhang et al. [2023]| Lasry-Lions Monotonicity | Yes |
> | Cui et al. [2024]| Lasry-Lions Monotonicity | Yes |
> | Zeng et al. [2025]| Herding Condition | Unknown |
>
> Current knowledge suggests that the three subclasses overlap but do not subsume one another. We consider the contraction assumption due to its common use in the literature, but believe a modified version of our analysis can handle the other two subclasses as well.
> The paper [1] suggested by the reviewer (Guo et al. [2022]) does not make any of the aforementioned assumptions, as it only investigates the SMFG problem structure and performs general sensitivity analysis, and does not propose/analyze any specific algorithm.
>
> 2) > Furthermore, the followers are called ‘rational’, given that entropy-regularized objectives are very common in representing boundedly rational agents. I think that ‘rational’ does not accurately describe the followers considered in this paper.
>
> We agree with the comment. While we deal with the low temperature (regularization weight) case, as the temperature increases, this can be seen as modeling bounded rational behavior in the spirit of Quantal Response Equilibrium. However, since the level of regularization is relatively low (compared to what would typically be used in bounded rationality work), we feel it is still valid to consider the followers rational, as is common in existing MFG literature. Bounded rational behavior in MFGs (even without the leader or hierarchical structure) represents a compelling direction for future work, which we have been investigating in parallel to this paper.
>
>
>  3) >* The assumption on the gradient alignment may be required to derive the theoretical result. However, the authors should discuss in more detail when this assumption does not hold. I believe that when the leader and followers have conflicting objectives, for example a zero-sum setting, this assumption will break.
> >* When does the gradient alignment assumption hold? Is it possible to make a conclusion on the validity of the assumption based on the objective functions?
>
> **Response:**  The reviewer is correct that the gradient alignment assumption breaks when there is competition between the leader and followers. For the assumption to hold, we need the SMFG to have at least one of the two following characteristics:
>
> 1) The leader's reward is such that $\nabla_ \mu J_ l(\phi,\mu)$ has a much smaller magnitude than $\nabla_ \phi J_ l(\phi, \mu)$, which allows $\nabla_ \phi \Phi(\phi)$ and $\nabla_ \phi J_ l (\phi,\mu)| _{\mu=\mu^ *(\phi)}$ to always form an acute angle.
>
> 2) The indirect gradient is always roughly "aligned" with the direct partial gradient. We provide a non-trivial example (Example 1 in Appendix E) that falls into the second case and satisfies the gradient alignment assumption. Note that the leader's reward function in Example 1 has two components $r_ l(b, \mu)=\textbf{1}(b=1)+\mu(b)$. The first component is associated with the direct portion of the gradient $\nabla_ \phi\Phi(\phi)$ and the second component with the indirect portion. If we flip the second component and consider $r_ l(b,\mu)=\textbf{1}(b=1)-2\mu(b)$, the example will become one violating the gradient alignment assumption, because of the conflict between the direct and indirect partial gradients.
>
> The assumption imposed in prior work is that the leader's reward is independent of the mean field, and that the follower's reward and transition are independent of the leader's action. Our works relaxes the assumption of independence between the leader and the followers, by allowing a form of coupling. This extends the class of problems that can be theoretically analyzed. While it is difficult to verify in practice whether the given problem instance satisfies this assumption, we show in simulations that the proposed algorithm performs well empirically, even when the assumption is explicitly violated or when we cannot verify if it is satisfied.
>
>
> References
>
> Guo, X., Hu, A., Xu, R. and Zhang, J., 2019. Learning mean-field games. Advances in neural information processing systems, 32.
>
> Mao, W., Qiu, H., Wang, C., Franke, H., Kalbarczyk, Z., Iyer, R. and Basar, T., 2022. A mean-field game approach to cloud resource management with function approximation. Advances in Neural Information Processing Systems, 35, pp.36243-36258.
>
> Yardim, B., Goldman, A. and He, N., 2024. When is Mean-Field Reinforcement Learning Tractable and Relevant?. arXiv preprint arXiv:2402.05757.
>
> Zhang, F., Tan, V., Wang, Z. and Yang, Z., 2023. Learning regularized monotone graphon mean-field games. Advances in Neural Information Processing Systems, 36, pp.67297-67308.

---

> > ### Comment · Reviewer_pcnH · 2025-08-01
> >
> > I thank the authors for addressing my comments regarding the assumptions and the setting. I find the additional information valuable. I believe that incorporating some of this discussion into the paper will justify their assumptions better.
> >
> > Additionally, I think that calling the follower 'rational' is still technically not precise. I recommend that the authors change the description or at least mention that such followers become rational with decreasing levels of temperature.

---

> > > ### Author Response · Authors · 2025-08-01
> > >
> > > We appreciate the reviewer's suggestion on incorporating the discussion above into the paper. We have added clarifying remarks regarding bullet point #3 and will add a discussion of contraction, Lasry-Lions monotonicity, and herding condition in the Related Literature section. We would like to thank the reviewer again for taking the time to evaluate our paper and providing valuable feedback.

---

> > > ### Author Response · Authors · 2025-08-01
> > >
> > > We also agree that it is not rigorous to call the follower's exactly rational, and have revised to paper to indicate that adding regularization translates to bounded rationality, with the rationality level increasing as the temperature drops.

---

### Official Review · Reviewer_XdJs · 2025-07-03

**Clarity:** 3
**Significance:** 3
**Originality:** 3
**Rating:** 5
**Confidence:** 3

**Summary:**

This paper investigates the strategy optimization problem in Stackelberg mean-field games (SMFG). The authors propose the AC-SMFG algorithm, a single-loop actor-critic algorithm that operates based on continuously generated Markov samples. Under a novel "gradient alignment" assumption, the paper not only proves that the algorithm converges to a stationary point of the Stackelberg objective at a rate of $\tilde{O}(k^{-1/2})$ within finite time and finite samples but also validates its effectiveness through comprehensive numerical experiments.

**Questions:**

Could the authors provide a paragraph introducing the techniques used in the convergence analysis of the AC-SMFG algorithm?

**Ethical Concerns:**

["NO or VERY MINOR ethics concerns only"]

**Final Justification:**

The authors' responses in their rebuttal have adequately addressed my concerns, leading me to decide to maintain my original score.

**Limitations:**

See weakness.

**Paper Formatting Concerns:**

There are no formatting issues.

**Quality:**

3

**Strengths And Weaknesses:**

Strengths:
（1）This work establishes the first non-asymptotic convergence guarantee for SMFG algorithms.
（2）The AC-SMFG algorithm exhibits a significantly faster non-asymptotic convergence rate than previous SMFG algorithms, achieved under more challenging problem settings where the lower-level problem of the bi-level optimization is non-strongly convex.

Weaknesses:
The authors should expand the discussion on the "gradient alignment" (Assumption 6). For instance, although $\nabla_w \Phi(\phi_w)$ and $\nabla_w J_l(\phi_w, \mu)_{\mid\mu=\mu^* (\phi_w)}$ differ by a derivative operation with respect to $\mu^*(\phi_w)$, it is necessary to analyze the impact of this difference in specific problem instances and whether the gradient alignment assumption still holds. As mentioned in the paper, the problem setting is more challenging than previous works. Thus, a paragraph discussing the techniques used in the convergence analysis should be added after Theorem 1 to highlight the improvements over prior research. The current description in the Introduction—i.e., developing an analytical argument to handle bi-level optimization with lower-level Polyak-Lojasiewicz (PL) condition—is too vague.

---

> ### Author Rebuttal · Authors · 2025-07-29
>
> We thank the reviewer for the time, effort, and valuable feedback on our paper. Please find our response below, which we have incorporated into the revised paper.
>
> >The authors should expand the discussion on the "gradient alignment" (Assumption 6) ... it is necessary to analyze the impact of this difference in specific problem instances and whether the gradient alignment assumption still holds.
>
> **Response:** We appreciate the reviewer's comment on more discussion on Assumption 6. The leader's objective $J_ l(\phi,\mu^\star(\phi))$ has two components. The first is the partial gradient $\nabla_ \phi J_ l(\phi,\mu)\mid_ {\mu=\mu^\star(\phi)}$ directly due to the leader's policy, and the second is $\frac{\partial J_ l}{\partial \mu^\star(\phi)}\cdot\frac{\partial \mu^\star(\phi)}{\partial \phi}$, which is the indirect component due to the effect of leader's policy on the mean field. The indirect partial gradient may be arbitrary in general SMFG instances, causing the gradient alignment assumption to break. For the assumption to hold, we need the SMFG to have at least one of the two following characteristics: 1) the leader's reward is such that $\nabla_ \mu J_ l(\phi,\mu)$ has a much smaller magnitude than $\nabla_ \phi J_ l(\phi,\mu)$, which allows $\nabla_ \phi\Phi(\phi)$ and the direct partial gradient $\nabla_ \phi J_ l(\phi,\mu)|_ {\mu=\mu^*(\phi)}$ to always form an acute angle, and 2) the indirect gradient is always roughly "aligned" with the direct partial gradient. We provide a non-trivial example in Appendix E that falls into the second case and satisfies the gradient alignment assumption, but violates a comparable leader-follower independence assumption introduced in the recent related work Cui et al. [2024].
>
> Though Assumption 6 is a relaxation of existing assumptions, we agree that it may not always hold in practical problems. This provides an important motivation for us to carry out extensive numerical simulations in environments that potentially violate the assumption. The environments in Section 5 cannot be verified to satisfy the gradient alignment condition, but the proposed algorithm still outperforms existing methods, demonstrating empirical effectiveness even in situations where the current analysis does not strictly apply.
>
>
> > * As mentioned in the paper, the problem setting is more challenging than previous works. Thus, a paragraph discussing the techniques used in the convergence analysis should be added after Theorem 1 to highlight the improvements over prior research. The current description in the Introduction—i.e., developing an analytical argument to handle bi-level optimization with lower-level Polyak-Lojasiewicz (PL) condition—is too vague.
> >* Could the authors provide a paragraph introducing the techniques used in the convergence analysis of the AC-SMFG algorithm?
>
> **Response:** We thank the reviewer for suggesting an in-depth discussion of the main technical innovation enabling the fast convergence rate and treatment of lower-level PL condition. Please find the discussion below. We will add a paragraph on the same in Section 4.1 after Theorem 1.
>
> 1) In bi-level optimization and two-time-scale stochastic approximation literature, the lower-level objective is typically assumed to be strongly convex (for example, see Hong et al. [2023]). However, in our case, it is non-convex and only satisfies the PL condition. Analyzing single-loop bi-level algorithms without strong convexity is a significant challenge, and handling this challenge is a first main technical innovation of our paper, as we now explain.
>
>     To analyze a bi-level optimization algorithm, we need to bound the residual of the lower-level decision variable (in our context, the follower's policy) and show its iteration-wise reduction. If the lower-level objective $J_ f$ were strongly convex, we would select the residual to be $\tilde\epsilon_ k^\pi = ||\pi_ {\theta_ k}-\pi^\star(\phi_ {\omega_ k},\hat{\mu}_ k)||^2$ and show that $\tilde\epsilon_ {k+1}^\pi-\tilde\epsilon_ k^\pi$ is approximately negative. As the learning target shifts from iteration $k$ to $k+1$, bounding $\tilde\epsilon_ {k+1}^\pi-\tilde\epsilon_ k^\pi$ requires controlling the drift $||\pi^\star(\phi_ {\omega_ {k+1}},\hat{\mu}_ {k+1})-\pi^\star(\phi_ {\omega_ k},\hat{\mu}_ k)||^2$, which can be easily bounded by $\mathcal{O}(||\phi_ {\omega_ {k+1}}-\phi_ {\omega_ {k}}||^2+||\hat{\mu}_ {k+1}-\hat{\mu}_ {k}||^2)$ under the Lipschitz continuity of $\pi^\star$. However, without strong convexity, we cannot consider the residual $\tilde\epsilon_ k^\pi$ (as $\pi^\star$ may not be unique in general) and instead employ $\epsilon_ k^\pi$ defined in Eq.(15), a residual in the value function space. We would like to bound the learning target shift $J_ f(\pi^\star(\phi_ {\omega_ {k+1}},\hat\mu_ {k+1}),\phi_ {\omega_ {k+1}},\hat\mu_ {k+1})-J_ f(\pi^\star(\phi_ {\omega_ k},\hat\mu_ k),\phi_ {\omega_ k},\hat\mu_ k)$, preferably still by $\mathcal{O}(||\phi_ {\omega_ {k+1}}-\phi_ {\omega_ {k}}||^2+||\hat{\mu}_ {k+1}-\hat{\mu}_ {k}||^2)$ to avoid rate deterioration. However, the Lipschitz continuity of $J_ f$ only yields a loose bound on the order of $\mathcal{O}(||\phi_ {\omega_ {k+1}}-\phi_ {\omega_ {k}}||+||\hat{\mu}_ {k+1}-\hat{\mu}_ {k}||)$, where the norms are not squared. Detailed in Proposition 3, we develop a careful error decomposition scheme to break down the residual difference $\epsilon_ {k+1}^\pi-\epsilon_ k^\pi$, and tightly bound the decomposed terms based on the PL condition, thereby avoiding the suboptimal bound implied by naive application of the Lipschitz property of $J_ f$.
>
> 2) The technique highlighted above allows us to achieve the same convergence rate in SMFGs as in bi-level optimization with lower-level strong convexity. We further improve upon the rate in Hong et al. [2023], leveraging a recent advance in the multi-time-scale stochastic approximation literature. Specifically, Han et al. [2024] shows that a faster convergence rate can be achieved under lower-level strong convexity, if the lower-level learning target (equivalent to operators $\mu^\star,\pi^\star$ in our context) has Lipschitz gradients. Our work adapts and extends the argument to the case where the lower-level objective is nonconvex.

---

> > ### Comment · Reviewer_XdJs · 2025-08-02
> >
> > I appreciate the authors' detailed response. All points raised in my initial review appear to have been adequately addressed. Furthermore, I acknowledge the authors’ consideration of my suggestions. Consequently, I maintain my original rating of the paper.

---

> > > ### Author Response · Authors · 2025-08-02
> > >
> > > We thank the reviewer once again for the positive rating and for the time and effort spent on evaluating the work and providing important feedback.

---

### Official Review · Reviewer_qsnQ · 2025-07-03

**Clarity:** 3
**Significance:** 3
**Originality:** 2
**Rating:** 4
**Confidence:** 3

**Summary:**

This paper tackles the problem of computing optimal leader policies in Stackelberg mean field games—an important framework for modeling strategic interactions in settings with a committed policy designer and a large population of followers, with applications in both public and private policy (e.g., economic platforms, government interventions).

A central challenge in this setting is that each leader policy induces a new equilibrium in the follower mean field game, typically requiring a nested-loop optimization: the follower policy and mean field must converge before the leader can update. To address this, the paper adopts a single-loop algorithm that jointly updates the leader, follower, and mean field using carefully tuned step sizes, under appropriate assumptions on agent objectives. While the single-loop structure itself is not novel, the key technical contribution lies in relaxing the commonly assumed strong convexity of the follower’s objective. The proposed method is empirically evaluated against nested-loop baselines and MARL approaches across several benchmark mean field game environments.

**Questions:**

- In the main text, you refer to the single-loop algorithms by Hong et al. as the foundation that your work generalizes. However, Appendix E presents examples that seem more directly related to the framework of Cui et al., whose assumptions appear more restrictive than those in Hong et al. I assume that extending the approach of Hong et al. to Stackelberg mean field games is non-trivial, and that your contribution is better viewed as a generalization of Cui et al.’s work—rather than Hong et al.’s. Could you kindly clarify this point?
- I may have missed this in the paper, but could you elaborate on the rationale for using a tabular softmax policy parameterization? Is this choice required for the theoretical analysis, or was it made primarily for experimental convenience?
- In the equilibrium pricing game, it appears that ADAGE outperforms your proposed method. Could you comment on why this might be the case, and whether all baselines—including ADAGE—are converging to equilibrium in your experiments?

**Ethical Concerns:**

["NO or VERY MINOR ethics concerns only"]

**Final Justification:**

The authors’ rebuttal has clarified the issues I had raised. I am therefore leaning toward recommending acceptance.

**Limitations:**

Yes

**Quality:**

3

**Strengths And Weaknesses:**

Strengths:
- The paper tackles a timely and important problem in policy design, contributing an algorithmic improvement that is both technically sound and relevant.
- The contribution is clear and the authors also provide example of settings that can be considered thanks to this technical generalization
- The writing is clear, and the experimental evaluation is thoughtful, with careful analysis of the learned policies.

Weaknesses:
- The technical novelty of the proposed algorithm appears incremental in light of prior work by Hong et al. on single-loop methods.
- While the experiments are well-executed, they are conducted in relatively simple environments and are restricted to tabular softmax policies, which limits the generalizability of the results.
- The formulation assumes a well-posed leader problem, including the existence and uniqueness of the induced mean field equilibrium—guaranteed only through sufficiently large regularization weights. Although common in the literature, this assumption may limit the method’s practical applicability.

---

> ### Author Rebuttal · Authors · 2025-07-29
>
> We thank the reviewer for the time, effort, and valuable feedback on our paper. Please find our response below, which we have incorporated into the revised paper.
>
> > The technical novelty of the proposed algorithm appears incremental in light of prior work by Hong et al. on single-loop methods.
>
> **Response:** Our analysis differs from that of Hong et al. [2023] in three key aspects, presented below in order of technical novelty and significance. We have revised the paper to include a detailed discussion of these distinctions in Section 4.1.
>
> 1) The lower-level objective is strongly convex in Hong et al. [2023], but is non-convex and only satisfies the PL (Polyak-Lojasiewicz) condition in our case. Analyzing single-loop bi-level algorithms *without* strong convexity is a significant challenge, and handling this challenge is a main technical innovation of our paper, as we now explain.
>
>     To analyze a bi-level optimization algorithm, we need to bound the residual of the lower-level decision variable (in our context, the follower's policy) and show its iteration-wise reduction. If the lower-level objective $J_ f$ were strongly convex, we would select the residual to be $\tilde\epsilon_ k^\pi = ||\pi_ {\theta_ k}-\pi^\star(\phi_ {\omega_ k},\hat{\mu}_ k)||^2$ and show that $\tilde\epsilon_ {k+1}^\pi-\tilde\epsilon_ k^\pi$ is approximately negative. As the learning target shifts from iteration $k$ to $k+1$, bounding $\tilde\epsilon_ {k+1}^\pi-\tilde\epsilon_ k^\pi$ requires controlling the drift $||\pi^\star(\phi_ {\omega_ {k+1}},\hat{\mu}_ {k+1})-\pi^\star(\phi_ {\omega_ k},\hat{\mu}_ k)||^2$, which can be easily bounded by $\mathcal{O}(||\phi_ {\omega_ {k+1}}-\phi_ {\omega_ {k}}||^2+||\hat{\mu}_ {k+1}-\hat{\mu}_ {k}||^2)$ under the Lipschitz continuity of $\pi^\star$.
>
>     However, without strong convexity, we cannot consider the residual $\tilde\epsilon_ k^\pi$ (as $\pi^\star$ may not be unique in general) and instead employ $\epsilon_ k^\pi$ defined in Eq.(15), a residual in the value function space. We would like to bound the learning target shift $J_ f(\pi^\star(\phi_ {\omega_ {k+1}},\hat\mu_ {k+1}),\phi_ {\omega_ {k+1}},\hat\mu_ {k+1})-J_ f(\pi^\star(\phi_ {\omega_ k},\hat\mu_ k),\phi_ {\omega_ k},\hat\mu_ k)$, preferably still by $\mathcal{O}(||\phi_ {\omega_ {k+1}}-\phi_ {\omega_ {k}}||^2+||\hat{\mu}_ {k+1}-\hat{\mu}_ {k}||^2)$ to avoid rate deterioration. However, the Lipschitz continuity of $J_ f$ only yields a loose bound on the order of $\mathcal{O}(||\phi_ {\omega_ {k+1}}-\phi_ {\omega_ {k}}||+||\hat{\mu}_ {k+1}-\hat{\mu}_ {k}||)$, where the norms are not squared. Detailed in Proposition 3, we develop a careful error decomposition scheme to break down the residual difference $\epsilon_ {k+1}^\pi-\epsilon_ k^\pi$, and tightly bound the decomposed terms based on the PL condition, thereby avoiding the suboptimal bound implied by naive application of the Lipschitz property of $J_ f$.
>
> 2) The technique highlighted above allows us to achieve the same convergence rate in SMFGs as in bi-level optimization with lower-level strong convexity. We further improve upon the rate in Hong et al. [2023], leveraging a recent advance in the multi-time-scale stochastic approximation literature. Specifically, Han et al. [2024] shows that a faster convergence rate can be achieved under lower-level strong convexity, if the lower-level learning target (equivalent to operators $\mu^\star,\pi^\star$ in our context) has Lipschitz gradients. Our work adapts and extends the argument to the case where the lower-level objective is nonconvex.
>
> 3) Third, the analysis in Hong et al. deals with a bi-level optimization problems with two decision variables. We describe our problem as a bi-level problem for the ease of understanding, but we in fact have to deal with three decision variables on three levels (leader's policy, follower's policy, mean field). Although the argument used to analyze two levels may carry over in principle, actually implementing the analysis in the presence of a third level entails significantly greater complexity and effort.
>
> > While the experiments are well-executed, they are conducted in relatively simple environments and are restricted to tabular softmax policies, which limits the generalizability of the results.
>
> **Response:** We agree that some experiments associated with MFGLib are simplistic; however, worth mentioning is the fact that this collection of previously constructed environments must be extended in a particular manner to incorporate leader-follower dynamics, which is itself a contribution of this work.
>
> In addition, we note that we conducted the Equilibrium Price experiment both in the tabular setting and under function approximation. Figure 2c) shows the results in the tabular setting, while Figure 4 shows the results when neural networks are used to represent the leader's policy. Please kindly refer to Remark 2 for a discussion on how to make our algorithm compatible with large state/action space and general function approximations, including neural networks.
>
> > In the main text, you refer to the single-loop algorithms by Hong et al. as the foundation that your work generalizes. However, Appendix E presents examples that seem more directly related to the framework of Cui et al., whose assumptions appear more restrictive than those in Hong et al. I assume that extending the approach of Hong et al. to Stackelberg mean field games is non-trivial, and that your contribution is better viewed as a generalization of Cui et al.’s work—rather than Hong et al.’s. Could you kindly clarify this point?
>
> **Response:** Our paper closely relates to Hong et al. in that we also study a bi-level optimization (loosely speaking) and analyze the convergence of a single-loop algorithm. However, from a technical perspective, we would not consider our work an extension of Hong et al. [2023], as the innovations discussed above to handle lower-level non-convexity and to achieve a faster convergence rate make the key technical analysis substantially distinct. The assumptions made in Hong et al. [2023] and our work may not exactly match, as they study a general optimization framework, and we specifically consider SMFGs and impose problem-specific assumptions.
>
> As the reviewer correctly pointed out, our assumptions align more closely with those in Cui et al. [2024], which also studies MFGs with a hierarchical structure.
>
> > I may have missed this in the paper, but could you elaborate on the rationale for using a tabular softmax policy parameterization? Is this choice required for the theoretical analysis, or was it made primarily for experimental convenience?
>
> **Response:** The tabular softmax parameterization is only for the purpose of theoretical analysis. The algorithm itself can be implemented with any function approximation, which we discuss in detail in Remark 2. In fact, we have conducted the Equilibrium Price experiment both in the tabular setting and with function approximation.
>
> We consider the tabular softmax parameterization in our analysis, so that we can adapt the innovations by Mei et al. [2020] on establishing the non-uniform Polyak-Lojasiewicz condition for the follower's objective. This condition allows us to bound the follower's optimality gap by its gradient norm.
>
>
> > In the equilibrium pricing game, it appears that ADAGE outperforms your proposed method. Could you comment on why this might be the case, and whether all baselines—including ADAGE—are converging to equilibrium in your experiments?
>
> **Response:** Yes, they reach equilibrium in the sense that the leader’s and every follower’s behavior converge to a stationary point and the magnitude of their updates tends to zero over time.
>
> ADAGE indeed gets a slightly higher leader’s reward in the Equilibrium Pricing environment (Fig 2c). This environment is particularly challenging and exceeds the representational capacity of purely tabular methods, as the action space of the leader is continuous [-1,1], as we explain in Remark 2. ADAGE leverages function approximation, by parameterizing the leader’s and followers’ policies by a neural network, whereas the proposed algorithm shown in Figure 2c) does not use function approximation but rather uses a tabular policy with the leader's action space discretized into {-1, -0.5, 0, 0.5, 1}. The loss of precision due to the discretization may be a main cause of the performance gap.
>
> We discuss in Remark 2 how function approximation can be integrated with the proposed SMFG approach, and conduct an additional experiment in the Equilibrium Pricing environment where we parameterize the leader's policy by a neural network. The results are shown in Fig 4. From Fig 4a, we see a much faster convergence rate for the proposed algorithm, showing the benefits of integrating function approximation in complex environments.
>
> We thank the reviewer for the question and have added the discussion above to the revised version of our paper.
>
> References
>
> Yardim, B., Goldman, A. and He, N., 2024. When is Mean-Field Reinforcement Learning Tractable and Relevant?. arXiv preprint arXiv:2402.05757.
>
> Zhang, F., Tan, V., Wang, Z. and Yang, Z., 2023. Learning regularized monotone graphon mean-field games. Advances in Neural Information Processing Systems, 36, pp.67297-67308.

---

### Note · Authors · 2025-08-13

We sincerely thank the reviewers for their positive assessment and insightful feedback, which have importantly helped us improve the clarity and presentation of the work. We are pleased that our responses have addressed all questions and concerns. Below we provide a summary of the main changes made to the initial submission.

1) In Section 4.1 after the presentation of Theorem 1, we sketch the main technical innovations that allowed us to derive the theorem.

2) We move Remark 2 to strengthen its connection to the algorithm presentation and add further remarks clarifying that the algorithm can be run with any function approximation and that we consider tabular softmax parameterization just for the purpose of mathematical analysis.

3) We clarify the setting of experiments, and that Figure 2 contains results in the tabular setting and Figure 4 is under function approximation.

4) We provide more intuition of the gradient alignment assignment with examples illustrating the cases where it holds and does not hold.

5) We add a discussion of the known classes of solvable MFGs in the introduction, covering papers that impose the contraction condition, Lasry-Lions monotonicity, and herding condition.

6) We discuss that our analytical framework does not restrict the leader and followers to take the same learning algorithm, and our analysis is designed to be modular -- we individually establish the leader's policy convergence, mean field convergence, and follower's convergence in Propositions 1, 2, and 3; if we switch the learning algorithm of the follower, Propositions 1 and 2 remain unaffected, and we simply need to redo the convergence analysis of the follower.

7) In the conclusion, we add a discussion comparing the advantages and limitations of modeling real-world leader–follower decision-making problems using either a Stackelberg MFG framework or an agent-based modeling approach.

Once again, we thank the reviewers for the insightful feedback, and for the recognition and positive assessment of our work.

---

### Decision · Program_Chairs · 2025-09-17

**Decision:**

Accept (poster)

**Comment:**

This paper proposes a single-loop actor–critic method (AC-SMFG) for Stackelberg mean-field games and proves $\tilde{O}(k^{-1/2})$ non-asymptotic convergence under a gradient-alignment condition, avoiding nested bilevel loops. Empirically, it shows faster, smoother convergence on standard SMFG benchmarks, including a neural function-approximation variant.

During discussion, reviewers mainly flagged clarity (notably the gradient-alignment assumption and proof sketch), scope/novelty relative to prior single-loop bilevel work, and whether results extend beyond tabular policies. The authors responded by adding intuition and examples for the assumption, a concise technique sketch after the main theorem, and clarifying that tabular choices are for analysis while providing additional neural results. Remaining concerns (e.g., regularization regime, homogeneous followers) are acknowledged and have limited impact on the core contribution.

Overall, the theoretical advance is solid and timely, the algorithm is practical, and the added clarifications strengthen the presentation. As a result, I recommend accept.